# LIMIT ANALYSIS FOR SYMBOLIC MULTI-STEP REASONING TASKS WITH INFORMATION PROPAGATION RULES BASED ON TRANSFORMERS

## ABSTRACT

Transformers are able to perform reasoning tasks, however the intrinsic mechanism remains widely open. In this paper we propose a set of information propagation rules based on Transformers and utilize symbolic reasoning tasks to theoretically analyze the limit reasoning steps. We show that the limit number of reasoning steps is between $O(2^{L-1})$ and $O(3^{L-1})$ for a model with $L$ attention layers in a single-pass.

## 1 INTRODUCTION

The transformer architecture introduced by (Vaswani et al., 2017) has demonstrated capabilities across a wide range of tasks (Liu et al., 2018; Devlin et al., 2019; Radford et al., 2019; Touvron et al., 2023; OpenAI, 2023), showing particularly significant progress in logical reasoning. These models can not only solve complex mathematical problems Davies et al. (2021) but have also reached performance levels comparable to top human contestants in the International Mathematical Olympiad (IMO) (Trinh et al., 2024). The reasoning capabilities of large language models are fundamentally shaped by the thinking strategies they employ. Widely adopted approaches include Chain-of-Thought (CoT) (Wei et al., 2022b), Tree-of-Thought (ToT) (Yao et al., 2023), and Diagram-of-Thought (DoT) (Zhang et al., 2024a). While these strategies substantially improve multi-step logical reasoning accuracy by prompting models to generate explicit intermediate reasoning steps, they often exhibit an over-thinking phenomenon that consumes excessive computational resources and increases response time. This inefficiency highlights a critical question: what is the intrinsic single-pass reasoning capacity of these models? Specifically, how many reasoning steps can a model effectively execute without requiring iterative prompting or external scaffolding?

In this work, in order to more intuitively understand the mechanism of multi-round information transmission in Transformer models and the boundary of information transmission capacity, we use a symbolic dataset that can simulate a class of multi-step reasoning to conduct our study(Wang et al., 2025a). And starting from the information propagation patterns induced by the attention mechanism in Transformer architectures, we propose a set of information propagation rules that faithfully simulate how Transformers transmit information. We then rigorously analyze, under these propagation rules, the theoretical limit of the reasoning depth that can be achieved by an $L$-layer Transformer with a sufficiently large hidden dimension in a single forward pass. Our analysis shows that the achievable number of reasoning steps lies between $O(2^{L-1})$ and $O(3^{L-1})$. This result follows from two key structural properties of Transformers: (i) in each layer, tokens can execute reasoning operations in parallel; and (ii) each embedding vector in a hidden layer can encode information from multiple tokens in distinct sublinear subspaces. Finally, we validate our theoretical analysis with numerical experiments, showing that a trained 3-layer Transformer with a sufficiently large hidden dimension can attain the theoretically predicted maximum reasoning depth.

## 2 TRANSFORMER AND REASONING MECHANISM

### 2.1 TRANSFORMER ARCHITECTURE

We investigate a decoder-only Transformer with $L$-layer attention blocks. For integer $n$, given any sequence $(x_i)_{1\leqslant i\leqslant n}$, we denote its one-hot encoding [1] as $\boldsymbol{X}^{\text{in}} \in \mathbb{R}^{n\times d}$ with $d$ as the dictionary size.

The model first applies an embedding layer including both token embedding and positional encoding to obtain the input representation as $\boldsymbol{X}^{(0)} = \boldsymbol{X}^{\text{emb}} + \boldsymbol{X}^{\text{pos}} \in \mathbb{R}^{n\times d_m}$. Moreover, we denote the set of word embeddings of each word in the dictionary as $W^E$. We shall use the single-head attention in each layer which is computed as follows:

$$\mathcal{A}^{(l)}(\boldsymbol{X}^{(l)}) = \text{SoftMax}\left(\frac{\text{mask}(\boldsymbol{X}^{(l)}\boldsymbol{W}^{q(l)}\boldsymbol{W}^{k(l),\mathsf{T}}\boldsymbol{X}^{(l),\mathsf{T}})}{\sqrt{d_k}}\right),$$

$$\boldsymbol{X}^{\text{qkv}(l)} = \mathcal{A}^{(l)}(\boldsymbol{X}^{(l)})\boldsymbol{X}^{(l)}\boldsymbol{W}^{v(l)}\boldsymbol{W}^{o(l)},$$

where $0 \leqslant l \leqslant L$ and $\tilde{\sigma}$ is the softmax operator. For simplicity of expression, we will abbreviate $\boldsymbol{W}^{q(l)}\boldsymbol{W}^{k(l),\mathsf{T}}$ as $\boldsymbol{W}^{qk(l)}$ and $\boldsymbol{W}^{v(l)}\boldsymbol{W}^{o(l),\mathsf{T}}$ as $\boldsymbol{W}^{vo(l)}$ in the following text. Also, we ignore the normalization coefficient $\sqrt{d_k}$ in later sections for notational simplicity. The output of the $(l+1)$-th layer is obtained as:

$$\boldsymbol{X}^{\text{ao}(l)} = \boldsymbol{X}^{(l)} + \boldsymbol{X}^{\text{qkv}(l)}, \quad \boldsymbol{X}^{(l+1)} = \text{LayerNorm}(f^{(l)}(\boldsymbol{X}^{\text{ao}(l)}) + \boldsymbol{X}^{\text{ao}(l)}),$$

where $f^{(l)}(\cdot)$ represents the feedforward neural network of the $(l+1)$-th layer. The final output (in the form of token indices within the vocabulary) is obtained as:

$$\boldsymbol{Y} = \text{SoftMax}(\boldsymbol{X}_n^{(L)}\boldsymbol{W}^p) \in \mathbb{R}^d.$$

### 2.2 INDUCTION REASONING MECHANISM

Based on numerous works on In-Context Learning, Induction Heads (Brown et al., 2020; Garg et al., 2022; Bietti et al., 2024; Nichani et al., 2024), and recent studies on multi-step reasoning (Wang et al., 2025a; Yu et al., 2025), the reasoning capability of Transformers can be largely attributed to a mechanism called the Buffer Mechanism for storing diverse information, together with adjacent position matching and same token matching for achieving information matching and transmission.

**Buffer Mechanism** The Buffer Mechanism is a crucial way for Transformers to store multiple pieces of information (Wang et al., 2025a). Specifically, the interaction of information among tokens in a Transformer occurs in the attention module. Figure 1(a) illustrates the information flow of a 3-layer model performing 2-step reasoning, i.e., given a sentence of the form "... [a] [b] ... [b] [c] ... [a]", the model is required to output [c]. The dashed lines denote residual connections, while the solid lines denote information propagation induced by the attention mechanism. When a token (e.g., [b]) attends to a previous token (e.g., [a]), its next-layer state is not simply [a]+[b], but rather [a]$\boldsymbol{W}^{vo}$ + [b]. In other words, the Transformer stores the two pieces of information into subspaces spanned by different matrices through a linear transformation.

**Adjacent Position Matching** Similar to humans, language models rely heavily on the immediately preceding word when predicting the next word (Barbero et al., 2024). That is, the model can leverage positional encodings to establish connections between adjacent tokens. In fact, constructing such an attention weight matrix is not difficult. Assuming the positional encodings approximately satisfy $\boldsymbol{p}_i^{\mathsf{T}}\boldsymbol{p}_i = 1, \boldsymbol{p}_i^{\mathsf{T}}\boldsymbol{p}_j = 0, i \neq j$, it suffices to construct:

$$\boldsymbol{W}^{qk} = \sum_{i=1}^{[l_{\text{seq}}/2]} \boldsymbol{p}_{2i}\boldsymbol{p}_{2i-1}^{\mathsf{T}}, \tag{1}$$

$$(x_{2i} + \boldsymbol{p}_{2i})\boldsymbol{W}^{qk}(x_{2i-1} + \boldsymbol{p}_{2i-1})^{\mathsf{T}} \approx 1, \tag{2}$$

Clearly, by this method, we can construct attention between any adjacent tokens of fixed length. However, due to the inherent diversity of language tasks, only the attention between the most adjacent pair is the most salient. We refer to this mechanism as adjacent position matching.

---

[1]One-hot encoding is a technique that represents categorical data as binary vectors, where only one bit is set to 1 others are set to 0.

**Same Token Matching** Same token matching is the most essential mechanism within induction heads. Its existence grants Transformers strong out-of-distribution generalization ability. As shown in the Figure 1(a), because both nodes in the first layer contain the same information $[a]$, they can attend to each other via the same token matching mechanism. Specifically, it suffices that the weight matrices satisfy $\boldsymbol{W}^{qk(1)}\boldsymbol{W}^{vo(0),\mathsf{T}} = I$, in which case

$$q([a])k([a]\boldsymbol{W}^{vo(0)} + [b])^{\mathsf{T}} \approx [a]\boldsymbol{W}^{qk(1)}\boldsymbol{W}^{vo(0),\mathsf{T}}[a]^{\mathsf{T}} = [a][a]^{\mathsf{T}} \approx 1, \qquad (3)$$

That is, the final token node will allocate nearly all of its attention to the previous node containing the same information $[a]$, thereby transmitting $[b]$ to the final node in the next layer. In this way, a single-step reasoning is achieved. Multi-step reasoning follows the same principle.

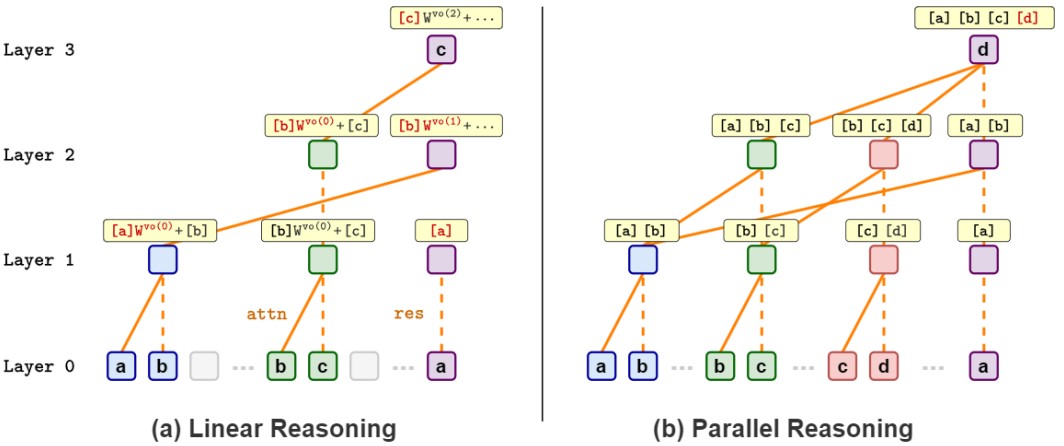

Figure 1: Illustration of linear reasoning and parallel reasoning.

## 2.3 PARALLEL REASONING

However, we note that the above-described mode, where each layer performs only one step of reasoning, is far from the upper limit of the Transformer model. As shown in Figure 1(b), adjacent position matching and same token matching can occur multiple times within a single layer, thereby enabling even shallow Transformer models to perform multi-step reasoning. We refer to this phenomenon as parallel reasoning. The central question considered in this paper is: given only adjacent position matching and same token matching, what are the upper and lower bounds of the parallel reasoning step that a transformer with $L$ layers attention blocks can perform?

## 3 INFORMAL THEOREMS

To investigate the above question, we first consider the simplest case in which all reasoning relations are arranged sequentially. As shown in Figure 2(a), the information flow of reasoning in this setting exhibits a clear "binary tree" structure. It then follows directly that the reasoning steps scale as $O(2^{L-1})$. In what follows, we will provide a rigorous proof of this result by mathematical induction. We note that permuting the order of reasoning pairs within the sequential arrangement does not disrupt the flow of sequential reasoning. Hence, when logical relations are arranged in sequence, the reasoning steps of a Transformer constitute the lower bound among all possible cases.

On the other hand, we observe that for a 3-layer model, when the data are arranged as illustrated in Figure 2(b), the final layer carries the maximum amount of information. The data in this case exhibit an evident fractal structure. The advantage of such a configuration is that each local terminal node can simultaneously match two preceding nodes by leveraging both the maximum and minimum information it carries, thereby expanding its information content. Consequently, the reasoning steps scale as $O(3^{L-1})$. Therefore, we arrive at the following informal conclusion:

**Theorem 3.1** (Informal Corollary 6.4). *The maximal number of reasoning steps a transformer with $L$ layers attention blocks can perform has a lower bound $O(2^{L-1})$ and an upper bound $O(3^{L-1})$.*

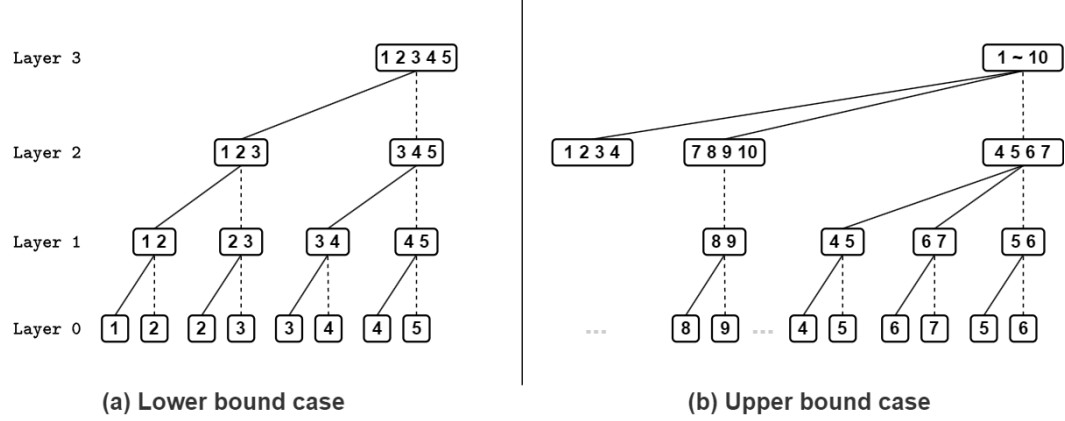

(a) Lower bound case        (b) Upper bound case

Figure 2: Example of lower bound and upper bound of parallel reasoning.

Next, we will provide a formal statement of the problem and a rigorous proof of the conclusion.

## 4 SYMBOLIC REASONING TASK

In this section, we give a brief introduction to the reasoning task and some related definitions. Moreover, we shall introduce the rules of information propagation.

A reasoning task typically involves a question and an answer to that question, along with the rule and process to get the answer. For example, given $A_1 \subseteq A_2$ and $A_2 \subseteq A_3$, the question is the relation of $A_1$ and $A_3$, and the answer is $A_1 \subseteq A_3$. We shall use a more symbolic way to express reasoning tasks as in the following example in figure 3.

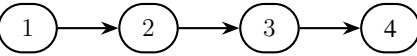

Figure 3: Three steps reasoning task. One step reasoning leads to 2, two steps reasoning leads to 3.

We use a sequence $(1, 2, 2, 3, 3, 4)$ to denote this reasoning task. Indeed, this sequence is composed of three bigrams $(1, 2)$, $(2, 3)$, $(3, 4)$, and each bigram represents one step of reasoning. We call these bigrams reasoning pairs, which we will define below.

**Definition 4.1.** *A reasoning pair is an element in $\mathbb{Z}^2$ of the form $\boldsymbol{a}_i = (\boldsymbol{a}_i^1, \boldsymbol{a}_i^2)$ where $i \in \mathbb{Z}$, $\boldsymbol{a}_i^1, \boldsymbol{a}_i^2 \in \mathbb{Z}$ and $\boldsymbol{a}_i^1 \neq \boldsymbol{a}_i^2$.*

The set of all reasoning pairs is denoted as $\mathcal{A}$. $a_i^1 \to a_i^2$ represents one step of reasoning.

It is natural that we shall define the $s$ step reasoning chain as follows.

**Definition 4.2.** *An $s$ step reasoning chain is a finite sequence $(\boldsymbol{a}_i)_{1 \leqslant i \leqslant s}$ with $\boldsymbol{a}_i \in \mathcal{A}$, and it shall satisfy the following conditions:*

- *$\boldsymbol{a}_i^2 = \boldsymbol{a}_{i+1}^1$ for $1 \leqslant i \leqslant s - 1$;*

- *For any subsequence $(\boldsymbol{a}_{i_m})_{i_m \in \mathcal{I}}$ of $(\boldsymbol{a}_i)_{1 \leqslant i \leqslant s}$, we have $\boldsymbol{a}_{\min\{\mathcal{I}\}}^1 \neq \boldsymbol{a}_{\max\{\mathcal{I}\}}^2$ (no loop), where $\mathcal{I}$ is a subset of $\{1, 2, \cdots, s\}$ containing at least two elements.*

The first condition ensures that the reasoning chain does not break before the final step, and the second condition ensures that there is no loop of arbitrary size in the reasoning task. For example, the sequence $((1, 2), (2, 3), (3, 1))$ and $((1, 2), (3, 4), (4, 5))$ are not reasoning chains.

Similarly, the subscript $i$ in this definition can take values from an infinite set like $\mathbb{Z}$, thereby defining an infinite chain of reasoning.

**Remark 4.3.** *For notational simplicity, here and in the sequel, we shall write $(\boldsymbol{a}_i)$ for $(\boldsymbol{a}_i)_{i \in I}$ when the index set $I$ is clear from the context.*

~~We shall also consider the case when the reasoning chain is of infinite length.~~

~~**Definition 4.4.** *A sequence $(\boldsymbol{a}_i)_{i \in \mathbb{Z}}$ is called a reasoning chain if it satisfies the following conditions:*~~

- ~~$\boldsymbol{a}_i \in \mathcal{A}$;~~

- ~~$\boldsymbol{a}_i^2 = \boldsymbol{a}_{i+1}^1$ for $i \in \mathbb{Z}$;~~

- ~~*For any subsequence $(\boldsymbol{a}_{i_m})_{i_m \in \mathcal{I}}$ of $(\boldsymbol{a}_n)$, we have $\boldsymbol{a}_{\min\{\mathcal{I}\}}^1 \neq \boldsymbol{a}_{\max\{\mathcal{I}\}}^2$, where $\mathcal{I} \subseteq \mathbb{Z}$ containing at least two elements.*~~

~~Note that for any $i_0, s \in \mathbb{Z}$, we can truncate the reasoning chain $(\boldsymbol{a}_n)$ as $\tilde{\boldsymbol{a}}_k = \boldsymbol{a}_{i_0+k-1}$, $1 \leqslant k \leqslant s$ to get an $s$ step reasoning chain $(\tilde{\boldsymbol{a}}_k)$.~~

In practice, a sentence may consist of reasoning pairs which are not in order. Due to the mask condition which is common in the LLM, the order of reasoning pairs may influence the information propagation. To describe the order of these reasoning pairs and their relation to the reasoning chain, we define the elements of a symmetric group as permutations, and focus on the symmetric group $\mathrm{Sym}(\mathbb{Z})$.

~~To describe the order of these reasoning pairs and their relation to the reasoning chain, we need to introduce the concept of permutation.~~

~~**Definition 4.5.** *A symmetric group $\mathrm{Sym}(S)$ on a countable set $S$ is a group whose elements are all bijective maps from $S$ to $S$ and whose group operation is that of function composition.*~~

~~The elements of a symmetric group are called permutations. And we shall focus on $\mathrm{Sym}(\mathbb{Z})$.~~

**Definition 4.6.** *Given a reasoning chain $(\boldsymbol{a}_m)_{m \in \mathbb{Z}}$ and a permutation $\sigma \in \mathrm{Sym}(\mathbb{Z})$, a sequence $(x_i)_{i \in \mathbb{Z}}$ is called a reasoning sequence constructed from $(\boldsymbol{a}_m)_{m \in \mathbb{Z}}$ and $\sigma$ if it satisfies:*

$$x_i = \boldsymbol{a}_{\sigma(\lfloor \frac{i+1}{2} \rfloor)}^{2-(i \bmod 2)}. \tag{4}$$

*Also, $(\boldsymbol{a}_m)_{m \in \mathbb{Z}}$ and $\sigma$ are called the constructing reasoning sequence and constructing permutation of $(x_i)$, respectively.*

When referring to a reasoning sequence $(x_i)$, we are actually denoting the tuple $((x_i), (\boldsymbol{a}_m), \sigma)$. Moreover, if $\sigma = \mathrm{Id}$, then the reasoning sequence is called a sorted reasoning sequence. Note that from the relation (4) we also have

$$\boldsymbol{a}_i = \boldsymbol{a}_{\sigma(\sigma^{-1}(i))} = (x_{2\sigma^{-1}(i)-1}, x_{2\sigma^{-1}(i)}), \tag{5}$$

where $\sigma^{-1}$ is the inverse of $\sigma$ satisfying $\sigma \circ \sigma^{-1} = \sigma^{-1} \circ \sigma = \mathrm{Id} \in \mathrm{Sym}(\mathbb{Z})$.

Likewise, in the case where the index ranges over a finite set, the reasoning sequence will be of finite length.

**Remark 4.7.** *In the definition of reasoning sequence we use the permutation to change the order of reasoning pairs which does not break the relation inside each reasoning pair. Moreover, no permutation should be applied to the original sequence $(x_i)$. For example, the sequence $(1, 2, 2, 3, 3, 4)$ can be $(2, 3, 3, 4, 1, 2)$ or $(3, 4, 2, 3, 1, 2)$ under some certain permutations. Both of these sequences are related to the reasoning chain $((1, 2), (2, 3), (3, 4))$. However, it cannot be transformed into $(1, 3, 3, 4, 2, 3)$ through any permutation that acts on reasoning pairs.*

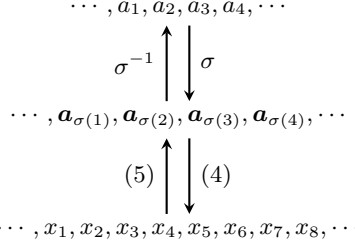

Figure 4: Relationship between reasoning chain and reasoning sequence.

Similarly, we shall also use a reasoning sequence of finite length.

**Definition 4.8.** *An $s$ step reasoning sequence $(x_i)_{1 \leqslant i \leqslant 2s}$ with constructing reasoning chain $(\boldsymbol{a}_m)_{1 \leqslant m \leqslant s}$ and constructing permutation $\sigma$ is defined as:*

$$x_i = \boldsymbol{a}_{\sigma(\lfloor \frac{i+1}{2} \rfloor)}^{2-(i \bmod 2)}, \text{ for } 1 \leqslant i \leqslant 2s. \tag{6}$$

**Example 4.9.** *The sequence $(x_i)_{i \geqslant 1} = (\lfloor \frac{i}{2} \rfloor)_{i \geqslant 1}$ can be seen as a sorted reasoning sequence with constructing permutation $\sigma = \text{Id}$ and constructing reasoning chain $((0,1),(1,2),(2,3),(3,4),\cdots)$.*

**Example 4.10.** *The sequence $(x_i) = (1,2,6,3,2,4,3,5,4,6)$ with constructing reasoning chain $(\boldsymbol{a}_m) = ((1,2),(2,4),(4,6),(6,3),(3,5))$ and constructing permutation $\sigma$ satisfying $\sigma(1) = 1$, $\sigma(2) = 4$, $\sigma(3) = 2$, $\sigma(4) = 5$, $\sigma(5) = 3$. In this example,*

$$\boldsymbol{a}_1 = (1,2) = (x_1, x_2), \quad \boldsymbol{a}_2 = (2,4) = (x_5, x_6), \quad \boldsymbol{a}_3 = (4,6) = (x_9, x_{10}).$$

Figure 5: Reasoning task represents by $(x_i)$ in example 4.10.

When considering a finite step reasoning sequence, the concept of reasoning start, which indicates where the reasoning task should begin, is also needed. More specifically, we consider an $s$ step reasoning sequence $(x_i)_{1 \leqslant i < 2s}$ with constructing reasoning chain $(\boldsymbol{a}_m)_{1 \leqslant m \leqslant s}$ and constructing permutation $\sigma$. Then one more element $x_{2s+1}$ is added to the end of the reasoning sequence $(x_i)$, and $x_{2s+1} = \boldsymbol{a}_{m_0}^1$ for some $\boldsymbol{a}_{m_0} \in (\boldsymbol{a}_m)$.

**Example 4.11.** *We set the sequence $(x_i) = (1,2,6,3,2,4,3,5,4,6,4)$ with constructing reasoning chain $(\boldsymbol{a}_m)$ and constructing permutation $\sigma$ as in example 4.10. In addition, the reasoning start is set to be $4$. Then one step reasoning result is $6$ and two steps reasoning result is $3$.*

Next we define nodes which serves as containers of information.

**Definition 4.12.** *A node corresponding to a reasoning sequence $(x_i)$ is a set of two sets. More specifically, for $i \in \mathbb{N}$, an $l$th layer node is defined as $N_i^l = \{V_i^l, I_i^l\}$ where $V_i^l$ is called a value set whose elements are integers, and $I_i^l \subseteq \mathbb{Z}$ is an index set. Also, we require that $V_i^l = \bigcup_{i_\alpha \in I_i^l} \{x_{i_\alpha}\}$.*

We define the information quantity of a node $N_i^l = \{V_i^l, I_i^l\}$ as $C_i^l = |V_i^l|$. Moreover, we denote $\mathcal{N}^l$ as the set of all $l$th layer nodes.

We denote the information propagation between two nodes as $N_i^{l+1} = N_m^l \star N_i^l$ where $N_m^l \star N_i^l := \{V_m^l \cup V_i^{l+1}, I_m^l \cup I_i^{l+1}\}$. The case $i \neq m$ represents the attention mechanism, more specifically, the node $N_m^l$ attends to the node $N_i^l$ and the result is stored in the node $N_i^{l+1}$. The case $i = m$ represents the residual connection, in which case $N_i^{l+1} = N_i^l$. Moreover, there may be more than one node transmitting information to a node $N_i^l$. In this case, we denote the set of all such nodes as $N_{\mathcal{I}}^l \subseteq \mathcal{N}^l$ where $\mathcal{I}$ is some index set. The information propagation process in this case is then defined by $N_i^{l+1} = N_{\mathcal{I}}^l \star N_i^l := \{\bigcup_{i_\alpha \in \mathcal{I}} V_{i_\alpha}^l \cup V_i^{l+1}, \bigcup_{i_\alpha \in \mathcal{I}} I_{i_\alpha}^l \cup I_i^{l+1}\}$.

## 5 RULES OF INFORMATION PROPAGATION

We can extract the following rules of information propagation from the behavior of transformer as follows.

- Rule 0 (Initial setup): The nodes in 0th layer are constructed as $N_i^0 = \{\{x_i\}, \{i\}\}$. For $l \geqslant 1$, the $l$th layer of nodes are initially constructed as $N_i^l = \{\emptyset, \emptyset\}$.

- Rule 1 (Mask Condition): Attention happens only from former nodes to later nodes. That is, the attention mechanism $N_i^{l+1} = N_m^l \star N_i^l$ is performed only when $m < i$. For the multiple nodes information transmission case, the operation $N_i^{l+1} = N_{\mathcal{I}}^l \star N_i^l$ is performed only when $i_\alpha < i$ for all $i_\alpha \in \mathcal{I}$.

- **Rule 2 (Adjacent position matching):** For $l = 1$, the information in an odd position node can be transmitted to the subsequent even position node. In this case, the mask condition is satisfied automatically. More specifically, the nodes in 1st layer are of the form $N_{2i}^1 = N_{2i-1}^0 \star N_{2i}^0$ after position matching.

- **Rule 3 (Same token matching):** For $l \geqslant 2$, a node $N_i^l$ is updated as $N_i^l = N_{\mathcal{I}}^{l-1} \star N_i^{l-1}$ provided there exists a set $N_{\mathcal{I}}^{l-1} \subseteq \mathcal{N}^{l-1}$ satisfying the mask condition $i_\alpha < i$ and $V_{i_\alpha}^{l-1} \cap V_i^{l-1} \neq \emptyset$ for all $i_\alpha \in \mathcal{I}$.

- **Rule 4 (Residual Connection):** For $l \geqslant 1$, $\forall N_i^l \in \mathcal{N}^l$, $N_i^l = \{V_i^{l-1} \cup V_i^l, I_i^{l-1} \cup I_i^l\}$.

**Remark 5.1.** *With a slight abuse of notation, we still denote the value set and index set of a node $N_i^l$ as $V_i^l$ and $I_i^l$ after the information propagation process. For example, through residual connection $N_i^l$ is updated as $N_i^l = \{V_i^{l-1} \cup V_i^l, I_i^{l-1} \cup I_i^l\}$, and we still denote the sets $V_i^{l-1} \cup V_i^l$ and $I_i^{l-1} \cup I_i^l$ as $V_i^l$ and $I_i^l$, since we only care about the result after each layer's information propagation.*

**Remark 5.2.** *It makes no difference whether the residual connection happens before or after same token matching or position match. The result stored in the next layer remains unchanged.*

**Remark 5.3.** *In the above information propagation rules we require that the adjacent position matching only happens when $l = 1$ and the same token matching only happens when $l \geqslant 2$. We can also set the same token matching to happen when $l = 1$ and adjacent position matching to happen when $l \geqslant 2$, since the index set $I_i^l$ in fact encodes the position information. The necessary condition for adjacent position matching to happen is $\exists I_k^l$ and $I_j^l$ s.t. there exist $i_j \in I_j^l$ and $i_k \in I_k^l$ satisfying $\min i_k, i_j \bmod 2 = 1$ and $|i_j - i_k| = 1$. It is easy to see that the index set $I_i^l$ in this same token matching first rules plays the same role as $V_i^l$ in above adjacent position matching first rules, and there will be no essential difference for the result in our main theorems under these two different rules. For simplicity, we only consider the above adjacent position matching first rules.*

If value matches occurs in the first layer, then in subsequent layers, only position matches can lead to an increase in information. This is because the only way a value match can occur in later layers is if at least one of the nodes involved comes from the residual. This however, would not lead to the increase of information quantity.

A simple example: consider the task $a \to b \to c \to d \to e \to f$, and the corresponding reasoning sequence $(a, b, e, f, b, c, d, e, c, d)$, with positions denoted as $(1, 2, 3, 4, 5, 6, 7, 8, 9, 10)$. If a value match occurs in the first layer, the nodes in the second layer take the following form: $\{\{a\}, \{1\}\}, \{\{b\}, \{2, 5\}\}, \{\{c\}, \{6, 9\}\}, \{\{d\}, \{7, 10\}\}, \{\{e\}, \{3, 8\}\}, \{\{f\}, \{4\}\}$, along with some residual nodes.

Clearly, no value match can happen between these nodes. Moreover, value matches involving residual nodes do not result in an increase in the amount of information.

Similarly, if a position match occurs in the first layer, then in subsequent layers, only value matches can lead to an increase in the amount of information. Therefore, only two cases are important: position match first rules and value match first rules. Mixed scenarios need not be considered. Additionally, based on the explanation in Remark 5.3, we will only consider the position match first rules adopted in the subsequent discussion.

Two concepts of layer arise in this framework: the layer of attention blocks and the layer of nodes. The $l$th layer attention block takes $(l-1)$th layer of nodes as input and produces the $l$th layer of nodes as output. Due to this relation, we use "layer $l$" referring to the $l$th layer of attention blocks and $l$th layer of nodes interchangeably.

## 6 MAIN THEOREMS

In this section we analyze the information quantity in the process of information propagation according to the above information propagation rules, and our main theorem follows.

**Theorem 6.1.** *Under the rules of information propagation, given any reasoning sequence $((x_i)_{i \in \mathbb{Z}}, (\boldsymbol{a}_m)_{m \in \mathbb{Z}}, \sigma)$, for any given $x \in \{x_i\}_{i \in \mathbb{Z}}$, and for any $l \in \mathbb{Z}^+$ there exists $i \in \mathbb{Z}^+$ such that $x \in V_i^l$, and we have the following bound for $T^l(x) = \max_{i \in \mathbb{Z}} \{C_i^l \mid x \in V_i^l\}$:*

$$2^{l-1} + 1 \leqslant T^l(x) \leqslant 3^{l-1} + 1. \tag{7}$$

The whole proof is based on mathematical induction. Here we only give a sketch of the proof. The complete proof can be found in the appendix B.

Given a reasoning sequence $((x_i), (a_m), \sigma)$, when considering the lower bound, by Rule 2 the value sets of nodes in layer 1 contain only one reasoning pair except the case where only residual connection happens. Suppose that $j < i$ and the node $N_i^1$ contains $a_{m_i}^1, a_{m_i}^2$ as value set, or simply we say $N_i^1$ contains $a_{m_i}$, and $N_j^1$ contains $a_{m_j}$. Two cases may happen, $n_i + 1 = n_j$ and $a_{m_i}^2 = a_{m_j}^1$ or $n_j + 1 = n_i$ and $a_{m_j}^2 = a_{m_i}^1$. Both cases will lead to $N_i^2$ containing $a_{m_i}$ and $a_{m_j}$ by Rule 3. This process is the same for any other two nodes containing two adjacent reasoning pairs respectively. The process for layer 3 is analogous to that for layer 2: information contained in two nodes in layer 2 is propagated to one node in layer 3. The whole structure is in fact a binary tree and hence the bound is powers of 2.

Regarding the upper bound, due to the mask condition (Rule 1), the permutation $\sigma$ may affect the information propagation. However, the upper bound is always bounded by the case when the mask condition is lifted. Therefore, we ignore the mask condition to find the upper bound. Just like the proof of the lower bound which use two adjacent reasoning pairs, now we use three adjacent reasoning pairs. Suppose the nodes $N_i^1$, $N_j^1$ and $N_k^1$ contain the reasoning pairs $a_{m_i}$, $a_{m_j}$ and $a_{m_k}$ respectively and $m_i + 1 = m_j = m_k - 1$. Then by Rule 3, at least one of the node $N_j^2$ contains $a_{m_i}$, $a_{m_j}$ and $a_{m_k}$. As for layer 3, there are nodes that contain information propagated from three nodes like $N_j^3$. The whole structure is a ternary tree and hence the bound is powers of 3.

**Theorem 6.2.** *Under the rules of information propagation (5), for any $s$ step reasoning sequence $((x_i)_{1 \leqslant i \leqslant 2s}, (a_m)_{1 \leqslant m \leqslant s}, \sigma)$ with reasoning start $x_{2s+1}$, for any $l$ satisfying that $1 \leqslant l \leqslant 1 + \log_2 s$, we have the following estimate for $C_{2s+1}^l$.*

$$2^{l-1} \leqslant C_{2s+1}^l \leqslant 3^{l-1}. \tag{8}$$

The proof of this theorem is similar to the proof of Theorem 6.1 with two main differences. First, we need to take into account the reasoning start which requires using mathematical induction twice: once on the node of reasoning start and once on the nodes in the sequence. Second, the reasoning sequence is now finite, and we require it to be long enough to support the structure of the binary tree and the ternary tree. The complete proof is included in the Appendix C.

**Remark 6.3.** *In the proof of this theorem, the mask condition was relaxed to obtain the upper bound. However, we showed in Appendix D that there is a way to construct a large class of reasoning sequences such that the upper bound is attained under mask condition. This proves that the upper bound is indeed tight.*

**Corollary 6.4.** *For a given transformer with $L$ layers of attention blocks and an input sequence of $n = 2s + 1$, where $s \in \mathbb{Z}^+$ and $1 \leqslant L \leqslant 1 + \log_2 s$, the maximal number of reasoning steps $S_p$ it can perform satisfies the following bounds:*

$$2^{L-1} - 1 \leqslant S_p \leqslant \frac{3^{L-1} - 1}{2}. \tag{9}$$

**Remark 6.5.** *Although the upper bound on information quantity as we show in Theorem 6.2 is $3^{L-1}$, this only implies that up to $3^{L-1} - 1$ reasoning steps are involved. However, since the process does not track information back through the reasoning chain, only $\frac{3^{L-1}-1}{2}$ reasoning steps are effective in the reasoning tasks. A quick example is the sequence $(0, 1, 1, 2, 1)$ with reasoning start $1$. For $l = 2$ there are two reasoning steps in total but only one effective reasoning step $1 \to 2$.*

## 7 EXPERIMENTS AND DISCUSSIONS

### 7.1 SIMULATION OF INFORMATION PROPAGATION

Considering that the real training process may introduce optimization errors and estimation errors, which can affect our assessment of the model's inherent approximation capability, we first use a simulation program—constructed based on the information propagation rules mentioned earlier—to simulate the process of information dissemination. Pseudocode 1 provides a detailed process of the Info-Prop Rules. Each `Node[i]` is a dictionary containing two attributes: pos (position) and val

---

**Algorithm 1** Information Propagation Rules

---

1: Initialize an empty dictionary Node
2: **for** each token i in the sequence **do**
3:     Node[i][pos] ← i
4:     Node[i][val] ← token i
5: **end for**
6: Node_new ← Node
7: Information propagation
8: **for** each node i in Node **do**
9:     **for** each node j > i in Node **do**
10:         rule1 ← any even number $a \in$ Node[i][pos] satisfies $a + 1 \in$ Node[j][pos]
11:         rule2 ← Node[i][val] ∩ Node[j][val] $\neq \varnothing$
12:         rule3 ← Node[i][pos] ∩ Node[j][pos] $\neq \varnothing$
13:         **if** rule1 or rule2 or rule3 **then**
14:             Node_new[j][pos] ← Node_new[j][pos] ∪ Node[i][pos]
15:             Node_new[j][val] ← Node_new[j][val] ∪ Node[i][val]
16:         **end if**
17:     **end for**
18: **end for**

---

(value). Node[i][pos] and Node[i][val] are two sets. Initially, each set contains only its own information and will be updated according to the Info-Prop Rules.

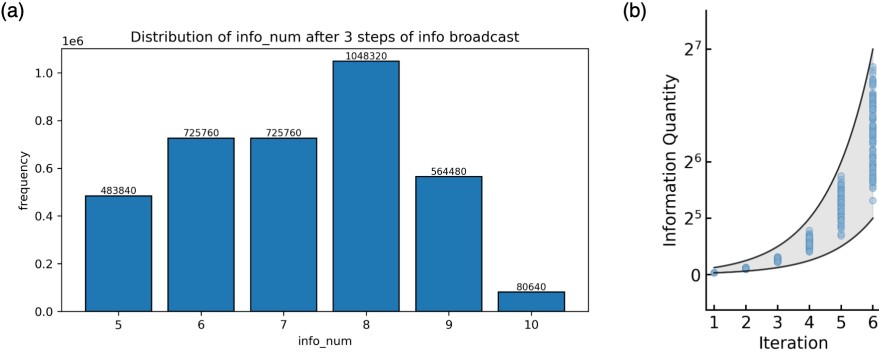

Figure 6: Simulation results based on the information propagation rules: (a) the statistical distribution of the information content at the last node for the case of L=3; (b) the growth trend of the information content stored in the last node with respect to the number of iterations, based on simulations over 1000 random sequences.

For the case of L=3, we enumerate all possible serialized representations of the reasoning chain $a_1 \to a_2 \to ... \to a_{11}$, resulting in 11!=3628800 distinct possibilities. Fig. 6(a) shows the statistical distribution of the information content at the last node across all these cases. We observe that the minimum information content of the last node is 5 and the maximum is 10, which agrees with the theoretical lower and upper bounds.

Fig. 6(b) shows the number of value information stored in the last node with respect to the number of iterations under our set of rules. It can be observed that the stored information exhibits an approximately exponential growth pattern with the number of iterations.

## 7.2 TRAINING TASK

To verify whether the Transformer model can achieve the expected reasoning capability, we conducted training specifically on Transformer architectures. As the number of layers increases, both the required amount of reasoning and the model capacity grow exponentially, leading to significant optimization difficulty. Therefore, we primarily focus on training a 3-layer Transformer model.

Our task aligns with the reasoning sequence described earlier. Specifically, we begin with a reasoning sequence denoted as $(x_i)_{1 \leq i \leq 2s}$. Subsequently, we introduce $x_{2s+1}$ as the starting point. The complete sequence $(x_i)_{1 \leq i \leq 2s+1}$ serves as the input to the transformer. The output corresponds to the reasoning result from the starting point with a fixed reasoning step (an example in Fig. 7).

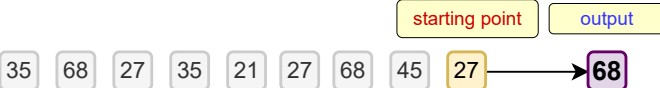

Figure 7: A two-step reasoning example.

The detailed training settings are provided in the appendix F. Reasoning pairs in the test set are totally different from the training set in order to prove transformer can learn reasoning instead of remembering these sequences. As predicted by our theoretical analysis, this model is capable of solving 3-step reasoning problems with perfect accuracy. Furthermore, we observe that the model dimension $d_m$ in our construction is notably large and a higher hidden dimension aids in storing more intermediate information. To validate these findings, we investigate whether a transformer can be trained to achieve exact accuracy on 3-step reasoning tasks and whether a large $d_m$ is indeed necessary. Our experiments confirm that such a model can be successfully trained, and that a sufficiently large $d_m$ is critical for achieving optimal performance. The relationship between the model dimension $d_m$ and test accuracy is summarized in the table below.

Table 1: Relationship between $d_m$ and test accuracy

| $d_m$ | 64 | 128 | 256 | 512 | 1024 |
|---|---|---|---|---|---|
| Test accuracy (%) | 9.6 | 11.9 | 72.2 | 99.6 | 99.6 |

As shown in Table 1, test accuracy increases monotonically with dimension $d_m$, eventually approaching $100\%$, which confirms that a 3-layer transformer can solve 3-step reasoning tasks.

We further investigate scenarios where the model exhibits partial or complete failure. According to the theoretical analysis presented earlier Corollary 6.4, for reasoning step lengths in the interval $[2^{L-1}, \frac{3^{L-1}-1}{2}]$, the model can produce correct answers under certain sequence ordering conditions. However, when the number of steps exceeds $\frac{3^{L-1}-1}{2}$, information propagation to the final token becomes insufficient, resulting in incorrect answers. Since this implies that high accuracy is unattainable in such regimes, experimental validation remains partial.

We consider a 3-layer Transformer. Given that $2^{3-1} = 4$ and $\frac{3^{3-1}-1}{2} = 4$, the model can solve 4-step reasoning tasks when sequence order conditions are satisfied, but fails for 5-step reasoning. Experimental results under these settings are as follows:

- For 4-step reasoning, the model achieves a test accuracy of 46.1%.

- For 5-step reasoning, the test accuracy drops to 25.1%.

The decrease in accuracy for the 5-step case provides empirical support for theoretical results. We also find that if the reasoning pairs satisfy a proper order, the network can obtain accurate results for 4-step cases 12 but not for 5-step cases **??**. Complete training curves and additional experiments are provided in the Appendix F.

**LLM / AI Tool Disclosure.** During the preparation of this manuscript, we used DeepSeek-V3 (July 2024) to assist with language polishing (grammar, wording). All substantive content, ideas, and core writing remain the authors' own, and the authors are fully responsible for any error including those introduced during polishing.

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

# A    RELATED WORK

In-context learning (ICL) was first introduced by Brown et al. (2020). This was subsequently followed by numerous studies (Olsson et al., 2022; Garg et al., 2022; Wang et al., 2022; Müller et al., 2021; Goldowsky-Dill et al., 2023; Bietti et al., 2024; Nichani et al., 2024; Edelman et al., 2024; Chen et al., 2024; Todd et al., 2023; Chen and Zou, 2024) to investigate the ICL using induction heads, which can be seen as a special case of one-step reasoning.

The ability of Transformers to perform complex reasoning is supported by theoretical work on their expressiveness. Prior research has shown that Transformers are Turing-complete in theory (Pérez et al., 2019) and are universal approximators of sequence-to-sequence functions (Yun et al., 2019). However, there is a known gap between this theoretical capability and what Transformers can reliably learn from data (Wei et al., 2022a), and their representational power has specific strengths and limitations Sanford et al. (2023).

To study multi-step reasoning in practice, various reasoning tasks have been proposed, such as recognizing context-free grammars (Zhao et al., 2023), learning sparse functions (Edelman et al., 2022), learning compositionality (Hupkes et al., 2020), generalizing out of distribution when learning Boolean functions (Abbe et al., 2024), matrix digits task (Webb et al., 2023), SET game tasks (Altabaa et al., 2023), reasoning tasks designed by anchor function (Zhang et al., 2024c). Kil et al. (2024); Li et al. (2024a) use Chain-of-Thought (CoT) reasoning (Wei et al., 2022b) to achieve multi-step reasoning via prompt engineering. Zhang et al. (2023) introduced the beam retrieval framework for multi-hop QA improving the few-shot QA performance of LLMs. Li et al. (2024b); Yang et al. (2024); Shalev et al. (2024) locate the potential intermediate answers within middle layers which play a causative role in shaping the final explicit reasoning results. Wang et al. (2025a) use a symbolic dataset to study the mechanism of multi-step reasoning in Transformer models. Their work focuses on serial reasoning, where each layer performs one step of reasoning. As an extension of that work, we adopt the same dataset setting, but investigate the mechanism of parallel reasoning. Zhang et al. (2024b); Yao et al. (2025) show that with small initialization, Transformers in condense regime (Luo et al., 2021; Xu et al., 2025) can learn reasoning better. Wang et al. (2025b) investigated the $k$-fold task which is similar to the k-hop induction head task in Sanford et al. (2024).

Sanford et al. (2024) focus on characterizing the minimum depth required to complete L-step reasoning, proving under an idealized setting that ignores attention masks and feed-forward networks that a Transformer with depth $O(\log L)$ can achieve $L$-step reasoning, thereby providing a lower bound on the number of layers for reasoning tasks. In contrast, our work adopts architectural assumptions more consistent with practical Transformers (including attention masks and FNNs) and studies explicit upper and lower bounds on the number of reasoning steps that a Transformer can complete under fixed-depth conditions. More importantly, we reveal a key structural factor previously unaddressed in the literature: even when atomic facts correspond to the same reasoning chain, different input orderings can lead to significant differences in the model's reasoning ability, indicating that the upper bound of reasoning depends not only on the model's structural capacity but is also profoundly influenced by the symbolic order structure of the input.

Guzmán et al. (2024) find that Transformers show limited generalization in recursivity and compositionality. Their experiments, however, use only a one-layer Transformer, which differs fundamentally from the multi-layer architectures analyzed in our work.

In terms of formal language frameworks, a conceptually related work, RASP (Weiss et al., 2021), proposes a formal language that describes the computation of Transformers through restricted attention operations. Unlike RASP, we further consider how information is distributed and stored across different subspaces. We leverage the nonlinear transformation capability of the feed-forward network (FFN) to perform subspace transitions, thereby enabling richer representations than those captured by RASP's linear attention structures.

# B    PROOF OF THEOREM 6.1

We first prove the lower bound.

According to Rule 0, the nodes in first layer are constructed as $N_i^1 = \{(x_i), \{i\}, \{i\}\}$. According to Rule 2, we have the nodes in second layer are constructed as $N_{2i}^1 = \{\{x_{2i-1}, x_{2i}\}, \{2i\}, \{2i-1, 2i\}\}$ for $i \in \mathbb{Z}$. Clearly, we have $C_{2i}^1 = 2$.

We use mathematical induction to prove that for $k \geqslant 2$, for any $x_j \in \{x_i\}_{i \in \mathbb{Z}}$, there exists an integer $i \in \mathbb{Z}$ such that

- $x_j \in V_i^k$,
- We denote $\sigma(I_i^k)$ to be the set $\{\sigma(\lfloor \frac{i_m+1}{2} \rfloor)\}_{i_m \in I_i^k}$, and we have $\max \sigma(I_i^k) - \sigma(\lfloor \frac{j+1}{2} \rfloor) \geqslant 2^{k-1} - 1$ which implies that $C_i^k \geqslant 2^{k-1} + 1$.

For $k = 2$, given any $x_j \in (x_i)$, according to Rule 2, we know that $x_j$ is contained in the value set $V_{2\lfloor \frac{j+1}{2} \rfloor}^1$ of node $N_{2\lfloor \frac{j+1}{2} \rfloor}^1$.

For simplicity, we assume that $j$ is an even number, then $V_{2\lfloor \frac{j+1}{2} \rfloor}^1 = V_j^1 = \{x_{j-1}, x_j\}$ and $I_j^1 = \{j-1, j\}$. We now want to find a node $N_2^i$ such that $V_i^1 \cap V_j^1 = \{x_j\}$. To do so, by definition of reasoning sequence and using the relation (4) we know that $x_{j-1}$ and $x_j$ comes from the reasoning pair $\boldsymbol{a}_{\sigma(\frac{j}{2})}$. By definition of reasoning sequence and relation (5) we have

$$
\begin{aligned}
\boldsymbol{a}_{\sigma(\frac{j}{2})}^2 &= \boldsymbol{a}_{\sigma(\frac{j}{2})+1}^1, \\
\boldsymbol{a}_{\sigma(\frac{j}{2})+1} &= \left( x_{2\sigma^{-1}\left(\sigma(\frac{j}{2})+1\right)-1}, x_{2\sigma^{-1}\left(\sigma(\frac{j}{2})+1\right)} \right).
\end{aligned}
\tag{10}
$$

Set $i = 2\sigma^{-1}\left(\sigma(\frac{j}{2})+1\right)$. It is now clear that the node $N_i^1$ have value set $V_i^1$ satisfying $V_i^1 \cap V_j^1 = \{x_j\}$ and index set $I_i^1 = \{i-1, i\}$. By Rule 3, the node $N_{\max\{i,j\}}^2$ have value set $V_{\max\{i,j\}}^2$ such that $\{x_{j-1}, x_j, x_{i+1}\} \subseteq V_{\max\{i,j\}}^2$. Hence, $C_{\max\{i,j\}}^2 \geqslant 3$ and $\sigma(\frac{j}{2})+1 - \sigma(\frac{j}{2}) = 1 \geqslant 2^{2-1} - 1$.

Now we assume the induction hypotheses hold for $3 \leqslant k \leqslant k_0$ and consider the case $k = k_0 + 1$. There exists a node $N_{m_0}^{k_0}$ such that

$$
x_j \in V_{m_0}^{k_0},
\tag{11}
$$

$$
\max \sigma(I_{m_0}^{k_0}) - \sigma(\frac{j}{2}) \geqslant 2^{k_0-1} - 1,
\tag{12}
$$

$$
C_{m_0}^{k_0} \geqslant 2^{k_0-1} + 1.
\tag{13}
$$

Set $\alpha = \max \sigma(I_{m_0}^{k_0})$ then using relation (5) we know that $\boldsymbol{a}_\alpha^2 = x_{2\sigma^{-1}(\alpha)} \in V_{m_0}^{k_0}$. Using definition of reasoning sequence and relation (4) we know that

$$
x_{2\sigma^{-1}(\alpha+1)-1} = \boldsymbol{a}_{\alpha+1}^1 = \boldsymbol{a}_\alpha^2 = x_{2\sigma^{-1}(\alpha)}.
\tag{14}
$$

Note that $\sigma(\lfloor \frac{2\sigma^{-1}(\alpha+1)-1+1}{2} \rfloor) = \alpha + 1$. And then the induction hypotheses implies that there is a node $N_{m_1}^{k_0}$ such that

$$
x_{2\sigma^{-1}(\alpha+1)-1} \in V_{m_1}^{k_0},
\tag{15}
$$

$$
\max \sigma(I_{m_1}^{k_0}) - (\alpha + 1) \geqslant 2^{k_0-1} - 1,
\tag{16}
$$

$$
C_{m_1}^{k_0} \geqslant 2^{k_0-1} + 1.
\tag{17}
$$

This construction ensures that $x_{2\sigma^{-1}(\alpha+1)-1} = x_{2\sigma^{-1}(\alpha)} \in V_{m_1}^{k_0} \cap V_{m_0}^{k_0}$, then by Rule 3, we know that the node $N_{\max\{m_0,m_1\}}^{k_0+1}$ have value set $V_{\max\{m_0,m_1\}}^{k_0+1}$ and index set $I_{\max\{m_0,m_1\}}^{k_0+1}$ satisfying that

$$
x_j \in V_{\max\{m_0,m_1\}}^{k_0+1},
\tag{18}
$$

$$
I_{m_0}^{k_0} \cup I_{m_1}^{k_0} \subseteq I_{\max\{m_0,m_1\}}^{k_0+1}.
\tag{19}
$$

Combining (12), (16) and (19) leads to

$$
\max \sigma(I_{\max\{m_0,m_1\}}^{k_0+1}) - \sigma(\frac{j}{2}) \geqslant 2^{k_0} - 1,
$$

which implies that $C^{k_0+1}_{\max\{m_0,m_1\}} \geqslant 2^{k_0+1-1} + 1$. This completes the proof for lower bound.

Next we prove the upper bound.

It is clear that if Rule 1 is removed, the information quantity in each node can only maintain unchanged or increase. Therefore, we consider the no-mask condition, without loss of generality, we consider the reasoning sequence to be $\{x_i = \lfloor \frac{i}{2} \rfloor\}_{i \in \mathbb{N}}$ with constructing permutation $\sigma = \mathrm{Id}$ and constructing reasoning chain $(\boldsymbol{a}_m) = ((m-1, m))$.

We will use mathematical induction to prove that for $k \geqslant 2$ and $m \in \mathbb{Z}$, the value set of node $N^k_{2m}$ satisfies $V^k_{2m} \subseteq \{m - \frac{3^{k-1}+1}{2}, m - \frac{3^{k-1}+1}{2} + 1, \cdots, m + \frac{3^{k-1}-1}{2}\}$ and $C^k_{2m} \leqslant 3^{k-1} + 1$.

Since the index set of a node will play no rule in this proof, we will just ignore them in the expression of a node.

When $k = 2$, By Rule 2 the nodes $N^1_i$ are of the form $N^1_{2i} = \{\{x_{2i-1}, x_{2i}\}, \{2i\}\}$. Then by Rule 3 the nodes in layer 3 are of the form $N^2_{2i} = \{\{i - 2, i - 1, i, i + 1\}\{2i\}\}$. Therefore, the conclusion holds for $k = 2$.

Assuming the conclusion holds for $k \leqslant k_0$. When $k = k_0 + 1$, by inductive hypotheses, there are three nodes $N^{k_0}_{2m}, N^{k_0}_{2m_1}, N^{k_0}_{2m_2}$ with value sets $V^{k_0}_{2m}, V^{k_0}_{2m_1}, V^{k_0}_{2m_2}$. We require that

$$
\begin{aligned}
m + \frac{3^{k_0-1} - 1}{2} &= m_1 - \frac{3^{k_0-1} + 1}{2}, \\
m_2 + \frac{3^{k_0-1} - 1}{2} &= m - \frac{3^{k_0-1} + 1}{2}.
\end{aligned}
\tag{20}
$$

Simple calculation shows that

$$
m_1 = m + 3^{k_0-1}, \; m_2 = m - 3^{k_0-1}.
\tag{21}
$$

And the three nodes $N^{k_0}_{2m}, N^{k_0}_{2(m+3^{k_0-1})}, N^{k_0}_{2(m-3^{k_0-1})}$ have value sets

$$
V^{k_0}_{2m} \subseteq \{m - \frac{3^{k_0-1} + 1}{2}, m - \frac{3^{k_0-1} + 1}{2} + 1, \cdots, m + \frac{3^{k_0-1} - 1}{2}\},
\tag{22}
$$

$$
V^{k_0}_{2(m+3^{k_0-1})} \subseteq \{m + \frac{3^{k_0-1} - 1}{2}, m + \frac{3^{k_0-1} - 1}{2} + 1, \cdots, m + \frac{3^{k_0-1} - 1}{2}\},
\tag{23}
$$

$$
V^{k_0}_{2(m-3^{k_0-1})} \subseteq \{m - \frac{3^{k_0-1} + 1}{2}, m - \frac{3^{k_0-1} + 1}{2} + 1, \cdots, m - \frac{3^{k_0-1} + 1}{2}\}.
\tag{24}
$$

Again by Rule 3, we know that the node $N^{k_0+1}_{2m} = N^{k_0}_{2(m+3^{k-2})} \star N^{k_0}_{2(m-3^{k-2})} \star N^{k_0}_{2m}$ have value set $V^{k_0+1}_{2m} \subseteq \{m - \frac{3^{k_0}+1}{2}, m - \frac{3^{k_0}+1}{2} + 1, \cdots, m + \frac{3^{k_0}-1}{2}\}$, and therefore, $C^{k_0+1}_{2m} \leqslant 3^{k_0} + 1$. This completes the proof.

## C PROOF OF THEOREM 6.2

We shall also use the mathematical induction to prove this theorem.

We first assert that for $1 \leqslant l \leqslant 1 + \log_2 s$ and for $j$ satisfying $1 \leqslant \sigma(\lfloor \frac{j+1}{2} \rfloor) \leqslant s - 2^{l-1} + 1$ there exists a node $N^l_k$ such that

$$
\bigcup_{0 \leqslant \alpha \leqslant 2^{l-1} - 1} \{x_{2\sigma^{-1}(\sigma(\lfloor \frac{j+1}{2} \rfloor)+\alpha)-1}, x_{2\sigma^{-1}(\sigma(\lfloor \frac{j+1}{2} \rfloor)+\alpha)}\} \subseteq V^l_k.
\tag{25}
$$

For $l = 1$, the relation (25) can be verified easily since after position matching the nodes in $\mathcal{N}^1$ are of the form $N^1_{2m} = \{\{x_{2m-1}, x_{2m}\}, \{2m\}, \{2m - 1, 2m\}\}$. And note that $x_j \in \{x_{2\sigma^{-1}(\sigma(\lfloor \frac{j+1}{2} \rfloor))-1}, x_{2\sigma^{-1}(\sigma(\lfloor \frac{j+1}{2} \rfloor))}\} \subseteq V^1_{2\lfloor \frac{j+1}{2} \rfloor}$.

Now we assume that (25) holds for $l = l_0$, then we prove it still holds for $l = l_0 + 1$. Given $x_j \in (x_i)$ with $1 \leqslant \sigma(\lfloor \frac{j+1}{2} \rfloor) \leqslant s - 2^{l_0} + 1$, it is clear that $1 \leqslant \sigma(\lfloor \frac{j+1}{2} \rfloor) \leqslant s - 2^{l_0-1} + 1$. Then by inductive

hypotheses we know that there exists a node $N_{k_1}^{l_0}$ such that

$$\bigcup_{0 \leqslant \alpha \leqslant 2^{l_0-1}-1} \left\{ x_{2\sigma^{-1}\left(\sigma(\lfloor \frac{j+1}{2} \rfloor)+\alpha\right)-1}, x_{2\sigma^{-1}\left(\sigma(\lfloor \frac{j+1}{2} \rfloor)+\alpha\right)} \right\} \subseteq V_{k_1}^{l_0}. \tag{26}$$

On the other hand, since $1 \leqslant \sigma(\lfloor \frac{j+1}{2} \rfloor) \leqslant s-2^{l_0}+1$ we have $1 \leqslant \sigma(\lfloor \frac{j+1}{2} \rfloor)+2^{l_0-1} \leqslant s-2^{l_0-1}+1$, and again by inductive hypotheses there exists a node $N_{k_2}^{l_0}$ such that

$$\bigcup_{0 \leqslant \alpha \leqslant 2^{l_0-1}-1} \left\{ x_{2\sigma^{-1}\left(\sigma(\lfloor \frac{j+1}{2} \rfloor)+2^{l_0-1}+\alpha\right)-1}, x_{2\sigma^{-1}\left(\sigma(\lfloor \frac{j+1}{2} \rfloor)+2^{l_0-1}+\alpha\right)} \right\} \subseteq V_{k_2}^{l_0}. \tag{27}$$

Note that by the definition of reasoning sequence we have

$$x_{2\sigma^{-1}\left(\sigma(\lfloor \frac{j+1}{2} \rfloor)+2^{l_0-1}\right)-1} = x_{2\sigma^{-1}\left(\sigma(\lfloor \frac{j+1}{2}+\rfloor)+2^{l_0-1}-1\right)}.$$

By Rule 3, there exists a node $N_{\max\{k_1,k_2\}}^{l_0+1}$ such that

$$\bigcup_{0 \leqslant \alpha \leqslant 2^{l_0}-1} \left\{ x_{2\sigma^{-1}\left(\sigma(\lfloor \frac{j+1}{2} \rfloor)+\alpha\right)-1}, x_{2\sigma^{-1}\left(\sigma(\lfloor \frac{j+1}{2} \rfloor)+\alpha\right)} \right\} \subseteq V_{k_1}^{l_0} \cup V_{k_2}^{l_0} \subseteq V_{\max\{k_1,k_2\}}^{l_0+1}.$$

This completes the proof of our assertion.

Set the reasoning start as $x_{2s+1} = x_j$, we then assert that

$$\bigcup_{0 \leqslant \alpha \leqslant 2^{l-1}-2} \left\{ x_{2\sigma^{-1}\left(\sigma(\lfloor \frac{j+1}{2} \rfloor)+\alpha\right)-1}, x_{2\sigma^{-1}\left(\sigma(\lfloor \frac{j+1}{2} \rfloor)+\alpha\right)} \right\} \subseteq V_{2s+1}^{l}. \tag{28}$$

When $l = 2$ the assertion (28) can be easily verified. We assume (28) holds for $l = l_0$ and consider the case $l = l_0 + 1$. By assertion (25), we know there exists $N_k^{l_0}$ such that

$$\bigcup_{0 \leqslant \alpha \leqslant 2^{l_0-1}-1} \left\{ x_{2\sigma^{-1}\left(\sigma(\lfloor \frac{j+1}{2} \rfloor)+2^{l_0-1}-1+\alpha\right)-1}, x_{2\sigma^{-1}\left(\sigma(\lfloor \frac{j+1}{2} \rfloor)+2^{l_0-1}-1+\alpha\right)} \right\} \subseteq V_k^{l_0}. \tag{29}$$

Again, by definition of reasoning sequence we have

$$x_{2\sigma^{-1}\left(\sigma(\lfloor \frac{j+1}{2} \rfloor)+2^{l_0-1}-1\right)-1} = x_{2\sigma^{-1}\left(\sigma(\lfloor \frac{j+1}{2}+\rfloor)+2^{l_0-1}-2\right)}.$$

And by Rule 3 we know that

$$\bigcup_{0 \leqslant \alpha \leqslant 2^{l_0}-2} \left\{ x_{2\sigma^{-1}\left(\sigma(\lfloor \frac{j+1}{2} \rfloor)+\alpha\right)-1}, x_{2\sigma^{-1}\left(\sigma(\lfloor \frac{j+1}{2} \rfloor)+\alpha\right)} \right\} \subseteq V_k^{l_0} \cup V_{2s+1}^{l_0} \subseteq V_{2s+1}^{l_0+1}.$$

This completes the proof of assertion (28), which implies that for $l \geqslant 2$, $C_{2s+1}^l \geqslant 2^{l-1}$.

For $l = 1$, it is clear that $C_{2s+1}^2 = 1$ since only residual connection happens. And we complete the proof for the lower bound.

The upper bound is proved similarly as in the proof of Theorem 6.1, except we consider mainly the reasoning start position, which results in one fewer element.

# D EXAMPLES WHERE THE THEORETICAL BOUNDS ARE ACHIEVED

In this section we give some examples related to Theorem 6.2. In fact, both the lower bound and upper bound can be attained.

## LOWER BOUND

We construct a reasoning sequence $(1, 2, 2, 3, 3, 4, 4, 5 \cdots, 2s - 1, 2s, 1)$. Assume that $1 \leqslant l \leqslant 1 + \log_2 s$, it is clear that $C_{2s+1}^l \geqslant 2^{l-1}$

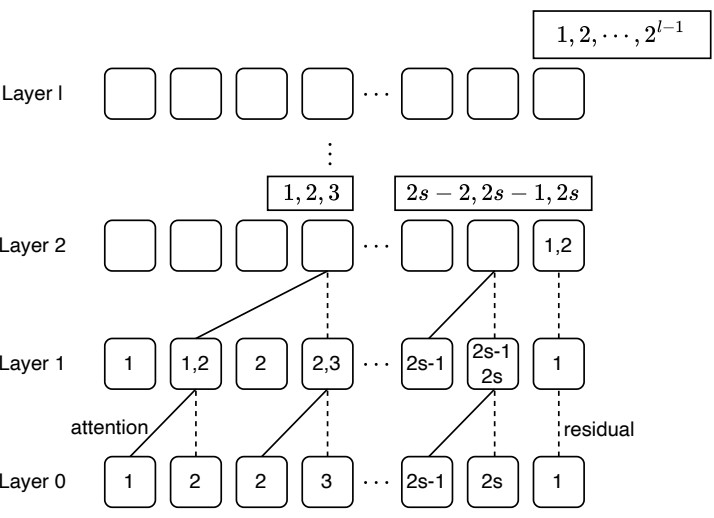

Figure 8: Lower bound example

## UPPER BOUND

In this section, we shall use the concept of truncation of a reasoning sequence. This truncated reasoning sequence contains a finite step reasoning chain and some irrelevant reasoning pairs which serves as some redundant information as in practice. In fact, we truncate a reasoning sequence $(x_i)$ as follows:

1. Firstly, we truncate the constructing reasoning sequence $(\boldsymbol{a}_m)$ to be an $s$ step reasoning chain $(\tilde{\boldsymbol{a}}_k) = (\boldsymbol{a}_i, \boldsymbol{a}_{i+1}, \cdots, \boldsymbol{a}_{i+s-1})$.

2. Secondly, we use the relation (5) to find those elements constructed from this $s$ step reasoning chain. That is

$$E = \{x_{2(\sigma^{-1}(i)-1)}, x_{2(\sigma^{-1}(i)-1)+1}, \cdots, x_{2(\sigma^{-1}(i+s-1)-1)}, x_{2(\sigma^{-1}(i+s-1)-1)+1}\}. \quad (30)$$

3. Thirdly, we set

$$\begin{aligned} \mathcal{I}_E = \{ & 2(\sigma^{-1}(i) - 1), 2(\sigma^{-1}(i) - 1) + 1, \cdots, \\ & 2(\sigma^{-1}(i + s - 1) - 1), 2(\sigma^{-1}(i + s - 1) - 1) + 1\}, \end{aligned} \quad (31)$$

which is the set of all subscripts of elements in $E$. Moreover, we take the subsequence $(\tilde{x}_i) = (x_{\min\{\mathcal{I}_E\}+i-1})$ as the truncated reasoning sequence, where $1 \leqslant i \leqslant \max\{\mathcal{I}_E\} - \min\{\mathcal{I}_E\} + 1$.

We call this sequence $(\tilde{x}_i)$ the truncated reasoning sequence containing $(\tilde{\boldsymbol{a}}_k)$.

With this definition of truncation, we provide a method to construct finite step reasoning sequences which allows the upper bound in Theorem 6.2 to be attained. This is achieved in a recursive way.

Consider a reasoning sequence $(\boldsymbol{a}_i)_{i \in \mathbb{Z}}$. For a given integer $i$ we define

$$\boldsymbol{s}_1(i) = (\boldsymbol{a}_i), \quad \boldsymbol{s}_2(i) = (\boldsymbol{a}_i, \boldsymbol{a}_{i+2}, \boldsymbol{a}_{i+1}),$$

and

$$\boldsymbol{s}_3(i) = (\boldsymbol{a}_i, \boldsymbol{a}_{i+2}, \boldsymbol{a}_{i+1}, \boldsymbol{a}_{i+6}, \boldsymbol{a}_{i+8}, \boldsymbol{a}_{i+7}, \boldsymbol{a}_{i+3}, \boldsymbol{a}_{i+5}, \boldsymbol{a}_{i+4}).$$

For simplicity, we use the notation

$$s_3(i) = (s_2(i), s_2(i+6), s_2(i+3))$$
$$:= (a_i, a_{i+2}, a_{i+1}, a_{i+6}, a_{i+8}, a_{i+7}, a_{i+3}, a_{i+5}, a_{i+4}),$$

and with this notation $s_3(i)$ is still a sequence of reasoning pairs.

For $k \geqslant 3$ we define

$$s_k(i) = (s_{k-1}(i), s_{k-1}(i+2\times 3^{k-2}), s_{k-1}(i+3^{k-2})).$$

Same notation as in $s_3(i)$ and all $s_k(i)$ are sequences of reasoning pairs.

We now construct our reasoning sequence.

Set $r_1 = (s_{l-1}(i - \frac{3^{l-1}-1}{2}), \cdots, s_3(i - 13), s_2(i - 4), s_1(i - 1), s_1(i), s_2(i + 1), s_3(i + 4), \cdots, s_{l-1}(i + \frac{3^{l-2}-1}{2}))$, it is clear that there exist $\sigma \in \mathrm{Sym}(\mathbb{Z})$ such that $(a_{\sigma(1-\frac{3^{l-1}}{2})}, \cdots, a_{\sigma(1)}, a_{\sigma(2)}, \cdots, a_{\sigma(\frac{3^{l-1}-1}{2})}) = r_1$. By the construction of $r_1$ we know all the reasoning pairs in $r_1$ forms a finite step reasoning chain $(a_{i-\frac{3^{l-1}-1}{2}}, \cdots a_i, a_{i+1}, \cdots, a_{i+\frac{3^{l-1}-3}{2}})$, we extend this finite step reasoning chain to an infinite reasoning chain $\tilde{a}$ by adding reasoning pairs to both sides. And we take a permutation $\tau \in \mathrm{Sym}(\mathbb{Z})$ satisfying $a_{\tau(i_m)} = a_{\sigma(m)}$ where $i_m \in I$ with $I = \{i_{1-\frac{3^{l-1}}{2}}, \cdots, i_1, i_2, \cdots, i_{\frac{3^{l-1}-3}{2}}\}$ and $i_{1-\frac{3^{l-1}}{2}} < \cdots < i_1 < i_2 < \cdots < i_{\frac{3^{l-1}-3}{2}}$.

We truncate the reasoning sequence $((x_j), \tilde{a}, \tau)$ to contain the finite reasoning chain $(a_{i-\frac{3^{l-1}-1}{2}}, \cdots, a_{i+\frac{3^{l-1}-3}{2}})$ to get a sequence $(\tilde{x}_j) = (x_{\min\{\mathcal{I}_E\}+j-1})_{1\leqslant j\leqslant \max\{\mathcal{I}_E\}-\min\{\mathcal{I}_E\}+1}$. Here $E$ and $\mathcal{I}_E$ are defined as in (30) and (31). This $(\tilde{x}_j)$ with reasoning start $x_{\max \mathcal{I}_E+1} = a_i^1$ is the sequence we want.

**Remark D.1.** *Under the rules of information propagation, the elements in $(\tilde{x}_j)$ contributing to the node $N^l_{\max \mathcal{I}_E+1}$ are those come from the reasoning chain $(a_{i-\frac{3^{l-1}-1}{2}}, \cdots, a_{i+\frac{3^{l-1}-3}{2}})$. Hence, the information in $N^l_{\max \mathcal{I}_E+1}$ will be the same if we consider the reasoning sequence constructed from $(a_{i-\frac{3^{l-1}-1}{2}}, \cdots, a_{i+\frac{3^{l-1}-3}{2}})$ and $\sigma$. However, the above complicated way we construct truncated sequence is still necessary which shows that there is a large class of reasoning sequence allows the upper bound to be attained.*

**Proposition D.2.** *We consider the reasoning sequence $(x_i)$ with constructing reasoning chain $(a_m) = ((m, m+1))_{1-\frac{3^{\tilde{l}-1}-1}{2}\leqslant m\leqslant \frac{3^{\tilde{l}-1}-1}{2}}$ and constructing permutation $\sigma$ satisfying that*

$$(a_{\sigma(1)}, a_{\sigma(2)}, \cdots, a_{\sigma(\frac{3^{\tilde{l}-1}-1}{2})}) = (s_1(1), s_2(2), s_3(5), \cdots, s_l(\frac{3^{l-1}+1}{2}), \cdots s_{\tilde{l}-1}(\frac{3^{\tilde{l}-2}+1}{2})), \tag{32}$$

$$(a_{\sigma(1-\frac{3^{\tilde{l}-1}}{2})}, a_{\sigma(2-\frac{3^{\tilde{l}-1}}{2})}, \cdots, a_{\sigma(0)}) =$$
$$(s_{\tilde{l}-1}(\frac{3-3^{\tilde{l}-1}}{2}), \cdots, s_l(\frac{3-3^l}{2}), \cdots s_3(-12), s_2(-3), s_1(0),). \tag{33}$$

*If the reasoning start is set be to $x_{2\sigma(\frac{3^{\tilde{l}-1}-1}{2})+1} = a_1^1$, then for $1 \leqslant l \leqslant \tilde{l}$, we have $C^l_{2\sigma(\frac{3^{\tilde{l}-1}-1}{2})+1} = 3^{l-1}$.*

To prove this proposition we need the following two lemmas.

**Lemma D.3.** *For $m \geqslant 1$, $\forall k \in \{\frac{3^{j-2}+1}{2}\}_{2\leqslant j\leqslant \tilde{l}}$, there exists $i_{(m,k)} \in [1, 3^{\tilde{l}-1}-1] \cap \mathbb{Z}$ depending on $m$ and $k$ such that the node $N^m_{i_{(m,k)}}$ have value set $V^m_{i_{(m,k)}}$ satisfying the following property.*

$$\{k, k+1, \cdots, k+3^{m-1}\} \subseteq V^m_{i_{(m,k)}}. \tag{34}$$

**Lemma D.4.** *For $m \geqslant 1$, $\forall k \in \{\frac{3-3^{j-1}}{2}\}_{2\leqslant j\leqslant \tilde{l}}$, there exist $i_{(m,k)} \in [1 - 3^{\tilde{l}-1}, 0] \cap \mathbb{Z}$ such that the node $N^m_{i_{(m,k)}}$ have value set $V^m_{i_{(m,k)}}$ satisfying the following property.*

$$\{k, k+1, \cdots, k+3^{m-1}\} \subseteq V^m_{i_{(m,k)}}. \tag{35}$$

**Proof of Proposition D.2** . The case $l = 1$ is trivial and omitted. We use mathematical induction to prove this proposition. We assert that for $l \geqslant 2$ the reasoning start node $N^l_{2\sigma(\frac{3^{\tilde{l}-1}-1}{2})+1}$ have value set $V^l_{2\sigma(\frac{3^{\tilde{l}-1}-1}{2})+1}$ satisfying

$$\{\frac{3-3^{l-1}}{2}, \frac{3-3^{l-1}}{2}+1, \cdots, 0, 1, 2, \cdots, \frac{3^{l-1}+1}{2}\} \subseteq V^l_{2\sigma(\frac{3^{\tilde{l}-1}-1}{2})+1}. \tag{36}$$

The case $l = 2$ can be verified easily. We assume that for $l = l_0 \in [2, \tilde{l}-1] \cap \mathbb{Z}$ the node $N^{l_0}_{2\sigma(\frac{3^{\tilde{l}-1}-1}{2})+1}$ have value set $V^{l_0}_{2\sigma(\frac{3^{\tilde{l}-1}-1}{2})+1}$ satisfying

$$\{\frac{3-3^{l_0-1}}{2}, \frac{3-3^{l_0-1}}{2}+1, \cdots, 0, 1, 2, \cdots, \frac{3^{l_0-1}+1}{2}\} \subseteq V^l_{2\sigma(\frac{3^{\tilde{l}-1}-1}{2})+1}. \tag{37}$$

Now for $l = l_0 + 1$, by Lemma D.3, we know there exists a node $N^{l_0}_{i(l_0, \frac{3^{l_0-1}+1}{2})}$ such that

$$\{\frac{3^{l_0-1}+1}{2}, \cdots, \frac{3^{l_0-1}+1}{2}+3^{l_0-1}\} \subseteq V^{l_0}_{i(l_0, \frac{3^{l_0-1}+1}{2})}.$$

And by Lemma D.4 there exist a node $N^{l_0}_{i(l_0, \frac{3-3^{l_0}}{2})}$ such that

$$\{\frac{3-3^{l_0}}{2}, \cdots, \frac{3-3^{l_0}}{2}+3^{l_0-1}\} \subseteq V^{l_0}_{i(l_0, \frac{3-3^{l_0}}{2})}.$$

We then have

$$
\begin{aligned}
V^{l_0}_{i(l_0, \frac{3^{l_0-1}+1}{2})} \cap V^{l_0}_{2\sigma(\frac{3^{\tilde{l}-1}-1}{2})+1} &= \{\frac{3^{l_0-1}+1}{2}\}, \\
V^{l_0}_{i(l_0, \frac{3-3^{l_0}}{2})} \cap V^{l_0}_{2\sigma(\frac{3^{\tilde{l}-1}-1}{2})+1} &= \{\frac{3-3^{l_0-1}}{2}\},
\end{aligned} \tag{38}
$$

and by Rule 3, we know that the node $N^{l_0+1}_{2\sigma(\frac{3^{\tilde{l}-1}-1}{2})+1}$ have value set $V^{l_0+1}_{2\sigma(\frac{3^{\tilde{l}-1}-1}{2})+1}$ satisfying

$$
\begin{aligned}
\{\frac{3-3^{l_0}}{2} \cdots, 0, 1, 2, \cdots, \frac{3^{l_0}+1}{2}\} &= V^{l_0}_{i(l_0, \frac{3^{l_0-2}+1}{2})} \cup V^{l_0}_{2\sigma(\frac{3^{\tilde{l}-1}-1}{2})+1} \cup V^{l_0}_{i(l_0, \frac{3-3^{l_0-1}}{2})} \\
&\subseteq V^{l_0+1}_{2\sigma(\frac{3^{\tilde{l}-1}-1}{2})+1}.
\end{aligned} \tag{39}
$$

This completes the proof of our assertion. From above assertion we know that $C^l_{2\sigma(\frac{3^{\tilde{l}-1}-1}{2})+1} \geqslant 3^{l-1}$. Combining the proof of Theorem 6.2, we know that $C^l_{2\sigma(\frac{3^{\tilde{l}-1}-1}{2})+1} \leqslant 3^{l-1}$. And therefore $C^l_{2\sigma(\frac{3^{\tilde{l}-1}-1}{2})+1} = 3^{l-1}$. $\qquad\square$

**Proof of Lemma D.3** . We use mathematical induction to prove this lemma. The case $m = 2$ can be verified easily through Rule 2. We assume that for $m = m_0$, $\forall k \in \{\frac{3^{l-2}+1}{2}\}_{2 \leqslant l \leqslant \tilde{l}}$,

$$\{k, k+1, \cdots, k+3^{m_0-1}\} \subseteq V^{m_0}_{i(m_0, k)}. \tag{40}$$

For $m = m_0 + 1$, by assumption, there exists $i(m_0, k)$, $i(m_0, k+3^{m_0-1})$, $i(m_0, k+2 \times 3^{m_0-1})$ such that

$$
\begin{aligned}
\{k, \cdots, k+3^{m_0-1}\} &\subseteq V^{m_0}_{i(m_0, k)}, \\
\{k+3^{m_0-1}, \cdots, k+2 \times 3^{m_0-1}\} &\subseteq V^{m_0}_{i(m_0, k+3^{m_0-1})}, \\
\{k+2 \times 3^{m_0-1}, \cdots, k+3^{m_0}\} &\subseteq V^{m_0}_{i(m_0, k+2 \times 3^{m_0-1})}.
\end{aligned} \tag{41}
$$

Set $i(m_0 + 1, k) = \max\{i(m_0, k), i(m_0, k + 3^{m_0-1}), i(m_0, k + 2 \times 3^{m_0-1})\}$, by Rule 3 and (41) we know that

$$\{k, \cdots, k + 3^{m_0}\} \subseteq V_{i(m_0,k)}^{m_0} \cup V_{i(m_0,k+3^{m_0-1})}^{m_0} \cup V_{i(m_0,k+2\times 3^{m_0-1})}^{m_0} \subseteq V_{i(m_0+1,k)}^{m_0+1}, \qquad (42)$$

which completes the proof. □

The proof of Lemma D.4 is analogous to that of Lemma D.3 and is omitted.

# E    CONSTRUCTION OF TRANSFORMER

## E.1    EMBEDDING

We assume that $d_m$ is large enough. We choose a suitable embedding (which can be done by choosing a suitable basis of $\mathbb{R}^{d_m}$) such that the non-zero elements of all value vector $\boldsymbol{X}^{pos}$ are located in the first $n$ coordinates and all the elements in $\boldsymbol{X}^{tgt}$ located at the first $n$ coordinates are zero.

We denote $\boldsymbol{X}^{tgt} = (\boldsymbol{X}_1^{tgt,\mathsf{T}}, \boldsymbol{X}_2^{tgt,\mathsf{T}}, \cdots, \boldsymbol{X}_n^{tgt,\mathsf{T}})^{\mathsf{T}}$ with each $\boldsymbol{X}_i^{tgt} = (\boldsymbol{X}_{i,1}^{tgt}, \boldsymbol{X}_{i,2}^{tgt}, \cdots, \boldsymbol{X}_{i,d_m}^{tgt}) \in \mathbb{R}^{d_m}$. Similarly, we denote $\boldsymbol{X}^{pos} = (\boldsymbol{X}_1^{pos,\mathsf{T}}, \boldsymbol{X}_2^{pos,\mathsf{T}}, \cdots, \boldsymbol{X}_n^{pos,\mathsf{T}})^{\mathsf{T}}$ and $\boldsymbol{X}^{(l)} = (\boldsymbol{X}_1^{(l),\mathsf{T}}, \boldsymbol{X}_2^{(l),\mathsf{T}}, \cdots, \boldsymbol{X}_n^{(l),\mathsf{T}})^{\mathsf{T}}$.

We correspond each element in the set $\tilde{E}$ to an index i.e., the set $\tilde{E}$ is of the form $\tilde{E} = \{\boldsymbol{v}_1, \boldsymbol{v}_2, \cdots, \boldsymbol{v}_d\}$. Moreover, each $\boldsymbol{v}_i$ is of the form $(\underbrace{0, 0, \cdots, 0}_{n \text{ zeros}}, 0, \cdots, 1, \cdots, 0)$, and we denote $k_i$ the position of 1. Let $n \leqslant k_1 < k_2 < \cdots < k_d$, we require the embedding is chosen such that $k_i$ satisfies that

$$\begin{cases} k_1 - n \geqslant 2(n+1)(3^L + 1), \\ k_i - k_{i-1} \geqslant 2(n+1)(3^L + 1), & \text{for } 2 \leqslant i \leqslant \dfrac{n-1}{2}, \\ d_m - k_{\frac{n-1}{2}} \geqslant 2(n+1)(3^L + 1). \end{cases} \qquad (43)$$

## E.2    CONSTRUCTION OF PARAMETERS

We set the weight matrices as follows:

$$\boldsymbol{W}^{q(l)} = \boldsymbol{I}(0 \leqslant l \leqslant L), \qquad (44)$$

$$\boldsymbol{W}^{vo(l)} = \boldsymbol{I}(1 \leqslant l \leqslant L), \ \boldsymbol{W}^{vo(0)} = \boldsymbol{R}, \qquad (45)$$

$$\boldsymbol{W}^{k(l),\mathsf{T}} = \sum_{i=-(n+1)3^L}^{-1} \boldsymbol{R}^i (l \geqslant 1), \ \boldsymbol{W}^{k(0)} = \sum_{i=1}^{\frac{n-1}{2}} \boldsymbol{p}_{2i-1} \boldsymbol{p}_{2i}^T. \qquad (46)$$

Here $\boldsymbol{p}_j$ is the positional encoding and arbitrary two positional encodings are orthogonal to each other. Moreover, $\boldsymbol{R} \in \mathbb{R}^{d_m \times d_m}$ is defined as $\boldsymbol{R} = [\boldsymbol{R}_{ij}]$, with $\boldsymbol{R}_{i+1,i} = 1 = \boldsymbol{R}_{1d_m}$ for $1 \leqslant i \leqslant d_m - 1$, and all the other elements of $\boldsymbol{R}$ are set to be 0. In fact,

$$\boldsymbol{R} = \begin{bmatrix} 0 & 0 & 0 & \cdots & 1 \\ 1 & 0 & 0 & \cdots & 0 \\ 0 & 1 & 0 & \cdots & 0 \\ \vdots & \vdots & \ddots & \ddots & \vdots \\ 0 & 0 & \cdots & 1 & 0 \end{bmatrix}.$$

This matrix $\boldsymbol{R}$ is called the left shift matrix. When we apply $\boldsymbol{R}$ to a vector $\boldsymbol{v} = (\boldsymbol{v}_1, \boldsymbol{v}_2, \cdots, \boldsymbol{v}_{d_m})$ then result is $\boldsymbol{v}\boldsymbol{R} = (\boldsymbol{v}_2, \cdots, \boldsymbol{v}_{d_m-1}, \boldsymbol{v}_{d_m}, \boldsymbol{v}_1)$. The notation $\boldsymbol{R}^i$ means that the matrix $\boldsymbol{R}$ act $i$ times, i.e., $\boldsymbol{v}\boldsymbol{R}^i = \boldsymbol{v}\boldsymbol{R}\boldsymbol{R}\cdots\boldsymbol{R}$. Also, the inverse of $mR$ is called the right shift matrix, and $\boldsymbol{R}^{-1} = \boldsymbol{R}^{\mathsf{T}}$.

For any given matrix $\boldsymbol{M} = [m_{ij}]$, we define the mask operation as

$$
\text{mask}(\boldsymbol{M}) = \begin{bmatrix} m_{1,1}^{(l)} & -\infty & -\infty & \cdots & -\infty \\ m_{2,1}^{(l)} & m_{2,2}^{(l)} & -\infty & \cdots & -\infty \\ m_{3,1}^{(l)} & m_{3,2}^{(l)} & m_{3,3}^{(l)} & \cdots & -\infty \\ \vdots & \vdots & \vdots & \ddots & \vdots \\ m_{n,1}^{(l)} & m_{n,2}^{(l)} & \cdots & m_{n,n-1}^{(l)} & m_{n,n}^{(l)} \end{bmatrix}.
\tag{47}
$$

Furthermore, we denote

$$
\boldsymbol{A}^{(l)} = \text{mask}(\boldsymbol{X}^{(l)}\boldsymbol{W}^{q(l)}\boldsymbol{W}^{k(l),\mathsf{T}}\boldsymbol{X}^{(l),\mathsf{T}}) = \begin{bmatrix} \boldsymbol{A}_{1,1}^{(l)} & -\infty & -\infty & \cdots & -\infty \\ \boldsymbol{A}_{2,1}^{(l)} & \boldsymbol{A}_{2,2}^{(l)} & -\infty & \cdots & -\infty \\ \boldsymbol{A}_{3,1}^{(l)} & \boldsymbol{A}_{3,2}^{(l)} & \boldsymbol{A}_{3,3}^{(l)} & \cdots & -\infty \\ \vdots & \vdots & \vdots & \ddots & \vdots \\ \boldsymbol{A}_{n,1}^{(l)} & \boldsymbol{A}_{n,2}^{(l)} & \cdots & \boldsymbol{A}_{n,n-1}^{(l)} & \boldsymbol{A}_{n,n}^{(l)} \end{bmatrix}.
\tag{48}
$$

We shall still use the notation $N_i^l$ to denote the $i$th node in $l$th layer with slightly modification: the value set $\boldsymbol{V}_i^l$ is now a set $\boldsymbol{V}_i^l$ which contains the embedded value as element.

Given any reasoning chain $(\boldsymbol{a}_u)_{1 \leqslant i \leqslant n}$, after embedding it is of the form

$$
(\tilde{\boldsymbol{a}}_u) = ((\tilde{\boldsymbol{a}}_u^1, \tilde{\boldsymbol{a}}_u^2))_{1 \leqslant i \leqslant n} = ((\boldsymbol{a}_u^1\boldsymbol{W}^{\text{emb}}, \boldsymbol{a}_u^2\boldsymbol{W}^{\text{emb}}))_{1 \leqslant i \leqslant n}.
$$

We define a sequence $(\tilde{\boldsymbol{b}}_v)_{1 \leqslant v \leqslant n+1}$ as follows:

$$
\begin{aligned}
\tilde{\boldsymbol{b}}_v &= \tilde{\boldsymbol{a}}_v^1, \ \text{ for } 1 \leqslant v \leqslant n, \\
\tilde{\boldsymbol{b}}_v &= \tilde{\boldsymbol{a}}_n^2 \ \text{ for } v = n+1.
\end{aligned}
\tag{49}
$$

Assume the node $N_i^l$ contains the information from $(\tilde{a}_i)$, then its value set $\boldsymbol{V}_i^l$ must contain the elements from a subsequence of $\tilde{\boldsymbol{b}}_{v \in [v_1, v_2] \cap \mathbb{Z}}$ where $v_1, v_2$ depends on $i$ and $l$. For simplicity, we set $\boldsymbol{b}_{i,v}^l = \tilde{\boldsymbol{b}}_{v_1+v-1}$ for $v \in [1, v_2 - v_1 + 1] \cap \mathbb{Z}$. In fact, $v_2 - v_1 + 1 = C_i^l$.

With these notations, we define the FNN as follows

$$
\begin{cases} L_N^l(f_i^l(\boldsymbol{X}_i^{ao(l)}) + (\boldsymbol{X}_i^{ao(l)})) = \displaystyle\sum_{1 \leqslant k \leqslant C_i^l} \boldsymbol{b}_{i;k}\boldsymbol{R}^{i3^L - j + k} \text{ for } i \geqslant 2, \\ L_N^l(f_i^l(\boldsymbol{X}_i^{ao(l)}) + (\boldsymbol{X}_i^{ao(l)})) = \boldsymbol{X}_1^{tgt}\boldsymbol{R}^{(n+1)3^L} \text{ for } i = 1. \end{cases}
\tag{50}
$$

Here $L_N$ stands for LayerNorm and $j \in [1, n] \cap \mathbb{Z}$ satisfies $\boldsymbol{b}_{i,j}^l = X_i^{tgt}$.

**Remark E.1.** *Note that this is not an accurate expression, since we can not simply define the output of a FNN. However, we can show that there exist $f_i^{(l)}$ for each layer such that the output of $L_N(f_i^l(\boldsymbol{X}_i^{ao(l)}) + (\boldsymbol{X}_i^{ao(l)}))$ differs very little from the right-hand side of (50). And also the information propagation won't be affected by this error. This will be explained in detail in later sections.*

By this construction the reasoning information contained in the node $N_i^l$ is encoded by $L_N(f_i^l(\cdot) + (\cdot))$.

In order to extract the result after m-step reasoning, we set

$$
\boldsymbol{W}^p = \boldsymbol{R}^{-n3^L - m + 1}\boldsymbol{Q}\boldsymbol{W}^{emb,\mathsf{T}},
\tag{51}
$$

where $\boldsymbol{Q}$ satisfies $\boldsymbol{Q}\boldsymbol{v}_1 = \boldsymbol{v}_1, ..., \boldsymbol{Q}\boldsymbol{v}_d = \boldsymbol{v}_d$ and maps other basis to 0. Note that last token on the $s$-th layer is $\boldsymbol{X}_n^{(L)} = \boldsymbol{b}_{n;1}\boldsymbol{R}^{n3^s-j+1} + ...... + \boldsymbol{b}_{n;t}\boldsymbol{R}^{n3^s+t-j}$, and we have

$$\boldsymbol{X}_n^{(L)}\boldsymbol{W}^p = \boldsymbol{b}_{n;1}\boldsymbol{R}^{-j-m+2}\boldsymbol{Q}\boldsymbol{W}^{emb,\mathsf{T}} + ...... + \boldsymbol{b}_{n;t}\boldsymbol{R}^{t-j-m+1}\boldsymbol{Q}\boldsymbol{W}^{emb,\mathsf{T}}, \tag{52}$$

and the output

$$\boldsymbol{Y} = argmax(\tilde{\sigma}(\boldsymbol{X}_n^{(L)}\boldsymbol{W}^p)). \tag{53}$$

When $m + j - 1 \leqslant t$, $\boldsymbol{X}_n^{(L)}\boldsymbol{W}^p = \boldsymbol{b}_{n;m+j-1}\boldsymbol{W}^{emb,\mathsf{T}}$ and the above setting yields the desired reasoning result. However, when $m + j - 1 > t$, $\boldsymbol{X}_n^{(L)}\boldsymbol{W}^p = 0$ and therefore we can not get a right reasoning result.

In fact, the sufficiency of transformer depth $s$ relative to the required reasoning steps $m$ is a key factor in ensuring accurate reasoning result.

We can roughly categorize this relationship into three cases

- **Case 1** $m \leqslant 2^{L-1} - 1$;
- **Case 2** $2^{s-1} \leqslant m \leqslant \frac{3^{L-1}+1}{2}$;
- **Case 3** $m > \frac{3^{L-1}-1}{2}$.

For the first case, note that $t - j + 1 \geq 2^{L-1} - 1$ and therefore $m + j - 1 \leq t$ which means we can get the result. For the third case, since $t - j + 1 \leq \frac{3^{L-1}+1}{2}$ and consequently $m + j - 1 > t$, the model can not derive the result. The second case is more complicated, since we can not derive the relationship of $t - j + 1$ and $t$ from the relationship of $m$ and $s$. Whether the model can derive the reasoning result now depends on $t - j + 1$ and $t$.

### E.3 EXPLANATION

First, we explain how the same token matching rule works in this construction. More specifically, the attention matrices defined above satisfy the following property.

**Lemma E.2.** *For $l \geqslant 1$, we have*

$$\begin{aligned}
\boldsymbol{A}_{i,j}^{(l)} &= 0, \ if \ j = 1, \\
\boldsymbol{A}_{i,j}^{(l)} &= 0, \ if \ i \geqslant j \geqslant 2, \ \boldsymbol{V}_i^l \cap \boldsymbol{V}_j^l = \emptyset, \\
\boldsymbol{A}_{i,j}^{(l)} &\geqslant 1, \ if \ i \geqslant j \geqslant 2, \ \boldsymbol{V}_i^l \cap \boldsymbol{V}_j^l \neq \emptyset.
\end{aligned} \tag{54}$$

*Proof.* Start with given any $\boldsymbol{X}^{(0)} = (\boldsymbol{X}_1^{(0)}, \boldsymbol{X}_2^{(0)}, \cdots, \boldsymbol{X}_n^{(0)})^\mathsf{T}$, we denote $\boldsymbol{X}^{(l)} = (\boldsymbol{X}_1^{(l)}, \boldsymbol{X}_2^{(l)}, \cdots, \boldsymbol{X}_n^{(l)})$ with $\boldsymbol{X}_i^{(l)} = \sum_{u=1}^{C_i^l} \boldsymbol{b}_{i;u}^l \boldsymbol{R}^{i3^L+u-d_i^{(l)}}$, where $C_i^l = |\boldsymbol{V}_i^l|$ and $\boldsymbol{b}_{i;u}^l \in \tilde{E}$.

$$\boldsymbol{A}_{i,j}^{(l-1)} = \boldsymbol{X}_i^{(l-1)} \boldsymbol{R}^{q(l-1)} \boldsymbol{R}^{k(l-1),\mathsf{T}} \boldsymbol{X}_j^{(l-1),\mathsf{T}} \tag{55}$$

$$= (\sum_{u=1}^{C_i^{l-1}} \boldsymbol{b}_{i;u}^{l-1}\boldsymbol{R}^{i3^L+u-d_i^{(l-1)}})(\sum_{m=-(n+1)3^L}^{-1} \boldsymbol{R}^m)(\sum_{v=1}^{C_j^{l-1}} \boldsymbol{b}_{j;v}^{l-1}\boldsymbol{R}^{j3^L+v-d_j^{(l-1)}})^\mathsf{T} \tag{56}$$

$$= \sum_{u=1}^{C_i^{l-1}} \sum_{m=-(n+1)3^L}^{-1} \sum_{v=1}^{C_j^{l-1}} \boldsymbol{b}_{i;u}^{l-1}\boldsymbol{R}^{i3^L+u-d_i^{(l-1)}-j3^L-v+d_j^{(l-1)}+m}\boldsymbol{b}_{j;v}^{l-1,\mathsf{T}} \tag{57}$$

For any $i \geqslant j \geqslant 2$ and $\boldsymbol{V}_i^{l-1} \cap \boldsymbol{V}_j^{l-1} = \emptyset$.

Since $-(n+1)3^L \leqslant i3^L + u - d_i^{(l-1)} - j3^L - v + d_j^{(l-1)} + m \leqslant (n+1)3^L$ and $\boldsymbol{b}_{i;u}^l \neq \boldsymbol{b}_{j;v}^l$, we have $\boldsymbol{b}_{i;u}^{l-1}\boldsymbol{R}^{i3^L+u-d_i^{(l-1)}-j3^L-v+d_j^{(l-1)}+m}\boldsymbol{b}_{j;v}^{l-1,\mathsf{T}} = 0$. Therefore, $\boldsymbol{A}_{i,j}^{(l-1)} = 0$.

When $i \geqslant j \geqslant 2$ and $\boldsymbol{V}_i^{l-1} \cap \boldsymbol{V}_j^{l-1} \neq \emptyset$,

$$\boldsymbol{A}_{i,j}^{(l-1)} = \sum_{u=1}^{C_i^{l-1}} \sum_{m=-(n+1)3^L}^{-1} \sum_{v=1}^{C_j^{l-1}} \boldsymbol{b}_{i;u}^{l-1} \boldsymbol{R}^{i3^L+u-d_i^{(l-1)}-j3^L-v+d_j^{(l-1)}+m} \boldsymbol{b}_{j;v}^{l-1,\mathsf{T}} \tag{58}$$

Since $\boldsymbol{V}_i^{l-1} \cap \boldsymbol{V}_j^{l-1} \neq \emptyset$, there exists $j, v, i, v_0$ such that $\boldsymbol{b}_{j;v}^{l-1} = \boldsymbol{b}_{i;v_0}^{l-1}$. Moreover, since $1 \leqslant i3^L + u - d_i^{(l-1)} - j3^L - v + d_j^{(l-1)} + m \leqslant (n+1)3^L$ and there exists $j, v, i, v_0$ such that $\boldsymbol{b}_{j;v}^{l-1} = \boldsymbol{b}_{i;v_0}^{l-1}$, there exists $m_0$ s.t. $\boldsymbol{b}_{j;v}^{l-1} \boldsymbol{R}^{i3^L+u_0-d_i^{(l-1)}-j3^L-v_0+d_j^{(l-1)}+m_0} \boldsymbol{b}_{j;v_0}^{(l-1),\mathsf{T}} = 1$. This indicates that $\boldsymbol{A}_{i,j}^{(l-1)} \geq 1$

When $j = 1$,

$$\boldsymbol{A}_{i,1}^{(l-1)} = \sum_{u=1}^{C_i^{l-1}} \sum_{m=-(n+1)3^L}^{-1} \boldsymbol{b}_{i;u}^{l-1} \boldsymbol{R}^{i3^L+u-d_i^{(l-1)}-(n+1)3^L-1+m} \boldsymbol{X}_1^{tgt,\mathsf{T}} \tag{59}$$

Since $-2(n+1)3^L - 1 \leqslant i3^L + u - d_i^{(l-1)} - (n+1)3^L - 1 + m \leqslant -1$, we know that $\boldsymbol{A}_{i,1}^{(l-1)} = 0$. $\quad\square$

### E.4  INFORMATION PROPAGATION

Next, we explain how the above defined transformer extract reasoning result.

$\boldsymbol{X}^{ao(0)} = \boldsymbol{X}^{(0)} + \boldsymbol{X}^{qkv(0)}$

When $l = 0$, for $i \geqslant j$, using (44), (45), (46) and (48) we have

$$\boldsymbol{A}_{i,j}^{(0)} = \boldsymbol{X}_i^{(0)} \left( \sum_{t=1}^{\frac{n-1}{2}} \boldsymbol{p}_{2t} \boldsymbol{p}_{2t-1}^{\mathsf{T}} \right) \boldsymbol{X}_j^{(0),\mathsf{T}}$$

$$= \sum_{t=1}^{\frac{n-1}{2}} \boldsymbol{X}_i^{tgt} \boldsymbol{p}_{2t} \boldsymbol{p}_{2t-1}^{\mathsf{T}} \boldsymbol{X}_j^{tgt,\mathsf{T}} + \sum_{t=1}^{\frac{n-1}{2}} \boldsymbol{X}_i^{pos} \boldsymbol{p}_{2t} \boldsymbol{p}_{2t-1}^{\mathsf{T}} \boldsymbol{X}_j^{tgt,\mathsf{T}} + \sum_{t=1}^{\frac{n-1}{2}} \boldsymbol{X}_i^{pos} \boldsymbol{p}_{2t} \boldsymbol{p}_{2t-1}^{\mathsf{T}} \boldsymbol{X}_j^{pos,\mathsf{T}}$$

$$+ \sum_{t=1}^{\frac{n-1}{2}} \boldsymbol{X}_i^{tgt} \boldsymbol{p}_{2t} \boldsymbol{p}_{2t-1}^{\mathsf{T}} \boldsymbol{X}_j^{pos,\mathsf{T}} \tag{60}$$

It is clear that

$$\boldsymbol{A}_{i,j}^{(0)} = 1, \text{ when } i = j+1 \text{ and } i \bmod 2 = 0,$$
$$\boldsymbol{A}_{i,j}^{(0)} = 0, \text{ when } i \neq j+1 \text{ or } i \bmod 2 \neq 0. \tag{61}$$

Then for $m \in [1, n] \cap \mathbb{Z}$, we get $\boldsymbol{X}_m^{ao(0)} = \sum_{i=1}^{m} \frac{\exp(\boldsymbol{A}_{m,i}^{(0)})}{\sum_{j=1}^{m} \exp(\boldsymbol{A}_{m,j}^{(0)})} \boldsymbol{X}_i^{(0)} \boldsymbol{R} + \boldsymbol{X}_m^{(0)}$, when $m \bmod 2 = 0$, we can get reasoning chain $(\boldsymbol{b}_{m;1}^{(1)}, \boldsymbol{b}_{m;2}^{(1)})$.

In fact, we set all the coefficient $\frac{\exp(0)}{\sum_{j=1}^{m} \exp(\boldsymbol{A}_{m,j}^{(0)})}$ to be 0, then $\boldsymbol{X}_m^{ao(1)}$ is of the form $\frac{\exp(1)}{\sum_{t=1}^{m} \exp(\boldsymbol{A}_{m,t}^{(0)})} \boldsymbol{b}_{m;1}^{(1)} \boldsymbol{R} + \boldsymbol{b}_{m;2}^{(1)}$.

By (50) we have $L_N^0(f_m^{(0)}(\boldsymbol{X}_m^{ao(0)}) + \boldsymbol{X}_m^{ao(0)}) = \boldsymbol{b}_{m;1}^{(1)} \boldsymbol{R}^{m3^L-1} + \boldsymbol{b}_{m;2}^{(1)} \boldsymbol{R}^{m3^L}$

When $m \bmod 2 \neq 0$ and $m > 1$, we can recognize $\boldsymbol{b}_{m;1}^{(0)} = \boldsymbol{X}_m^{tgt}$, and $L_N^0(f_m^{(0)}(\boldsymbol{X}_m^{ao(0)}) + \boldsymbol{X}_m^{ao(0)}) = \boldsymbol{b}_{m;1}^{(1)} \boldsymbol{R}^{m3^s}$

When $m = 1$, we can recognize $\boldsymbol{b}_{1;1}^{(1)} = \boldsymbol{X}_1^{tgt}$, and $L_N^0(f_1^{(0)}(\boldsymbol{X}_1^{ao(0)} + \boldsymbol{X}_1^{ao(0)})) = \boldsymbol{b}_{1;1}^{(1)}\boldsymbol{R}^{(n+1)3^L}$.

After the first layer decoder, the even positions pass information to subsequent odd positions. And we can eliminate the position vectors on the second floor.

In fact, by our construction we have

$$
\boldsymbol{X}_i^{ao(l-1)} = \sum_{j=1}^{i} \frac{\exp(\boldsymbol{A}_{i,j}^{(l-1)})}{\sum_{t=1}^{i} \exp(\boldsymbol{A}_{i,t}^{(l-1)})} \boldsymbol{X}_j^{(l-1)} + \boldsymbol{X}_i^{(l-1)} \tag{62}
$$

$$
= \sum_{j=1}^{i} \frac{\exp(\boldsymbol{A}_{i,j}^{(l-1)})}{\sum_{t=1}^{i} \exp(\boldsymbol{A}_{i,t}^{(l-1)})} \sum_{u=1}^{C_j^{l-1}} \boldsymbol{b}_{j;u}^{l-1}\boldsymbol{R}^{j3^L + u - d_j^{(l-1)}} + \sum_{u=1}^{C_i^{l-1}} \boldsymbol{b}_{i;u}^{l-1}\boldsymbol{R}^{i3^L + u - d_i^{(l-1)}}. \tag{63}
$$

For any $i$, $j$ such that $i \neq j$, note that $|u - d_j^{(l-1)}| \leqslant \frac{3^L - 1}{2}$ and $|v - d_i^{(l-1)}| \leqslant \frac{3^L - 1}{2}$, by theorem 6.2, we have

$$
|j3^L + u - d_j^{(l-1)} - i3^L - v + d_i^{(l-1)}| = |(j - i)3^L + u - v - d_j^{(l-1)} + d_i^{(l-1)}| \tag{64}
$$

$$
\geqslant 3^L - |u - v - d_j^{(l-1)} + d_i^{(l-1)}| \geqslant 1, \tag{65}
$$

$$
|i3^L + u - d_i^{(l-1)}| \leqslant (n+1)3^L. \tag{66}
$$

And therefore, the nonzero element of $\boldsymbol{b}_{j;u}^{l-1}\boldsymbol{R}^{j3^L + u - d_j^{(l-1)}}$ and $\boldsymbol{b}_{i;u}^{l-1}\boldsymbol{R}^{i3^L + u - d_i^{(l-1)}}$ $(i \neq j)$ won't locate at the same coordinate.

Denote $\boldsymbol{X}_i^{ao(l-1)} = (\boldsymbol{X}_{i,1}^{ao(l-1)}, \cdots, \boldsymbol{X}_{i,d_m}^{ao(l-1)})$. Firstly, set the number on the axis which equals to $min_{1 \leqslant j \leqslant i}\{\boldsymbol{X}_{i,j}^{ao(l-1)} > 0\}$ to be 0.

Suppose the nonzero component of $X_1^{tgt}$ located at the $k_p$-th axis. Then, $X_{i,k_p - (n+1)3^L}^{ao(l-1)} = \frac{exp(0)}{\sum_{1 \leqslant j \leqslant i} exp(A_{i,j}^{(l-1)})}$. Since $\min_{1 \leqslant j \leqslant i}\{\boldsymbol{X}_{i,j}^{ao(l-1)} > 0\} = \frac{1}{\sum_{t=1}^{i} \exp(\boldsymbol{A}_{i,t}^{(l-1)})}$, there remains the information propagated from $\boldsymbol{X}_j^{(l-1)}$ s.t. $\boldsymbol{A}_{i,j}^{(l-1)} \geqslant 1$ which indicates that $\boldsymbol{V}_i^{l-1} \cap \boldsymbol{V}_j^{l-1} \neq \emptyset$. We use the sequence $(\boldsymbol{b}_{i;1}^{l-1}, \cdots, \boldsymbol{b}_{i;C_i^{l-1}}^{l-1})$ associated to $N_i^l$ and the sequence $(\boldsymbol{b}_{j;1}^{l-1}, \cdots, \boldsymbol{b}_{j;C_j^{l-1}}^{l-1})$ associated to $N_j^{l-1}$ to construct a new sequence $(\boldsymbol{b}_i^l)$ associated to $N_i^l$ in the following two rules.

- If $\boldsymbol{V}_i \subseteq \boldsymbol{V}_j$(resp. $\boldsymbol{V}_j \subseteq \boldsymbol{V}_i$), then we set $(\boldsymbol{b}_i^l) = (\boldsymbol{b}_i^{l-1})$(resp. $(\boldsymbol{b}_i^l) = (\boldsymbol{b}_j^{l-1})$).
- If $\boldsymbol{V}_i \not\subseteq \boldsymbol{V}_j$ and $\boldsymbol{V}_j \not\subseteq \boldsymbol{V}_i$. Since $\boldsymbol{V}_i \cap \boldsymbol{V}_j \neq \emptyset$ without loss of generality we assume the set $\boldsymbol{V}_i \cap \boldsymbol{V}_j$ is of the form $\{\boldsymbol{b}_{i;1}^{l-1}, \boldsymbol{b}_{i;2}^{l-1}, \cdots, \boldsymbol{b}_{i;k_i}^{l-1}\}$ for some $k_i \leqslant C_i^{l-1}$. Also, there exist $k_j \leqslant C_j^l$ such that $\boldsymbol{b}_{j;k_j}^{l-1} = \boldsymbol{b}_{i;1}^{l-1}$. And therefore the sequence $\boldsymbol{b}_i^l$ is set to be $(\boldsymbol{b}_{j;1}^{l-1}, \boldsymbol{b}_{j;2}^{l-1}, \cdots \boldsymbol{b}_{j;k_j}^{l-1}, \boldsymbol{b}_{i;2}^{l-1}, \boldsymbol{b}_{i;3}^{l-1} \cdots \boldsymbol{b}_{i;C_i^{l-1}}^{l-1})$.

Moreover, for a given node $N_i^{l-1}$ there might exist more than one node $N_j^{l-1}$ satisfying $\boldsymbol{A}_{i,j}^{(l-1)} \geqslant 1$. Denote $\Lambda_i^l = \{j | \boldsymbol{A}_{i,j}^{(l)} \geqslant 1\}$, then the information in each node $N_k^{l-1}$ with $k \in \Lambda_i^l$ is transmitted to $N_i^{l-1}$ as the above way by treating $\boldsymbol{b}_i^l$ as $\boldsymbol{b}_i^{l-1}$ each time. More specifically, we set initially $N_i^l = N_i^{l-1}$ and correspondingly $\boldsymbol{b}_i^l = \boldsymbol{b}_i^{l-1}$. Then for each $k \in \Lambda_i^l$ and for $\boldsymbol{b}_k^{l-1}$ associated to $N_k^{l-1}$, we update $\boldsymbol{b}_i^{(l)}$ as in the above two rules by setting $\boldsymbol{b}_i^{l-1} = \boldsymbol{b}_i^l$ and $\boldsymbol{b}_j^{l-1} = \boldsymbol{b}_k^{l-1}$.

**Remark E.3.** *The information propagation in this transformer satisfies the rules as we defined in section 5. Although we ignore the information propagated from the node $N_1^{l-1}$ by setting $\boldsymbol{A}_{i,1}^{(l-1)} = 0$ for $l \geqslant 1$, there won't be any information loss. Since the first node in each layer will only contain one value which is also contained in $N_2^1$ by Rule 2.*

### E.5 EXISTENCE OF APPROXIMATING FNN AND ERROR ANALYSIS

We find the FNN we required in three steps.

- **Step 1** Find continuous functions that decode the information;
- **Step 2** Extend the continous function to allow small error;
- **Step 3** Use universal approximation theorem to find a FNN to approximate the extended continuous functions.

STEP 1, CONTINUOUS FUNTIONS

Since $\{(\boldsymbol{z}_1, \cdots, \boldsymbol{z}_{d_m}) | \boldsymbol{z}_i \in \{0, 1\}\} \subseteq \mathrm{Range}(L_N)$, there exists continuous functions $f_i^{l-1}$ s.t.

$$L_N^{l-1}(f_i^{l-1}(\boldsymbol{X}_i^{ao(l-1)})) = \sum_{u=1}^{C_i^{(l)}} \boldsymbol{b}_{i;u}^l \boldsymbol{R}^{i3^{s-1}+u-d_i^{(l)}}. \tag{67}$$

By the universal approximation theorem (Theorem E.9), we know that a neural network can approximate any continuous function with arbitrarily small error. In fact, we can prove the following theorem.

**Lemma E.4.** *If $L_N(f)$ is a simple function, there exists a single-hidden-layer neural network $f'$ for any $\epsilon'$, such that:*

$$\sup_{x \in K} ||L_N(f(x)) - L_N(f'(x))|| < \epsilon'$$

*where $K \subseteq \mathbb{R}^d$ is an arbitrary compact set.*

Here and in the sequel, we use the notation $|| \cdot ||$ to denote the $\infty$ norm of vectors, i.e. for $v = (v_1, v_2, \cdots, v_{d_m}) \in \mathbb{R}^{d_m}$, $||v|| = \max_{1 \leqslant i \leqslant d_m} |v_i|$.

As we have discussed earlier in remark E.1, we can not define a FNN such that it satisfies (50). However, as we have shown in Lemma E.4, we can find a FNN such that it differs from (50) by a small error $\varepsilon'$. And we now analyze the effect caused by this small error during the information propagation.

STEP 2: EXPANSION OF $f_i$

To proceed, we shall use the following notations. Note that for a given node $N_i^l$ the reasoning sequence contained in this node can be transmitted from an input matrix $(\boldsymbol{X}_i^{tgt})$ or a permutation of $(\boldsymbol{X}_i^{tgt})$ say $(\boldsymbol{X}_{\sigma(i)}^{tgt})$ where $\sigma \in \mathbb{Z} \cap [1, n]$. In this case we denote correspondingly the output of $l$th layer in transformer as $\boldsymbol{X}_{\sigma,i}^{(l)}$ or simply $\boldsymbol{X}_{\sigma}^{(l)}$. Also, when the input matrix is set to be $\boldsymbol{X}_{\sigma(i)}^{tgt}$, we denote correspondingly the sequence associated to each node $N_i^l$ as $\boldsymbol{b}_{\sigma,i;u}^l$, the attention matrices $\boldsymbol{A}$ as $\boldsymbol{A}_{\sigma,i,j}$ and $(\boldsymbol{X}_i^{ao(l)})$ as $(\boldsymbol{X}_{\sigma,i}^{ao(l)})$.

Moreover, for a given input matrix $(\boldsymbol{X}_i^{tgt})$ and for fixed $i$, $l$ and $\sigma$ we denote the set

$$\begin{aligned} D_{\sigma,i}^l = \{\boldsymbol{Y} \in \mathbb{R}^{d_m} : L_N^l \circ f_i^l(\boldsymbol{Y}) = \boldsymbol{X}_{\sigma,i}^{(l)}\} \\ \cap \{\boldsymbol{Y} \in \mathbb{R}^{d_m} : \boldsymbol{Y} = \boldsymbol{X}_{\sigma,i}^{ao(l-1)}\} \end{aligned} \tag{68}$$

And for a given input matrix $(\boldsymbol{X}_i^{tgt})$ we define the equivalence class of permutations as follows

**Definition E.5.** *For fixed $i \in [1, n] \cap \mathbb{Z}$ and fixed $l \in [1, L] \cap \mathbb{Z}$, two permutations $\sigma \in \mathrm{Sym}(\mathbb{Z} \cap [1, n])$ and $\tau \in \mathrm{Sym}(\mathbb{Z} \cap [1, n])$ are said to be $N_i^l$ level equivalent if and only if they satisfy*

$$\boldsymbol{X}_{\sigma,i}^{(l)} = \boldsymbol{X}_{\tau,i}^{(l)}. \tag{69}$$

*And the equivalence class of $\sigma$ is denoted as*

$$[\sigma]_i^l = \{\tau \in \mathrm{Sym}(\mathbb{Z} \cap [1, n]) : \boldsymbol{X}_{\sigma,i}^{(l)} = \boldsymbol{X}_{\tau,i}^{(l)}\}. \tag{70}$$

*Moreover, we denote $E_i^l$ the set of all the $N_i^l$ level equivalent classes.*

It is clear that $D_{\sigma,i}^l$ are all finite sets since $\mathrm{Sym}(\mathbb{Z} \cap [1,n])$ is a finite set. We shall also use the notation $d(x,y) = \|x - y\|$.

To expand the $f_i^l$ defined in (67), we need the following lemma.

**Lemma E.6.** *For $\sigma_1, \sigma_2 \in \mathrm{Sym}(\mathbb{Z} \cap [1,n])$, and for $i \in \mathbb{Z} \cap [1,n]$, if $(b_{\sigma_1,i}^{l+1}) \neq (b_{\sigma_2,i}^{l+1})$, then for $\boldsymbol{X} \in D_{\sigma_1,i}^{l+1}$ and $\boldsymbol{Y} \in D_{\sigma_2,i}^{l+1}$ we have $d(\boldsymbol{X}, \boldsymbol{Y}) > 0$.*

Since the sets $D_{\sigma,i}^l$ are all finite, then by Lemma E.6 for $\sigma_1$ and $\sigma_2$ satisfying $\boldsymbol{V}_{\sigma_1}^l \neq \boldsymbol{V}_{\sigma_2}^l$ we have

$$d_i^{l+1} = \min_{\boldsymbol{X} \in D_{\sigma_1,i}^l, \boldsymbol{Y} \in D_{\sigma_2,i}^l} d(\boldsymbol{X}, \boldsymbol{Y}) > 0. \tag{71}$$

We now define the expansion of $f_i^l$ as follows

$$\tilde{f}_i^l = (L_N^l)^{-1} \left( \sum_{[\sigma]_i^l \in E_i^l} \mathbf{1}_{D_{\sigma,i}^{l+1} + [-\delta_i^{l+1}, \delta_i^{l+1}]^n} \right), \tag{72}$$

for some $\delta_i^{l+1} \in (0, d_i^{l+1})$. Here the symbol $+$ in $D_{\sigma,i}^{l+1} + [-\delta_i^{l+1}, \delta_i^{l+1}]^n$ denotes the addition of sets in $\mathbb{R}^n$, and the condition $\delta_i^{l+1} < d_i^{(l+1)}$ ensures that (72) are well-defined

**STEP 3**, APPROXIMATING FNN

Note that for given $i$, $l$ and $\sigma$ the set $D_{\sigma,i}^{l+1} + [-\delta_i^{l+1}, \delta_i^{l+1}]^n$ is a compact set, according to Lemma E.4, $\forall \eta_i^{l+1} > 0$, there exist a single-hidden-layer neural network $\hat{f}_i^l$ such that

$$\sup_{\boldsymbol{X} \in D_{\sigma,i}^{l+1} + [-\delta_i^{l+1}, \delta_i^{l+1}]^n} \|L_N^l(\hat{f}_i^l)(\boldsymbol{X}) - L_N^l(\tilde{f}_i^l)(\boldsymbol{X})\| < \eta_i^{(l+1)}. \tag{73}$$

This $\hat{f}$ is the FNN we are looking for which can tranmit information as $f$. In fact we have following proposition.

**Proposition E.7.** *For given $\boldsymbol{X}_{\sigma,i}^{ao(l)}$, we have*

$$\begin{aligned} L_N^l(f_i^{(l)}(\boldsymbol{X}_{\sigma,i}^{ao(l)})) &= \boldsymbol{X}_{\sigma,i}^{(l+1)}; \\ L_N^l(\hat{f}_i^{(l)}(\boldsymbol{X}_{\sigma,i}^{ao(l)})) &\in \{\boldsymbol{X}_{\sigma,i}^{(l+1)}\} + [-\epsilon, \epsilon]^n. \end{aligned} \tag{74}$$

*Proof.* Without loss of generality we set $L_N^l(\hat{f}_i^{(l)}(\boldsymbol{X}_{\sigma,i}^{ao(l)})) = \sum_{u=1}^{C_i^{(l+1)}} \boldsymbol{b}_{iu}^{l+1} R^{i3^{s-1} + u - d_i^{(l+1)}} + \epsilon_i^{(l+1)} \boldsymbol{r}_i^{(l+1)} = \boldsymbol{X}_{\sigma,i}^{(l+1)}$, where $|\boldsymbol{r}_i| = 1$, $\boldsymbol{r}_i \in \mathbb{R}^{d_m}$. Take $\epsilon_i^{(l)} < \epsilon$ and set

$$\bar{\boldsymbol{A}}_{i,j}^{(l)} = (\sum_{u=1}^{C_i^l} \boldsymbol{b}_{i;u}^l \boldsymbol{W}^{i3^L + u - d_i^{(l)}} + \epsilon_i^{(l)} \boldsymbol{r}_i^{(l)})(\sum_{m=-(n+1)3^L}^{-1} \boldsymbol{W}^m)(\sum_{v=1}^{C_j^l} \boldsymbol{b}_{j;v}^l \boldsymbol{W}^{j3^L + v - d_j^{(l)}} + \epsilon_j^{(l)} \boldsymbol{r}_j^{(l)})^{\mathsf{T}}$$

Since $\bar{\boldsymbol{A}}_{i,j}^{(l)} = \boldsymbol{A}_{i,j}^{(l)} + \epsilon_i^{(l)} \boldsymbol{r}_i^{(l)} (\sum_{m=-(n+1)3^L}^{-1} \boldsymbol{W}^m) \epsilon_j^{(l)} \boldsymbol{r}_j^{(l)}$, we have $\forall i,j,l$, $|\bar{\boldsymbol{A}}_{i,j}^{(l)} - \boldsymbol{A}_{i,j}^{(l)}| \leqslant 3^L(n+1)\epsilon_i^{(l)}\epsilon_j^{(l)} \leqslant \eta_0$.

And therefore,

$$\bar{\boldsymbol{X}}_{\sigma,i}^{ao(l)} = \sum_{j=1}^{i} \frac{\exp(\bar{\boldsymbol{A}}_{i,j}^{(l)})}{\sum_{t=1}^{i} \exp(\bar{\boldsymbol{A}}_{i,t}^{(l)})} \bar{\boldsymbol{X}}_{\sigma,j}^{(l)} + \bar{\boldsymbol{X}}_{\sigma,i}^{(l)}$$

$$= \sum_{j=1}^{i} \frac{\exp(\bar{\boldsymbol{A}}_{i,j}^{(l)})}{\sum_{t=1}^{i} \exp(\bar{\boldsymbol{A}}_{i,t}^{(l)})} (\sum_{u=1}^{C_j^l} \boldsymbol{b}_{j;u}^l \boldsymbol{W}^{j3^L+u-d_j^{(l)}} + \epsilon_j^{(l)} \boldsymbol{r}_j^{(l)}) + \sum_{u=1}^{C_i^l} \boldsymbol{b}_{i;u}^l \boldsymbol{W}^{i3^L+u-d_i^{(l)}} + \epsilon_i^{(l)} \boldsymbol{r}_i^{(l)}$$

$$= \sum_{j=1}^{i} \frac{\exp(\bar{\boldsymbol{A}}_{i,j}^{(l)})}{\sum_{t=1}^{i} \exp(\bar{\boldsymbol{A}}_{i,t}^{(l)})} (\sum_{u=1}^{C_j^l} \boldsymbol{b}_{j;u}^l \boldsymbol{W}^{j3^L+u-d_j^{(l)}}) + \sum_{u=1}^{C_i^l} \boldsymbol{b}_{i;u}^l \boldsymbol{W}^{i3^L+u-d_i^{(l)}}$$

$$+ \sum_{j=1}^{i} \frac{\exp(\bar{\boldsymbol{A}}_{i,j}^{(l)})}{\sum_{t=1}^{i} \exp(\bar{\boldsymbol{A}}_{i,t}^{(l)})} \epsilon_j^{(l)} \boldsymbol{r}_j^{(l)} + \epsilon_i^{(l)} \boldsymbol{r}_i^{(l)}, \tag{75}$$

along with

$$\bar{\boldsymbol{X}}_{\sigma,i}^{ao(l)} - \boldsymbol{X}_{\sigma,i}^{ao(l)} = \sum_{j=1}^{i} \left[ \frac{\exp(\bar{\boldsymbol{A}}_{i,j}^{(l)})}{\sum_{t=1}^{i} \exp(\bar{\boldsymbol{A}}_{i,t}^{(l)})} - \frac{\exp(\boldsymbol{A}_{i,j}^{(l)})}{\sum_{t=1}^{i} \exp(\boldsymbol{A}_{i,t}^{(l)})} \right] (\sum_{u=1}^{C_j^l} \boldsymbol{b}_{ju}^l \boldsymbol{W}^{j3^{s-1}+u-d_j^{(l)}})$$

$$+ \sum_{j=1}^{i} \frac{\exp(\bar{\boldsymbol{A}}_{i,j}^{(l)})}{\sum_{t=1}^{i} \exp(\bar{\boldsymbol{A}}_{i,t}^{(l)})} \epsilon_j^{(l)} \boldsymbol{r}_j^{(l)} + \epsilon_i^{(l)} \boldsymbol{r}_i^{(l)} \tag{76}$$

Direct calculation and (76) leads to

$$\|\bar{\boldsymbol{X}}_{\sigma,i}^{ao(l)} - \boldsymbol{X}_{\sigma,i}^{ao(l)}\| \leqslant \mathrm{I} + \mathrm{II}, \tag{77}$$

where

$$\mathrm{I} = \max_{i,j,l} |\frac{\exp(\bar{\boldsymbol{A}}_{i,j}^{(l)})}{\sum_{t=1}^{i} \exp(\bar{\boldsymbol{A}}_{i,t}^{(l)})} - \frac{\exp(\boldsymbol{A}_{i,j}^{(l)})}{\sum_{t=1}^{i} \exp(\boldsymbol{A}_{i,t}^{(l)})}|, \tag{78}$$

$$\mathrm{II} = |\sum_{j=1}^{i} \frac{\exp(\bar{\boldsymbol{A}}_{i,j}^{(l)})}{\sum_{t=1}^{i} \exp(\bar{\boldsymbol{A}}_{i,t}^{(l)})} \epsilon_j^{(l)} \boldsymbol{r}_j^{(l)} + \epsilon_i^{(l)} \boldsymbol{r}_i^{(l)}|. \tag{79}$$

Take $\eta_0$ small enough such that $\forall |x| < \eta_0, |\exp(x) - 1| < 2x$. We then have

$$\mathrm{I} \leqslant \max \left\{ \frac{|\exp(\bar{\boldsymbol{A}}_{i,j}^{(l)}) - \exp(\boldsymbol{A}_{i,j}^{(l)})| \, |\sum_{t=1}^{i} \exp(\boldsymbol{A}_{i,j}^{(l)})|}{(\sum_{t=1}^{i} \exp(\boldsymbol{A}_{i,j}^{(l)}))(\sum_{t=1}^{i} \exp(\bar{\boldsymbol{A}}_{i,j}^{(l+1)}))} \right.$$

$$+ \frac{|\sum_{t=1}^{i}(\exp(\boldsymbol{A}_{i,j}^{(l)}) - \exp(\bar{\boldsymbol{A}}_{i,j}^{(l)}))| \, |\exp(\boldsymbol{A}_{i,j}^{(l)})|}{(\sum_{t=1}^{i} \exp(\boldsymbol{A}_{i,j}^{(l)}))(\sum_{t=1}^{i} \exp(\bar{\boldsymbol{A}}_{i,j}^{(l)}))} \right\}$$

$$\leqslant \max \left\{ n \times |\exp(\bar{\boldsymbol{A}}_{i,j}^{(l)} - \boldsymbol{A}_{i,j}^{(l)}) - 1| \exp(\boldsymbol{A}_{i,j}^{(l)}) \exp(M) \right. \tag{80}$$

$$+ (\sum_{t=1}^{i} |\exp(\bar{\boldsymbol{A}}_{i,t}^{(l)} - \boldsymbol{A}_{i,t}^{(l)}) - 1|) \exp(\boldsymbol{A}_{i,j}^{(l)}) \exp(M)) \right\}$$

$$\leqslant 2n\eta_0 \exp(2M) + 2n\eta_0 \exp(2M)$$

$$\leqslant 4n\eta_0 \exp(2M)$$

$$\mathrm{II} \leqslant (n+1)\epsilon. \tag{81}$$

Combining (76), (80) and (81) leads to

$$\|\bar{\boldsymbol{X}}_{\sigma,i}^{ao(l)} - \boldsymbol{X}_{\sigma,i}^{ao(l)}\| \leqslant \mathrm{I} + \mathrm{II} \leqslant 4n\eta_0 \exp(2M) + (n+1)\epsilon. \tag{82}$$

We then choose $\eta_0$ and $\epsilon$ small such that $4n\eta_0 \exp(2M) + (n+1)\epsilon < \delta_i^{(l)}$, and thus $\bar{\boldsymbol{X}}_{\sigma,i}^{ao(l)} \in D_{\sigma,i}^{l+1} + (-\delta_i^{l+1}, \delta_i^{l+1})^n$. Moreover, we have

$$L_N^l(\hat{f}_i^{(l)}(\boldsymbol{X}_{\sigma,i}^{ao(l)})) = \sum_{u=1}^{C_i^{(l+1)}} \boldsymbol{b}_{i;u}^{l+1} \boldsymbol{R}^{i3^{s-1}+u-d_i^{(l+1)}} + \epsilon_i^{(l+1)} \boldsymbol{r}_i^{(l+1)} \in \boldsymbol{X}_{\sigma,i}^{(l+1)} + [-\epsilon, \epsilon]^n, \tag{83}$$

where $|\boldsymbol{r}_i| = 1, \boldsymbol{r}_i \in \mathbb{R}^{d_m}$. This completes the proof of our proposition. $\square$

**Lemma E.8.** *LayerNorm of the form $L_N(x) = \alpha \frac{x - \mathrm{E}(x)}{\sqrt{\mathrm{Var}(x)+\epsilon}} + \beta$, where $\alpha, \beta$ and $\epsilon$ are constants and the function $\mathrm{E}(\cdot), \mathrm{Var}(\cdot)$ stands for the expectation and variance respectively, is injective (i.e., For any $x_1 \neq x_2$, we have $L_N(x_1) \neq L_N(x_2)$).*

***Proof of Lemma E.8.*** Note that $L_N(x) = \alpha \frac{x - \mathrm{E}(x)}{\sqrt{\mathrm{Var}(x)+\epsilon}} + \beta$,

For any $x_1 \neq x_2$, if $L_N(x_1) = L_N(x_2)$, then we have

$$(\alpha \frac{x_1^1 - \mathrm{E}(x_1)}{\sqrt{\mathrm{Var}(x_1)+\epsilon}} + \beta, \cdots, \alpha \frac{x_1^n - \mathrm{E}(x_1)}{\sqrt{\mathrm{Var}(x_1)+\epsilon}} + \beta) = (\alpha \frac{x_2^1 - \mathrm{E}(x_2)}{\sqrt{\mathrm{Var}(x_2)+\epsilon}} + \beta, \cdots, \alpha \frac{x_2^n - \mathrm{E}(x_2)}{\sqrt{\mathrm{Var}(x_2)+\epsilon}} + \beta),$$

which leads to

$$\alpha \frac{x_1^i - \mathrm{E}(x_1)}{\sqrt{\mathrm{Var}(x_1)+\epsilon}} + \beta = \alpha \frac{x_2^i - \mathrm{E}(x_2)}{\sqrt{\mathrm{Var}(x_2)+\epsilon}} + \beta, \text{ for } 1 \leqslant i \leqslant n. \tag{84}$$

Therefore, summation from $1$ to $n$ in both sides of (84) leads to

$$\alpha \frac{n\,\mathrm{E}(x_1) - \mathrm{E}(x_1)}{\sqrt{\mathrm{Var}(x_1)+\epsilon}} + n\beta = \alpha \frac{n\,\mathrm{E}(x_2) - \mathrm{E}(x_2)}{\sqrt{\mathrm{Var}(x_2)+\epsilon}} + n\beta,$$

which indicates that

$$\frac{\mathrm{E}(x_1)}{\sqrt{\mathrm{Var}(x_1)+\epsilon}} = \frac{\mathrm{E}(x_2)}{\sqrt{\mathrm{Var}(x_2)+\epsilon}}. \tag{85}$$

By (84), we also have

$$\frac{x_1^i}{\sqrt{\mathrm{Var}(x_1)+\epsilon}} = \frac{x_2^i}{\sqrt{\mathrm{Var}(x_2)+\epsilon}}, \text{ for } 1 \leqslant i \leqslant n. \tag{86}$$

Combining (85) and (86) yields that

$$\frac{x_1}{x_2} = \frac{\mathrm{E}(x_1)}{\mathrm{E}(x_2)} = \frac{\sqrt{\mathrm{Var}(x_1)+\epsilon}}{\sqrt{\mathrm{Var}(x_2)+\epsilon}}. \tag{87}$$

Set $k = \frac{\mathrm{E}(x_1)}{\mathrm{E}(x_2)}$, then $x_1^i = kx_2^i$, $\mathrm{E}(x_1) = k\,\mathrm{E}(x_2)$, and therefore $\mathrm{Var}(x_1) = k^2\,\mathrm{Var}(x_2)$. These relations together with (86) lead to

$$\frac{x_1^i}{\sqrt{\mathrm{Var}(x_1)+\epsilon}} = \frac{kx_1^i}{\sqrt{k^2\,\mathrm{Var}(x_1)+\epsilon}}, \text{ for } 1 \leqslant i \leqslant n. \tag{88}$$

Since $\epsilon > 0$ and $\alpha \neq 0$, it is cleat that $k$ must be 1, which contradicts with $x_1 \neq x_2$. $\square$

***Proof of Lemma E.6.*** We prove this lemma by contradiction.

Suppose that there exist $\boldsymbol{X} \in D^{l+1}_{\sigma_1,i}$ and $\boldsymbol{Y} \in D^{l+1}_{\sigma_2,i}$ such that $d(\boldsymbol{X}, \boldsymbol{Y}) = 0$.

Since $(\boldsymbol{b}^{l+1}_{\sigma_1,i}) \neq (\boldsymbol{b}^{l+1}_{\sigma_2,i})$, there exists $i_0$ and $j_0$ such that $\boldsymbol{b}^l_{\sigma_1,i_0,j_0} \in \{\boldsymbol{b}^{l+1}_{\sigma_1,i,1}, ..., \boldsymbol{b}^{l+1}_{\sigma_1,i,c^{(l+1)}_{\sigma_1,i}}\}$ and $\boldsymbol{b}^l_{\sigma_1,i_0,j_0} \notin \{\boldsymbol{b}^{l+1}_{i,\sigma_2,1}, ..., \boldsymbol{b}^{l+1}_{i,\sigma_2,c^{(l+1)}_{i,\sigma_2}}\}$. For simplicity, we denote $h = \boldsymbol{b}^l_{\sigma_1,i_0,j_0}$.

By (68), we know that

$$\boldsymbol{X} = \boldsymbol{X}^{ao(l)}_{\sigma_1,i} = \sum_{j=1}^{i} \frac{\exp(A^{(l)}_{\sigma_1,i,j})}{\sum_{t=1}^{i} \exp(A^{(l)}_{\sigma_1,i,t})} \boldsymbol{X}^{(l)}_{\sigma_1,j} + \boldsymbol{X}^{(l)}_{\sigma_1,i} \tag{89}$$

$$= \sum_{j=1}^{i} \frac{\exp(A^{(l)}_{\sigma_1,i,j})}{\sum_{t=1}^{i} \exp(A^{(l)}_{\sigma_1,i,t})} \sum_{u=1}^{C^l_{\sigma_1,j}} \boldsymbol{b}_{\sigma_1,j;u} \boldsymbol{R}^{j \times 3^L + u - d^{(l)}_{\sigma_1,j}} + \sum_{u=1}^{C^l_{\sigma_1,j}} \boldsymbol{b}^l_{\sigma_1,i;u} \boldsymbol{R}^{i \times 3^L + u - d^{(l)}_{\sigma_1,i}} \tag{90}$$

$$\boldsymbol{Y} = \boldsymbol{X}^{ao(l)}_{\sigma_2,i} = \sum_{j=1}^{i} \frac{\exp(A^{(l)}_{\sigma_2,i,j})}{\sum_{t=1}^{i} \exp(A^{(l)}_{\sigma_2,i,t})} \boldsymbol{X}^{(l)}_{\sigma_2,j} + \boldsymbol{X}^{(l)}_{\sigma_2,i} \tag{91}$$

$$= \sum_{j=1}^{i} \frac{\exp(A^{(l)}_{\sigma_2,i,j})}{\sum_{t=1}^{i} \exp(A^{(l)}_{\sigma_2,i,t})} \sum_{u=1}^{C^l_{\sigma_2,j}} \boldsymbol{b}_{\sigma_2,j;u} \boldsymbol{R}^{j \times 3^L + u - d^{(l)}_{\sigma_2,j}} + \sum_{u=1}^{C^l_{\sigma_2,j}} \boldsymbol{b}^l_{\sigma_2,i;u} \boldsymbol{R}^{i \times 3^L + u - d^{(l)}_{\sigma_2,i}} \tag{92}$$

If $h \notin \cup_{t=1}^{i} V^l_t$, then we know that $X^{ao(l)}_{\sigma_1,i,u} \geqslant \frac{\exp(1)}{\sum_{t=1}^{i} \exp(\boldsymbol{A}^{(l)}_{\sigma_1,t})}$ and $X^{ao(l)}_{\sigma_2,i,u} = 0$ which contradicts with our assumption.

If $h \in \cup_{t=1}^{i} V^l_t$ then $X^{ao(l)}_{\sigma_2,i,u}$ can either be $\frac{\exp(0)}{\sum_{t=1}^{i} \exp(\boldsymbol{A}^{(l)}_{\sigma_2,t})}$ or $0$. The case $X^{ao(l)}_{\sigma_2,i,u} = 0$ clearly contradicts with our assumption. Hence, we consider only the case $X^{ao(l)}_{\sigma_2,i,u} = \frac{\exp(0)}{\sum_{t=1}^{i} \exp(\boldsymbol{A}^{(l)}_{\sigma_2,t})}$. And we then know that

$$X^{ao(l)}_{\sigma_1,i,u} \geqslant \frac{\exp(1)}{\sum_{t=1}^{i} \exp(\boldsymbol{A}^{(l)}_{\sigma_1,t})}, \quad X^{ao(l)}_{\sigma_2,i,u} = \frac{\exp(0)}{\sum_{t=1}^{i} \exp(\boldsymbol{A}^{(l)}_{\sigma_2,t})}. \tag{93}$$

Suppose the nonzero components of $\boldsymbol{X}^{(tgt)}_{\sigma_1,1}$ and $\boldsymbol{X}^{(tgt)}_{\sigma_2,1}$ located at as the $k_{\sigma_1,1}$-th axis and the $k_{\sigma_2,1}$-th axis respectively. Note that

$$\boldsymbol{X}^{(l)}_{\sigma_1,1} = \boldsymbol{X}^{tgt}_{\sigma_1,1} R^{(n+1)3^L}, \tag{94}$$

$$\boldsymbol{X}^{(l)}_{\sigma_2,1} = \boldsymbol{X}^{tgt}_{\sigma_2,1} R^{(n+1)3^L}, \tag{95}$$

and that by (43) the distance between any two embedding axis $\geqslant 2(n+1)(3^L + 1)$, we have

$$\boldsymbol{X}^{ao(l)}_{\sigma_1,i,k_{\sigma_1,1}-3^{s-1}n} = \frac{\exp(0)}{\sum_{t=1}^{i} \exp(\boldsymbol{A}^{(l)}_{\sigma_1,i,t})}, \tag{96}$$

$$\boldsymbol{X}^{ao(l)}_{\sigma_2,i,k_{\sigma_2,1}-3^{s-1}n} = \frac{\exp(0)}{\sum_{t=1}^{i} \exp(\boldsymbol{A}^{(l)}_{\sigma_2,i,t})}. \tag{97}$$

If $k_{\sigma_1,1} \neq k_{\sigma_2,1}$,

$$\frac{\exp(0)}{\sum_{t=1}^{i} \exp(\boldsymbol{A}_{\sigma_2,i,t}^{(l)})} = \boldsymbol{X}_{\sigma_1,i,k_{\sigma_1,1}-3^L n}^{ao(l)} = \boldsymbol{X}_{\sigma_2,i,k_{\sigma_1,1}-3^L n}^{ao(l)} = 0$$

,

which is impossible. Therefore, we have $k_{\sigma_1,1} = k_{\sigma_2,1}$.

In addition, there exist $\boldsymbol{v}_q \in \tilde{E}$ and the corresponding $k_q$ such that $k_q = k_{\sigma_1,1} = k_{\sigma_2,1}$, the components of $\boldsymbol{X}_{\sigma_1,i}^{ao(l)}$ and $\boldsymbol{X}_{\sigma_2,i}^{ao(l)}$ on the $(k_q - (n+1)3^L)$-th axis are equal, which leads to

$$\frac{1}{\sum_{t=1}^{i} \exp(A_{\sigma_1,i},t)} = \frac{1}{\sum_{t=1}^{i} \exp(A_{i,\sigma_2},t)}. \tag{98}$$

Combining (93) and (98) leads to contradiction with $d(\boldsymbol{X},\boldsymbol{Y}) = 0$. And we complete the proof of Lemma E.6. $\square$

***Proof of Lemma E.4.*** By Lemma E.8, we can easily know that $(L_N^l)^{-1}(L_N^l(f)) = f$ is also a simple function.

According to the lemma above,

there exists a single-hidden-layer neural network $f'$ for any $\epsilon_0$ s.t

$$\sup_{x \in K} ||f(x) - f'(x)|| < \epsilon_0$$

Set $M = \max_{x \in K} ||f'(x)||$, for any $||x - y|| < \epsilon_0$,

$$\begin{aligned}
|L_N(x) - L_N(y)| &= |\alpha \frac{x - \mathrm{E}(x)}{\sqrt{\mathrm{Var}(x) + \epsilon}} - \alpha \frac{y - \mathrm{E}(y)}{\sqrt{\mathrm{Var}(y) + \epsilon}}| \\
&= |\alpha| \times |\frac{(x - \mathrm{E}(x))\sqrt{\mathrm{Var}(y) + \epsilon} - (y - \mathrm{E}(y))\sqrt{\mathrm{Var}(x) + \epsilon}}{\sqrt{\mathrm{Var}(y) + \epsilon}\sqrt{\mathrm{Var}(x) + \epsilon}}| \\
&\leqslant |\alpha| \times \frac{||(x - y - (\mathrm{E}(x) - \mathrm{E}(y)))||\sqrt{\mathrm{Var}(y) + \epsilon}}{\epsilon} \\
&+ |\alpha| \times \frac{||y - \mathrm{E}(y)|| \, |\sqrt{\mathrm{Var}(x) + \epsilon} - \sqrt{\mathrm{Var}(y) + \epsilon}|}{\varepsilon}.
\end{aligned} \tag{99}$$

Since $||\mathrm{E}(x) - \mathrm{E}(y)|| \leqslant ||x - y|| \leqslant \epsilon_0$ and $\mathrm{Var}(y) = \mathrm{E}(y)^2 - (\mathrm{E}(y))^2 \leqslant \mathrm{E}(y)^2 \leqslant (M + \epsilon_0)^2$ we have

$$\begin{aligned}
|\mathrm{E}(x)^2 - \mathrm{E}(y)^2| &= |\mathrm{E}(x - y)(x + y)| \\
&\leqslant \sqrt{\mathrm{E}(x - y)^2 \, \mathrm{E}(x + y)^2} \\
&\leqslant \sqrt{\epsilon_0^2 \, (2M + \epsilon_0)^2},
\end{aligned} \tag{100}$$

and

$$\begin{aligned}
|\sqrt{\mathrm{Var}(x) + \epsilon} - \sqrt{\mathrm{Var}(y) + \epsilon}| &= \frac{|\mathrm{Var}(x) - \mathrm{Var}(y)|}{\sqrt{\mathrm{Var}(x) + \epsilon}\sqrt{\mathrm{Var}(y) + \epsilon}} \\
&\leqslant \frac{|\mathrm{E}(x)^2 - \mathrm{E}(y)^2| + |(\mathrm{E}(x))^2 - (\mathrm{E}(y))^2|}{\epsilon} \\
&\leqslant \frac{\epsilon_0(2M + \epsilon_0) + \epsilon_0(2M + \epsilon_0)}{\epsilon},
\end{aligned} \tag{101}$$

$$|L_N(x) - L_N(y)| \leqslant |\alpha| \frac{2\epsilon_0 \sqrt{(M + \epsilon_0)^2 + \epsilon} + (M + \epsilon_0)(\frac{\epsilon_0(2M+\epsilon_0)+\epsilon_0(2M+\epsilon_0)}{\epsilon})}{\epsilon}. \tag{102}$$

We can set $\epsilon_0$ small enough such that

$$\sup_{x \in K} ||L_N(f(x)) - L_N(f'(x))|| \leqslant \sup_{\forall |x-y| < \epsilon_0} |L_N(x) - L_N(y)| < \epsilon'.$$

$\square$

**Theorem E.9** (Universal Approximation Theorem(Cybenko, 1989))**.** *For any given continuous function* $f : \mathbb{R}^d \to \mathbb{R}^n$ *and an allowable error* $\epsilon > 0$*, there exists a single-hidden-layer neural network* $f_\theta$ *with appropriate parameters* $\theta$*, such that:*

$$\sup_{x \in K} ||f(x) - f_\theta(x)||_\infty < \epsilon,$$

*where* $K \subseteq \mathbb{R}^d$ *is an arbitrary compact set.*

## F    DETAILS OF THE EXPERIMENT

### F.1    DATASET

We require reasoning sequence $(x_i)_{1 \leq i \leq 2s}$ of the training set satisfy the following condition.

$$x_{2i} - x_{2i-1} \bmod 5 \in \{0, 1, 4\} \tag{103}$$

The sequence of the test set satisfy:

$$x_{2i} - x_{2i-1} \bmod 5 \in \{2, 3\} \tag{104}$$

The values of each token range from 20 to 100,i.e.,$x_i \in [20, 100]$.

### F.2    HYPERPARAMETERS

In this section, the fixed and tunable hyperparameters employed in the model are outlined.

The fixed hyperparameters are as follows. Transformer architecture uses one attention head per layer. The dataset is partitioned into a training set comprising 90% of the data and a test set comprising the remaining 10%. Training is conducted over 2000 epochs. A weight decay of 0.1 is applied. The dimension of the model $d_m$ is set equal to the key dimension $d_k$. The feed-forward network dimension $d_{feedforward}$ is set to 1200.

Table 2:

| the number of reasoning steps | 3 | 4 | 5 |
|---|---|---|---|
| the size of datasets | 1200000 | 6000000 | 30000000 |

### F.3    LOWER BOUND: EXPERIMENTAL RESULTS OF A 3-LAYER TRANSFORMER ON 3-STEP REASONING

We train a 3-layer Transformer on sequences of length 21, using a batch size of 1000 and a learning rate of $5e{-}5$, to perform 3-step reasoning. Fig. 9 shows our training results. As the hidden dimension $d_m$ increases, the model generalizes more easily during training.

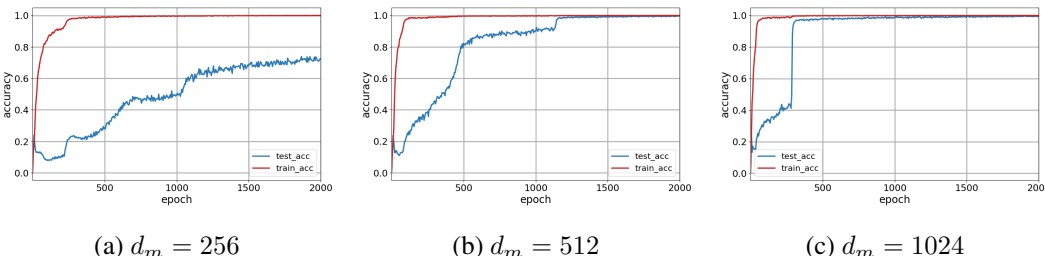

(a) $d_m = 256$        (b) $d_m = 512$        (c) $d_m = 1024$

Figure 9: Training curves of the 3-layer Transformer on 3-step reasoning under different hidden dimensions.

### F.4 UPPER BOUND: EXPERIMENTAL RESULTS OF A 3-LAYER TRANSFORMER ON 4-STEP REASONING

We train a 3-layer Transformer with $d_m = 1024$ on sequences of length 25, using a batch size of 1000 and a learning rate of $5e-5$, to perform 3-step reasoning. Fig. 10 shows our training results.

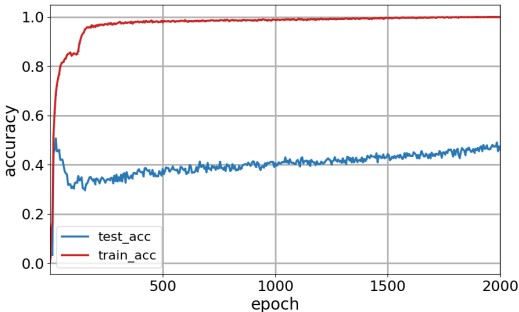

Figure 10: Training curves of the 3-layer Transformer on 4-step reasoning

From Fig. 10, we observe that the test accuracy remains around 35%–40%. In fact, this is fully consistent with our theoretical expectations. Our theory indicates that a Transformer exhibits generalization only on data that satisfy a specific order structure. For clarity, we introduce the following two definitions:

- Canonical sequence: In the 4-step reasoning task, a sentence in which the 3rd fact appears after the 2nd and 4th facts.

- Non-canonical sequence: A sentence that does not satisfy the above condition.

According to our theoretical analysis, the model is expected to correctly resolve canonical sequences, whereas for non-canonical sequences it is unable to accumulate the necessary information and therefore effectively resorts to guessing.

As shown in Fig. 11, we draw out 1000 sequences from canonical sequences and non-canonical sequences to calculate accuracy. And we found that canonical sequences exhibit consistently high accuracy, while non-canonical sequences display substantially lower accuracy, thereby providing empirical support for our theoretical predictions.

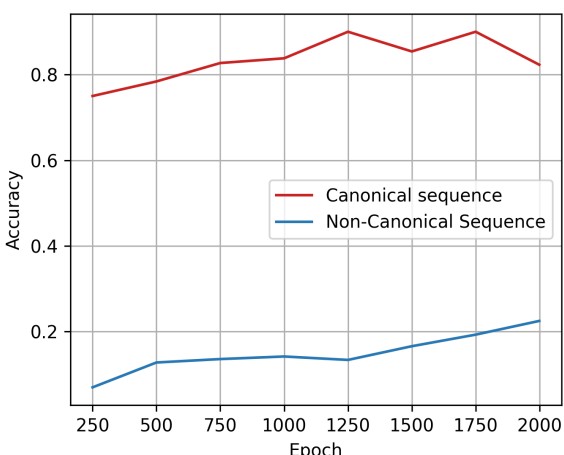

Figure 11: Test accuracy of sentences with different sequence relationships

### F.5    CAUSAL INTERVENTION EXPERIMENT

In this section, we investigate whether transformer is capable of genuine reasoning or merely memorizes the answers, under the settings of 4-step and 5-step reasoning. We then describe the experimental methodology employed to obtain the results.

First, a sequence that can be answered correctly will be selected. Subsequently, a specific attention line or residual connection is masked. If transformer produces an incorrect output after the masking of a particular attention line or residual connection, that line will be marked in blue. If the model's output remains correct, the line will left unchanged. The resulting attention graph retains only those connections that critically influence the outcome. This approach allows for conclusions to be drawn regarding whether the model has learned to perform reasoning.

To further support the validity of our theoretical analysis, we provide two additional contrastive plots (Fig.12). Both plots use inputs containing the same reasoning chain, except that the order of the chain elements is arranged differently. According to our theory, one particular ordering enables the model to aggregate sufficient information to arrive at the correct answer, whereas the alternative ordering prevents the model from collecting enough information to respond correctly. The experimental results align perfectly with these theoretical predictions: when (67, 74) is placed after (74, 42) and (59, 67), the critical attention path propagates exactly along the route predicted by our theory. However, if we change the relative order of these three arrays so that (67, 74) is no longer positioned after (74, 42) and (59, 67), the model's key attention patterns become chaotic and disorganized.

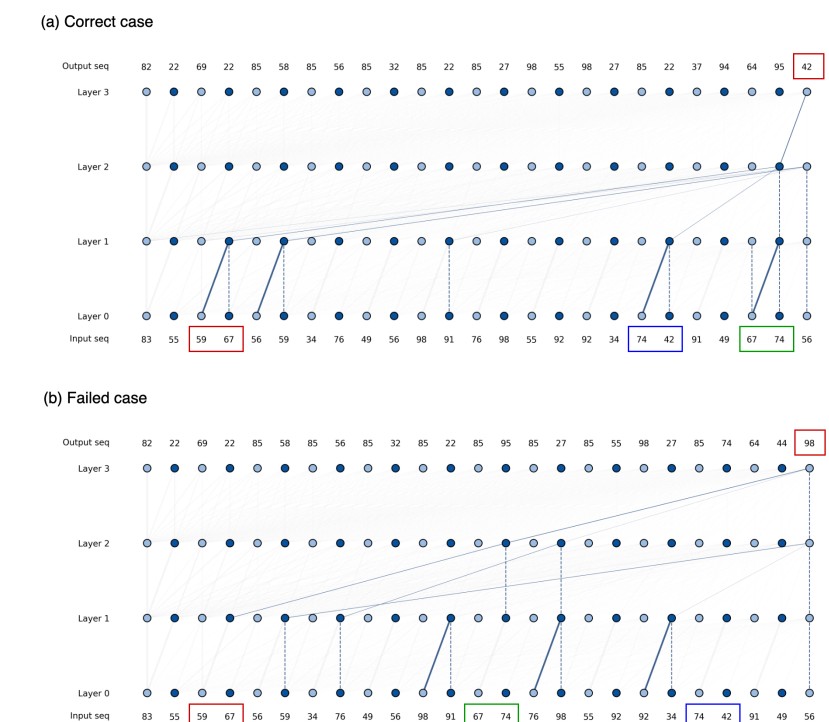

Figure 12: (a) Attention propagation under the valid ordering predicted by our theory. When (67, 74) appears after both (74, 42) and (59, 67), the model is able to consolidate information and the critical attention route follows exactly the theoretically derived path.(b) Attention breakdown under an invalid ordering. If the relative order of the three arrays is altered such that (67, 74) is no longer placed after the other two, the model fails to aggregate sufficient information, resulting in fragmented and chaotic attention patterns

## F.6 EXPERIMENTAL RESULTS OF A 2-LAYER TRANSFORMER

| Table 3: | | |
|---|---|---|
| the number of reasoning steps | 1 | 2 |
| test accuracy(%) | 100 | 8.4 |

The experiment involves training a two-layer, single-headed transformer model with the following hyperparameter configuration: a model dimension $(d_m)$ and key/value dimension $(d_k)$ of 1024, a feedforward dimension $(d_{ffn})$ of 1200, a sequence length of 25, and training for 2000 epochs.

According to Corollary 6.4, a 2-layer transformer can theoretically solve at most one step of reasoning. The empirical results presented in Table 3 support the theoretical predictions: The model achieves complete accuracy on the one-step reasoning test set, which verifies the theoretical prediction for one-step reasoning. And a low test accuracy of $8.4\%$ is observed on the two-step reasoning test set, which indicates that the two-layer transformer's performance is not significantly better than random guessing, thereby validating the theoretical limitation for two-step reasoning.

## F.7 GUESS ABOUT $d_m$

Based on the aforementioned experiments, it can be observed that training a model capable of parallel reasoning—where the number of reasoning steps exceeds the depth of the transformer (i.e., number of layers minus one)—requires a substantially large model dimension $d_m$. It is therefore

hypothesized that for string reasoning, wherein the number of reasoning steps equals the depth of the transformer (layers minus one), a significantly smaller $d_m$ may suffice.

We train a 4-layer Transformer model to perform 3-step reasoning. In this experiment, the sequence length is set to 21, the batch size is 1000, and the learning rate is $5 \times 10^{-5}$. The model is trained for 500 epochs with a hidden dimension of $d_m = 128$.

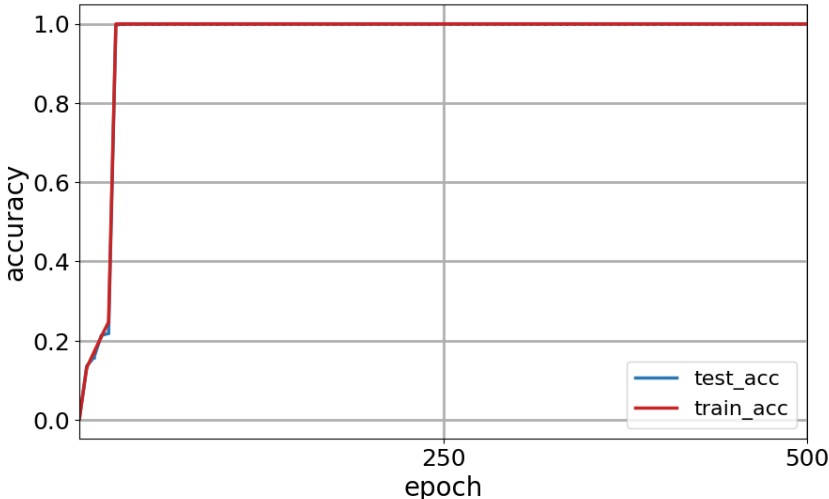

Figure 13: L=4 $d_m = 128$

The experimental results indicate that the blue and red strings both rapidly approach 100% success rates. Under the string reasoning condition, a transformer model with 128 hidden dimensions demonstrates the capability to effectively perform 3-step reasoning tasks (figure 13). In contrast, under the parallel reasoning condition, an architecturally equivalent model with the same number of hidden dimensions achieves only an 11.9% success rate.

## F.8 COMPUTATIONAL RESOURCES

The experiments were conducted on a server with the following configuration:

- 64 AMD EPYC 7742 64-Core Processor.
- 256GB of total system memory.
- 4 A100 GPUs with 40GB of memory.

The task can be completed within 1 week with 1 NVIDIA A100 GPU.

