# OpenReview forum: "Limit Analysis for Symbolic Multi-step Reasoning Tasks with Information Propagation Rules Based on Transformers"
_ICLR.cc/2026/Conference — Submitted to ICLR 2026_

### Official Review · Reviewer_HCxV · 2025-10-19

**Soundness:** 2
**Presentation:** 1
**Contribution:** 1
**Rating:** 2
**Confidence:** 4

**Summary:**

This work studies the question: "how many 'reasoning steps' can an $L$-layer Transformer carry out in a single forward pass?". To answer this question, the paper posits a formulation of "reasoning chains" as sequences of pairs of integers $a_i^1 \to a_i^2$, which can be permuted. It then formalizes an abstract model intended to represent multi-layer Transformers by defining rules of "information propagation", where the information at each token in the transformer is a set $V_i^l$ that expands at each layer (i.e., $V_i^{l+1} = V_{\mathcal{I}}^{l} \cup V_i^{l}$). The question of how many reasoning steps a transformer can carry out is mapped to the question of how big $C_i^l = | V_i^l|$ can be. The main result of the paper is to show that this is between $2^{l-1}$ and $3^{l-1}$.

**Strengths:**

This work tackles an important problem that is of great interest to the machine learning community, namely, characterizing the theoretical capabilities of Transformers to perform "reasoning". A key aspect that is highlighted is the potential to perform parallel computation across all tokens at each layer.

**Weaknesses:**

- The main weakness of this work lies in the modeling of its notions of "reasoning" and "Transformers", which underpins the main theoretical result. The framework is presented in a highly abstract (and arguably unnecessarily so) manner, lacking clear motivation or grounding in real tasks or implementations of Transformers. Formalizing vague concepts like “reasoning” is inherently difficult, but the formulation proposed here is neither clearly useful nor evidently interesting. It is unclear what class of problems this artificial setup captures.
- Moreover, the formulation of Transformer computation is also limited and artificial. On the one hand, it considers only a very limited subset of the types of computations Transformers can do (namely, "same token matching" and "adjacent position matching"). On the other hand, even those limited computations are modeled in an overly simplistic manner, which is disconnected from the parameterization of Transformers and the distributed vector representations at each layer.
- The central result is unsurprising given the artificial nature of the model. In particular, the toy model of Transformer computation crucially does not consider the capacity of finite-precision vector embeddings, which has been shown to be critical in determining the function class of neural networks, including Transformers, in prior work.
- The experimental section is limited and does not substantively test the theoretical results. While the experiments are designed around the paper’s own notion of “reasoning,” they only consider $L=3$ and yet claim empirical support for the theory. To meaningfully support the theoretical result, the authors would need to vary $L$ and demonstrate that the conclusions hold more generally. Since the theory concerns scaling with depth, fixing the depth in experiments fails to test the key claim.
- The Related Work section lacks substantive discussion. It merely lists a large number of papers with minimal analysis. In particular, relevant work on the approximation / representation capacity of Transformers is missing (e.g., Perez et al. 2019; Yun et al. 2020; Wei et al. 2022; Sanford et al. 2023; etc.).
- Communication and structure are also weak points. The logical flow of the paper is difficult to follow. For instance, Section 3 presents an informal preview of the theoretical result before introducing the necessary definitions of “reasoning” or “information propagation”. The mathematical formalism is often redundant or unnecessarily complicated. For example, Definitions 4.2 and 4.4 are redundant, Definitions 4.6 and 4.8 are redundant, and Definition 4.5 appears unnecessary. Overall, much of the abstraction overcomplicates otherwise simple concepts.

**Questions:**

- Does this theory make any predictions about how the number of “reasoning steps” computable by a Transformer scales with the number of heads, in addition to the number of layers?
- How does the proposed model of “information propagation” relate to existing formal models of Transformer computation, such as those in RASP?
- In Rule 2 of the information propagation rules, why is it stipulated that “adjacent position matching” information must flow from odd-position nodes to subsequent even-position nodes? This appears to be an artificial constraint intended to align the model with the pair structure of the “reasoning chains.” How would such a rule correspond to realistic models of position-based attention?

Typos?
- line 200: $a_i^1 \to a_I^2$
- Line 285: $(\mathbf{a} a_m)_{1 \leq m \leq s}$

---

> ### Author Response · Authors · 2025-11-20
>
> We sincerely thank the reviewer for the careful reading of our manuscript and for the valuable comments and suggestions. We deeply appreciate the time and effort invested in evaluating our work. Below, we provide detailed responses to each of the reviewer’s questions and concerns.
>
> **[w1][Clarifications on Theoretical Analysis and Motivation]**
>
> (1)To provide a clearer exposition of the research scope and motivation underlying this work, we have significantly enriched the Introduction in the updated manuscript.
>
> (2)Our theoretical results may be able to reflect an interesting phenomenon: in multi-step reasoning tasks, sequentially providing reasoning conditions may instead achieve only the lower bound of information capacity, i.e., O($2^L$). This phenomenon may reflect counter-intuitive behavior of causal-mask autoregressive models when processing text. This potential phenomenon is indeed very interesting, but because the internal behavior of real language models is highly complex, conducting reliable empirical analysis requires more rigorous experimental settings and controls. Therefore, for now we have chosen not to include this exploratory conjecture in the paper. We plan to carry out dedicated experiments on this direction in future work to obtain more robust conclusions and facilitate the early landing of the theoretical work.
>
> (3)Abstract modeling of real-world problems is a common approach in theoretical research. For example, papers [1,2] study the reasoning mechanisms of Transformers using symbolic datasets that simulate reasoning, and propose algorithms to improve reasoning abilities.
>
> [1] Boix-Adsera E, et al. When can transformers reason with abstract symbols? ICLR 2024.
>
> [2] Wang Z, et al. Understanding the LLM to Solve the Symbolic Multi-Step Reasoning … EMNLP 2025.
>
> **[w2][The modeling of attention]**
>
> The essence of Transformer attention is to search for semantic similarity under linear transformations. Here, “semantics” includes two types: one is the concrete semantic information provided by the token itself, and the other is the positional information provided by the positional encoding. In general, concrete semantic information and positional information are decoupled. Therefore, our consideration of “same token matching” and “adjacent position matching” does not simplify the expressive power of attention. Linear transformations can extract a wide variety of information.
>
> **[w3][Model capacity] & [w4][Additional experimental validation]**
>
> Due to space limitations in the response, and because these two issues were raised by multiple reviewers, we addressed them in the **official comment** as common concerns.
>
> **[w5][Additional related work]**
> We refined our discussion of the references and added the related work you mentioned regarding the approximation/representational power of Transformers.
>
> **[w6][Improving the manuscript presentation]**
>
> Following your suggestions, we restructured the manuscript and removed some redundant definitions. Specifically, we now use a single paragraph to cover Definitions 4.4, 4.5, and 4.8 from the original manuscript, and we deleted these three definitions. (In the new manuscript, we use strikethrough to indicate deleted content.)
>
> However, considering that our theoretical study requires lengthy preliminary definitions and complex statements, we still believe that informal exposition is necessary. It helps readers quickly grasp the core conclusions of the paper.
>
> **[Q1][Effect of the number of attention heads on reasoning ability]**
>
> Although intuitively multi-head attention allows different heads to focus on different semantic relations, the upper bound of reasoning ability does not increase with the number of heads. In fact, since the three reasoning rules we define do not impose any restrictions on the amount of attention each node can pay, in other words, what we consider is essentially a Transformer with infinitely many heads.
>
> **[Q2][Discussion of RASP]**
>
> Thank you for the reminder. We will include a discussion of RASP in the related work. In fact, RASP essentially uses the attention structure to express finite-state machines and algorithmic steps, which is very similar to our information-propagation rules. However, the difference is that we further consider high-fidelity information extraction. To achieve this, we need to rely on the nonlinear capability of the FFN to perform subspace transformations(Appendix E).
>
> **[Q3][Information propagation rules]**
>
> Implementing multi-step reasoning requires completing (1) identification of all single-step reasoning relations and (2) multi-step information transmission. Our restriction of attention from odd positions to even positions is essentially simulating the process by which the Transformer identifies all single-step reasoning relations, and does not affect the core conclusion. Real Transformers accomplish this process through more complex patterns.
>
> **[Q4][Typos]** Corrected. Thanks for your suggestion.

---

> > ### Comment · Reviewer_HCxV · 2025-11-26
> >
> > I thank the authors for their response and for the revisions. I have reviewed both the rebuttal and the updated manuscript.
> >
> > ---
> >
> > Overall, my main concerns remain unresolved. Below is a point-by-point reply.
> >
> > [w1] Clarifications on theoretical analysis and motivation
> >
> > The primary issue is the idealized computational model used to represent Transformers, referred to as “Information Propagation Rules.” This model is effectively a program with infinite memory. It does not meaningfully approximate real Transformer networks. In particular, it assumes infinite capacity, which does not hold, and the claim that Transformers can perform an exponential number of “reasoning steps” applies only to this program (Algorithm 1 in the revised manuscript), not to actual Transformer architectures.
> >
> > > Abstract modeling of real-world problems is a common approach in theoretical research. For example, papers [1,2] study the reasoning mechanisms of Transformers using symbolic datasets that simulate reasoning, and propose algorithms to improve reasoning abilities.
> >
> > To be clear, the key concern is not merely the use of synthetic tasks. It is that the computational model of “Transformers” is so idealized that it produces unsound conclusions, including infinite capacity and exponential “reasoning steps.” The particular synthetic task used also lacks strong motivation, and the claim that it represents a general notion of reasoning steps is overstated, but this is secondary to the core issue of soundness.
> >
> > [w2] Modeling of attention
> >
> > > Therefore, our consideration of “same token matching” and “adjacent position matching” does not simplify the expressive power of attention. Linear transformations can extract a wide variety of information.
> >
> > This is incorrect. Transformers can represent a much broader class of computations. Line 112 asserts that it is sufficient for the weight matrices to satisfy (W_{qk} W_{vo}^T = I), implying no expressivity loss. This is not right. For example, attention mechanisms can be asymmetric: queries and keys can rely on different attributes, and this asymmetry is essential even for basic behaviors such as induction heads. Your model does not capture this.
> >
> > Moreover, even the limited computations your model does consider are represented in a way that is overly simplified and disconnected from the actual parameterization of Transformers and their distributed vector representations (see w1 and w3).
> >
> > [w3] Model capacity
> >
> > > We note that the reviewers mentioned concerns about the practical capacity of the model. First, this study focuses on the mechanism by which a Transformer performs parallel multi-step reasoning, as well as the maximum number of reasoning steps that a Transformer can achieve under this mechanism. To simplify this problem, we consider precise information-matching rules while ignoring the limitations imposed by model capacity.
> >
> > This simplification renders the “Information Propagation Rules” model too idealized to capture real Transformers. I suspect this is why the simulations fail to match the theoretical predictions.
> >
> > > Second, in Appendix D of the original manuscript, we construct a Transformer architecture that can match our information propagation rules and point out that, under exact matching, the required hidden-space dimension is on the order of (O(n \cdot 3^L \cdot d)), where (n) is the sentence length, (L) is the number of layers, and (d) is the vocabulary size.
> >
> > This construction is trivial, and the bound is so large as to be vacuous. It scales exponentially with the number of layers and with sequence length. This highlights the central limitation: if the theory requires model dimension to scale exponentially to hold, then its scope and applicability are extremely limited.
> >
> > ---
> >
> > [w4] Additional experimental validation
> >
> > > We note that multiple reviewers expressed concern that the experiment with (L = 3) shown in our paper is insufficient to support our conclusions. Considering that for (L > 3) the computational cost and optimization difficulty grow exponentially with (L), we instead use a simulation program that models information propagation to justify our conclusions.
> >
> > The claim about computational cost is unclear. Training Transformers with (L > 3) layers on a small synthetic task requires only modest resources and could likely be done on free cloud platforms. The models are very small.
> >
> > Moreover, the simulation program does not support the paper’s conclusions because it implements only the idealized “Information Propagation Rules,” not the behavior or complexity of real parameterized Transformers.
> >
> > ---
> >
> > Since these concerns remain unaddressed, I will maintain my current score and recommendation for rejection. I encourage the authors to continue developing this line of work, with particular focus on constructing a more accurate model of Transformer networks grounded in their actual parameterization rather than an idealized program with infinite memory.

---

> > > ### Author Response · Authors · 2025-11-28
> > >
> > > We greatly appreciate that you are willing to spend time engaging in further in-depth communication with us. We are very happy to clarify main issues you are concerned about.
> > >
> > > We notice that your concerns mostly come from worrying our theoretical assumptions do not match actual experiments. But in fact, our theoretical assumptions are all derived from observations of existing studies [1,2,3] and our experiments. When observing how Transformers propagate information, we, also previous studies, found that Transformers use methods such as same token matching and adjacent position matching; therefore, we formalize the information propagation rules as the the start point of our theory.
> > >
> > > In addition, our experiments under the case of L=3 are well consistent with our theoretical results. This is reflected in the following aspects:
> > >
> > > (1) For the lower bound of the number of reasoning steps, i.e., s=3, theoretically three steps are possible, and experimentally the model can generalize to 100%. (Fig9)
> > >
> > > (2) A very critical validation is as follows. For the upper bound of the number of reasoning steps, i.e., s=4 for L=3, we theoretically prove that Transformers have strong generalization ability only on data that satisfies a specific order relation; the proportion of sequences that satisfy this type of order relation in the whole dataset is about 33%. The training results in Fig10 show that in the early stage of training, the generalization accuracy of the model is indeed around 33%. **We further supplemented additional analysis of these experimental results (Fig11、12)**. We found: when sequences follow the order relation we expect, the Transformer can reach an accuracy of over 85% on this type of sequences (Fig.11), and the information propagation path matches our theoretical results (Fig.12, i.e., the 1st attn execution position matching, and subsequent attn layers perform same toekn matching). However, when sequences do not satisfy the order relation we expect, the Transformer performs very poorly on this type of dataset. This indicates that Transformers indeed depends on order for solving reasoning problems, and such order dependence can be predicted by our theory.
> > >
> > > (3) The reviewer argues the infinite capcity assumpt may be ideal. However, our experients indicate a reasonable hidden dimension can solve the problem well. In addition, the performance can be improved as the hidden dimension dm increases. (Fig. 9)
> > >
> > > [1] In-Context Learning and Induction Heads. Transformer Circuits, Anthropic, 2022.
> > >
> > > [2] Understanding the LLM to Solve Symbolic Multi-Step Reasoning … EMNLP 2025.
> > >
> > > [3] When can transformers reason with abstract symbols? In ICLR 2024.
> > >
> > > Below are our point-by-point replies to the other issues you mentioned:
> > >
> > > **[w3][Model capacity]**
> > >
> > > We hope to treat “**Inherent Expressive Power**” and “**Dimension-Constrained Capacity**” as two independent topics, and our research focuses on the former.
> > >
> > > The contribution of our paper is to prove that, under ideal conditions, the Transformer model has exponential reasoning capability, which is not a trivial conclusion. For example, consider an infinitely wide multi-layer fully connected model: it is completely unable to perform multi-step reasoning on unseen knowledge. Therefore, our research provides a new understanding of the capability boundaries of Transformers from the perspective of approximation theory.
> > >
> > > The interesting thing is that our estimation is well consistent with the results for three-layer network as we mentioned previously. The reviewer wonder if we need to assume infinite model capacity. Note that the upper bound of the dimension estimate we provide is calculated under the situation where each reasoning step is an exact match; this is merely a loose assumption for the convenience of carrying out theoretical derivations. But in fact, in attention mechanisms with softmax, only fuzzy matching is needed to implement it. For example, for x=[0.4,0.3,0.3], softmax(100x)$\approx$ [1,0,0]. Therefore, under fuzzy matching, only a log-scale version of the original upper bound is needed [4,5]. In addition, in the experiments shown in Fig9, although from the perspective of idealized assumptions we need at least a 40000-dimensional hidden space, due to the existence of fuzzy matching, a hidden dimension of 512 is sufficient for the model to generalize. Therefore, our theoretical result is closely related with practical training.
> > >
> > > [4] Near-Optimal Signal Recovery From Random Projections.
> > >
> > > [5] Extensions of Lipschitz mappings into a Hilbert space.

---

> > > > ### Author Response · Authors · 2025-11-28
> > > >
> > > > **[w2][Modeling of attention]**
> > > >
> > > > As mentioned above, our theoretical assumptions are all exactly observed in experiments of existing studies [1,2,3] and our own experiments. In a generalized sense, same token matching and adjacent position matching are among the underlying meta-operations of the attention mechanism. Attention mechanism has many other meta-operations, but experiments indicate that these two operations are exactly the functions performed by the attention mechanism when carrying out this type of reasoning task.
> > > >
> > > > *Remark: the **same token matching** we mention here does not mean that matching can only happen between two strictly identical tokens. For example, "father", "dad" are semantically similar (close in embedding space but not exactly the same), so they can be regarded as the same token through a simple linear transformation (i.e., Wq or Wk).*
> > > >
> > > > The induction head you mentioned belongs to a special case of the multi-step reasoning we study, namely single-step reasoning, which can be fully completed by these two matching operations [1,2,6].
> > > >
> > > > [6] Anchor function: a type of benchmark functions for studying language models
> > > >
> > > > **[w1] Clarifications on theoretical analysis and motivation**
> > > >
> > > > The issue about model capacity has already been answered in [w4]. The answers to the remaining questions are as follows:
> > > >
> > > > > The particular synthetic task used also lacks strong motivation, and the claim that it represents a general notion of reasoning steps is overstated
> > > >
> > > > This type of symbolic task can represent a large class of multi-step reasoning tasks, especially for math reasoning tasks. This type of symbolic task allows us to study the mechanism of multi-round info transmission in Transformers. To avoid overclaim, we added the following content:
> > > >
> > > > > (new added) In order to more intuitively understand the mechanism of multi-round information transmission in Transformer models and the boundary of information transmission capacity, we use a symbolic dataset that can simulate a class of multi-step reasoning to conduct our study.
> > > >
> > > > > The primary issue is the formalized a computational model used to faithfully represent Transformers.
> > > >
> > > > According to the further explanation of attn in [w2], we believe that in a generalized sense, same token and adjacent pos matching are the meta-operations used by the attn mechanism that we observed in experiments when Transformers perform this type reasoning tasks. The purpose of the Info Propagation Rules we established based on this is actually similar to RASP and induction head: it aims to provide a clean and abstract theoretical tool that can reflect the Transformer’s Inherent Expressive Power.
> > > >
> > > > **[w4] Additional experimental validation**
> > > >
> > > > We supplemented the following content in the appendix to demonstrate the computational resources:
> > > >
> > > > > The experiments were conducted on a server with the following configuration:
> > > > > - 64 AMD EPYC 7742 64-Core Processor.
> > > > > - 256GB of total system memory.
> > > > > - 4 A100 GPUs with 40GB of memory.
> > > > > - The task can be completed within 1 week with 1 NVIDIA A100 GPU.
> > > >
> > > > Based on our accumulated experience, for an s-step reasoning task, the data length should be at least 4(s+1)+1. For L=3, when we do 4-step reasoning, the data length is 21; but to verify the case of L=4 (s=13), we need at least length 57, and the data volume and training difficulty also increase dramatically compared to 4-step reasoning. Therefore, under the conditions allowed by our resources, we have already thought as much as possible about how to present experimental results.
> > > >
> > > > > it implements only the idealized“Info Propagation Rules,”not the behavior or complexity of real parameterized Transformers.
> > > >
> > > > To have a rigorous understanding of a complex network like Transformer in a specific tasks, it requires a good model of the mechanism of how it solve such tasks. In this paper, we use the info propagation rules observed in real experiments to build a such model, which allows us to give an accurate and meaningful estimate for the reasoning steps. For example, we can predict how information order can affect the results:
> > > >
> > > > (1) From Fig2, we find that in multi-step reasoning tasks, sequentially providing reasoning conditions may instead achieve only the lower bound of information capacity, i.e., O(2^L). This may reflect counter-intuitive behavior of causal-mask autoregressive models when processing text.
> > > >
> > > > (2) From Fig13, we find that by appropriately reordering the logical relations in the data, it is possible to significantly improve the model’s generalization ability.**
> > > >
> > > > Finally, we again thank the reviewer for spending substantial time carefully reviewing this paper and providing many specific and constructive comments. This work, based on the observation from experiments, renders interesting feedback for experiments. We appreciate for the encouragement given by the reviewer via such detailed discussion in our starting point of understanding the reasoning ability of Transformers.

---

### Official Review · Reviewer_qBEY · 2025-10-23

**Soundness:** 3
**Presentation:** 3
**Contribution:** 4
**Rating:** 8
**Confidence:** 3

**Summary:**

This paper studies the intrinsic single-pass reasoning depth of transformer decoders with L attention layers. Within a symbolic multi-step reasoning setup, the authors formalize “information propagation rules” (adjacent-position matching, same-token matching, residuals) and prove bounds on the maximum number of effective reasoning steps in one forward pass lies between. They give matching constructions showing a binary-tree lower bound and a ternary-tree upper bound, and provide experiments (e.g., a 3-layer model achieving perfect 3-step reasoning with sufficiently large hidden dimension) consistent with the theory.

**Strengths:**

A clear, formal definition of “reasoning step” tied to concrete token-level propagation primitives (adjacent and same-token matching). Theorems bound capacity as a function of depth.

The bounds are constructive and interpretable.

Empirical checks (3-layer transformer; accuracy vs. hidden size) align with the theoretical picture that wider representations help store intermediate facts but do not break the depth-dependent step limit.

**Weaknesses:**

It’s not yet clear (at least to me) how to map the notion of the reasoning step from the paper to what counts as a step in naturalistic prompting in reasoning experiments with LMs. For instance, Guzmán et al. (NAACL Findings 2024) document persistent failures of the transformer's logical generalization (recursivity and compositionality). Could such phenomena be explained through the main result of the paper?

**Questions:**

Do your bounds predict the specific deficits seen in compositional and recursive reasoning in transformers?

How would the bounds change without the single pass assumption?

---

> ### Author Response · Authors · 2025-11-20
>
> We sincerely thank the reviewer for the thorough evaluation of our work and the very positive overall assessment. The reviewer’s insightful comments are highly valuable for further improving the quality of the paper.
>
> **[w1][Connection with the Vargas paper]**
>
> The article considers only a single-layer Transformer. Our work points out that Transformers need to attend to different types of information at different layers in order to form multi-step logical structures. A single-layer Transformer is insufficient to capture these logical relations. We have added a citation to this paper in the related work section.
>
> > Vargas Guzmán et al. (2024) find that Transformers show limited generalization in recursivity and compositionality. Their experiments, however, use only a one-layer Transformer, which differs fundamentally from the multi-layer architectures analyzed in our work.
>
> **[Q1][Predicting specific limitations of Transformers in compositional and recursive reasoning]**
>
> This is an excellent question. Our theoretical results may be able to reflect an interesting phenomenon: in multi-step reasoning tasks, sequentially providing reasoning conditions may instead achieve only the lower bound of information capacity, i.e., O($2^L$). This may reflect counter-intuitive behavior of causal-mask autoregressive models when processing text.
>
> This potential phenomenon is very interesting; however, the internal behavior of real language models is highly complex, and reliable empirical analysis requires more rigorous experimental settings and controls. Therefore, for now we have chosen not to include this exploratory conjecture in the paper. We plan to conduct dedicated experiments on this direction in future work to obtain more robust conclusions and facilitate the early realization of the theoretical findings.
>
> **[Q2][CoT]**
>
> In previous related work, there has been discussion on the setting where the model is allowed to use Chain of Thought (CoT) reasoning [1]. That study shows that when CoT is allowed, a Transformer with only two layers can theoretically perform an unbounded number of reasoning steps. The key reason is that without CoT, different pieces of information must be stored in different subspaces, and thus the amount of information is limited by the size of these subspaces. Under the CoT mode, however, new information overwrites the information in the original subspace, allowing reasoning to evolve indefinitely with only a small number of subspaces.
>
> [1] Wang, Z. et al. (2025). Understanding the language model to solve the symbolic multi-step reasoning problem from the perspective of buffer mechanism. EMNLP 2025.

---

> > ### Comment · Reviewer_qBEY · 2025-11-25
> >
> > Thank you for your answers and for the paper.

---

### Official Review · Reviewer_sWgX · 2025-10-27

**Soundness:** 4
**Presentation:** 3
**Contribution:** 4
**Rating:** 8
**Confidence:** 2

**Summary:**

This paper provides a theoretical analysis of the symbolic multi-step reasoning limits of a Transformer in a single forward pass. By modeling information propagation through adjacent position and same-token matching, the authors show that an $L$-layer Transformer can support between $O(2^{L-1})$ and $O(3^{L-1})$ reasoning steps. Experiments on symbolic reasoning tasks confirm these bounds, showing that a 3-layer model succeeds on up to 4-step reasoning but fails beyond that. The work offers a formal and tight characterization of single-pass reasoning capacity, extending beyond prior empirical analyses.

**Strengths:**

This paper makes a strong contribution by providing a clear and rigorous theoretical understanding of the intrinsic reasoning capacity of Transformer models.

More specifically, it establishes formal upper and lower bounds on the number of reasoning steps a Transformer can perform in a single forward pass, moving beyond purely empirical observations. The information propagation framework is well structured, connecting low-level attention mechanisms to high-level symbolic reasoning, which enhances interpretability. The proof techniques are clean and logically sound, offering precise insights into how reasoning depth scales with model depth. The experimental results closely match the theoretical analysis, validating the tightness of the proposed bounds. Overall, the work addresses a fundamental and timely problem, making it both theoretically significant and practically relevant.

**Weaknesses:**

While the paper makes a solid theoretical contribution, it also has some limitations. The analysis is based on highly idealized symbolic reasoning tasks, which may not fully capture the complexity of real-world natural language reasoning scenarios. However, this might be understandable given the difficulty of rigorous theoretical analysis.

In addition, although the theoretical bounds are tight, the paper offers limited discussion of how these limits might interact with optimization dynamics, initialization, and data distribution, and a deeper treatment of these aspects would make the results more practically relevant.

**Questions:**

How do you expect the theoretical bounds derived under idealized symbolic settings to translate to more realistic scenarios, where the reasoning process involves natural language, semantic ambiguity, and potentially noisy or unstructured inputs?

---

> ### Author Response · Authors · 2025-11-20
>
> We sincerely thank the reviewer’s careful assessment of our paper and the strong overall evaluation. The reviewer’s thoughtful suggestions are extremely helpful for further refining the manuscript.
>
> **[w1][The assumptions are overly idealized]**
>
> Thank you for your understanding. Abstracting and simplifying the problem is a common methodology in theoretical research. We use symbolic datasets to simulate multi-step reasoning tasks while preserving the core characteristics of reasoning. Such simplified datasets enable deeper theoretical derivations.
>
> **[w2 & Q1][Potential practical value]**
>
> This is a good question. Our theoretical results may be able to reflect an interesting phenomenon: in multi-step reasoning tasks, sequentially providing reasoning conditions may instead achieve only the lower bound of information capacity, i.e., O($2^L$). This may reflect counter-intuitive behavior of causal-mask autoregressive models when processing text.
>
> This potential phenomenon is very interesting; however, the internal behavior of real language models is highly complex, and reliable empirical analysis requires more rigorous experimental settings and controls. Therefore, for now we have chosen not to include this exploratory conjecture in the paper. We plan to conduct dedicated experiments on this direction in future work to obtain more robust conclusions and facilitate the early realization of the theoretical findings.

---

> > ### Comment · Reviewer_sWgX · 2025-11-26
> >
> > The author response has addressed most of my concerns. I will retain my score recommending acceptance.

---

### Official Review · Reviewer_owtV · 2025-11-01

**Soundness:** 2
**Presentation:** 2
**Contribution:** 2
**Rating:** 2
**Confidence:** 3

**Summary:**

The paper studies the intrinsic, single-pass reasoning capacity of decoder-only transformers by introducing an idealized information-propagation model that abstracts attention and residual connections into set-based rules over “reasoning pairs.” The core task is symbolic: inputs are made of shuffled pairs (a_i^1, a_i^2) that form a reasoning chain, and the model must recover the target reached after multiple reasoning steps. Within this abstract model, the authors prove that for an L-layer system following their rules, the number of reasoning steps whose information can reach the final position is between a lower bound of 2^{L-1}-1 and an upper bound of \frac{3^{L-1}-1}{2}, i.e. exponential in depth with bases 2 and 3 respectively. These bounds are made concrete by constructing special input permutations that attain the two extremes. They further provide a constructive mapping from the abstract rules to a hand-designed single-head masked transformer that realizes adjacent-position matching and same-token matching with specific Q/K/V choices, arguing this shows the rules are implementable in principle. Experiments on synthetic data with a 3-layer transformer show: (i) near-perfect 3-step reasoning when width is large, (ii) partial success on 4-step when the ordering is favorable, and (iii) clear degradation on 5-step, which the authors present as consistent with their theoretical limit. However, the experiments do not test the theory’s central scaling claim in L, and the symbolic abstraction is stronger than what real, trained models are guaranteed to implement.

**Strengths:**

- The authors provide both a constructive lower bound and a constructive upper bound.
- Appendix D gives an explicit way to choose Q/K/V and use residuals/FFNs so that a single-head, causal Transformer can behave like the rule system.

**Weaknesses:**

- The theorems (Sec. 6) are proven for a set-union model of information, not for the actual vector-space, soft-attention transformer. It does not show that trained transformers implement those rules. The proofs are also dimensionality-agnostic, implicitly assuming each node can store arbitrarily many distinct reasoning items regardless of hidden size. However, the authors’ own experiments (Table 1) show that performance depends strongly on hidden size (larger widths are required for multi-step reasoning) indicating that representational capacity is a limiting factor.
- The empirical results insufficiently probe the core scaling law: the theory predicts reasoning capacity should grow exponentially with depth, yet all reported experiments evaluate only one depth (L=3).
- The assumption that adjacent position matching occurs only in layer 1 and same-token matching only in layers L >= 2 (Rule 2 & 3) is entirely artificial. There is no architectural or theoretical reason for this strict layering of function.

**Questions:**

- Can you add experiments for L=2,4,5 on the same symbolic task, with width sweeps, to show whether the “steps vs. depth” curve is actually exponential in practice?
- Your own results show strong dependence on hidden size. Can you give a dimensionality or capacity argument along the lines of “to realise N parallel reasoning items at layer L, you need at least O(N) (or similar) width”?

---

> ### Author Response · Authors · 2025-11-20
>
> We sincerely appreciate the reviewer’s careful assessment and insightful recommendations, which substantially helped us improve the presentation and rigor of the paper. Below, we respond to each comment in detail.
>
> **[w1 & Q2][Model capacity estimation]**
>
> (1)This study focuses on the mechanism by which a Transformer performs parallel multi-step reasoning, as well as the maximum number of reasoning steps that a Transformer can achieve under this mechanism. To simplify this problem, we consider precise information-matching rules while ignoring the limitations imposed by model capacity.
>
> (2)In Appendix D of the original manuscript, we construct a Transformer architecture that can match our information propagation rules, and we point out that, under exact matching, the required hidden space dimension is on the order of O($n3^Ld$), where n is the sentence length, L is the number of layers, and d is the vocabulary size. However, the constructed vectors are extremely sparse, and we do not need the exact matching required by the construction—approximate matching is sufficient (because layernorm and softmax can amplify small differences). According to compressed sensing theory and the Johnson–Lindenstrauss lemma, the intrinsic dimension we actually need can be far smaller than the dimensionality of the current construction. We will provide a rigorous proof in future work.
>
> **[w2 & Q1][Additional experimental validation]**
>
> Considering that for L>3 the computational cost and optimization difficulty grow exponentially with L, we instead use a simulation program that models information propagation to support our conclusions. For the case L = 3, we enumerated all possible serialized representations of the reasoning chain a1 → a2 → … → a11, amounting to 11! = 3,628,800 possibilities. Across all possibilities, the minimum information content of the final node is 5 and the maximum is 10, consistent with the theoretical lower and upper bounds. The relevant experiments can be found in Section 7.
>
> In addition, we supplement the study with experiments on a 2-layer Transformer. The single-step reasoning test accuracy reaches 100%, while the two-step reasoning accuracy is 8.4% (indicating failure). This aligns with our theoretical results. The corresponding results can be found in Appendix F.
>
> **[w3][Clarifying concerns about the assumptions]**
>
> (1)We make this assumption solely for the convenience of theoretical derivation, and it does not affect the core conclusions. If value matching occurs in the first layer, then in subsequent layers only positional matching can increase information, because in later layers the only possible value match for a node can come from the residual. A simple example: for the reasoning sequence (a, b, c, d, e, f) and its reordered sequence (a, b, e, f, b, c, d, e, c, d) with corresponding positions (1, 2, 3, 4, 5, 6, 7, 8, 9, 10), if value matching occurs in the first layer, then in the second layer the nodes take the following forms: ({a}, {1}), ({b}, {2,5}), ({c}, {6,9}), ({d}, {7,10}), ({e}, {3,8}), ({f}, {4}), along with some nodes from the residual. It is clear that these nodes can no longer undergo value matching, and value matching from residual nodes will not increase the amount of information.
>
> (2)According to the experimental results mentioned above, under the condition that position and value matching may occur at any layer, the maximum reasoning capacity that a 3-layer model can achieve is 10, which is consistent with our theoretical upper bound.

---

> > ### Comment · Reviewer_owtV · 2025-11-26
> >
> > Thank you for your response. It has addressed most of my main concerns. While the main result (exponentially scalable reasoning ability) matches intuitions from prior results (e.g. [1]) and are thus not surprising, I think the formal bounds are interesting and potentially useful. I still think the paper would benefit from empirically validating their identified bounds on wider range of problems and model depths. Based on that, I raise my score to 4.
> >
> > [1] C. Sanford, D. Hsu, and M. Telgarsky, ‘Transformers, parallel computation, and logarithmic depth’, in Proceedings of the 41st International Conference on Machine Learning, 2024.

---

> > > ### Author Response · Authors · 2025-11-28
> > >
> > > We sincerely appreciate the reviewer’s further feedback and the decision to raise the score, and we also thank you again for pointing out the value of the related work.
> > >
> > > Regarding the work you mentioned, Transformers, parallel computation, and logarithmic depth (Sanford et al., 2024), we fully agree with its importance. That paper theoretically proves that a Transformer with depth O(log L) can realize L-step reasoning, thereby providing a lower bound on the number of layers required to complete a given number of reasoning steps.
> > >
> > > In contrast, our research also focuses on another “reverse” question:
> > >
> > > **Given a fixed depth, how many reasoning steps can a model complete at most? Do there exist theoretical upper and lower bounds for this achievable number of reasoning steps?**
> > >
> > > Our main contributions are:
> > >
> > > 1. We propose and prove that the number of reasoning steps achievable by a fixed-depth Transformer has clear upper and lower bounds(i.e. $2^{L-1}$ and $3^{L-1}$), which has not been systematically discussed in the existing literature.
> > >
> > > 2. **A very critical and interesting phenomenon we discovered in the derivation is that the ordering of reasoning pairs in the input sequence affects the model’s performance, even when they correspond to the same reasoning chain**. For example, for the reasoning chain a→b→c→d→e, we find that when the input pattern is a→b, b→c, d→e, c→d, a 3-layer model can correctly predict that the 4-step reasoning result of a is e. However, when the input pattern is a→b, b→c, c→d, d→e, the model produces an incorrect result. This counter-intuitive phenomenon has not been discussed in prior literature. It further shows that the boundary of the model’s capability is not only determined by the model’s own capacity, but also has a deep relationship with the symbolic order structure in the input.
> > >
> > > For Contribution 2, we also verify the theoretical results through experiments:
> > >
> > > -  For the lower bound of reasoning steps, i.e., s = 3, the model can theoretically perform three steps, and empirically its test accuracy generalizes to 100% (Fig. 10).
> > >
> > > -  For the upper bound of reasoning steps, i.e., s = 4, we theoretically prove that the Transformer has strong generalization ability only on data satisfying specific order relations. The proportion of such sequences in the entire dataset is about 33%, and Fig. 10 shows that in the early stage of training, the model’s generalization accuracy is indeed around 33%. We further provide additional analysis of the experimental results (Fig. 11, 12). We find that: when the sequences obey our expected order relations, the Transformer can achieve over 85% accuracy on such sequences (Fig. 11), and the information-propagation paths match our theoretical results (Fig. 12, i.e., the first attention layer performs position matching, and subsequent attention layers perform same-token matching). However, when sequences do not satisfy our expected order relations, the Transformer performs poorly on this part of the dataset. This shows that the Transformer indeed develops order preferences during training, and these order preferences can be predicted by our theory.
> > >
> > > In the original manuscript, we already cited Sanford et al. (2024):
> > >
> > > > Our paper's difference from Sanford et al. (2024) is that we consider the transformer with mask and FNN whereas they ignore mask and FNN, which indicates that our framework operates under assumptions that align more directly with practical Transformers.
> > >
> > > To more clearly discuss the essential differences between our work and theirs, we have modified it to the following:
> > >
> > > > Sanford et al. (2024) focus on characterizing the minimum depth required to complete L-step reasoning, proving under an idealized setting that ignores attention masks and feed-forward networks that a Transformer with depth O(log L) can achieve L-step reasoning, thereby providing a lower bound on the number of layers for reasoning tasks. In contrast, our work adopts architectural assumptions more consistent with practical Transformers (including attention masks and FNNs) and studies explicit upper and lower bounds on the number of reasoning steps that a Transformer can complete under fixed-depth conditions. More importantly, we reveal a key structural factor previously unaddressed in the literature: even when atomic facts correspond to the same reasoning chain, different input orderings can lead to significant differences in the model’s reasoning ability, indicating that the upper bound of reasoning depends not only on the model’s structural capacity but is also profoundly influenced by the symbolic order structure of the input.
> > >
> > > We once again express our sincere gratitude for your thoughtful comments and the time you have devoted to reviewing our work. Your feedback has greatly helped us refine both the technical clarity and the presentation of our contributions. We hope that the revisions and additional explanations address your concerns.

---

### Author Response · Authors · 2025-11-20

We would like to express our sincere gratitude to the reviewers for their valuable insights and comments, which we have incorporated into our revised manuscript. In response to the reviewers' suggestions, we have devoted considerable time and effort to implement several significant revisions, including the addition of supplementary experimental results and numerous improvements to the overall presentation of the paper.

Below are the specific modifications and additions we have made. The changes in the revised manuscript are marked in red.

**[Additional numerical experiments]**

We note that multiple reviewers expressed concern that the experiment with L=3 shown in our paper is insufficient to support our conclusions. Considering that for L > 3 the computational cost and optimization difficulty grow exponentially with L, we instead use a simulation program that models information propagation to justify our conclusions.

For the case L=3, we enumerated all possible serialized representations of the reasoning chain a1 → a2 → … → a11, totaling (11! = 3,628,800) different possibilities. Across all possibilities, the minimum information content of the final node is 5 and the maximum is 10, which matches the theoretical lower and upper bounds. The corresponding experiments can be found in Section 7.

In addition, we supplemented our study with experiments on a 2-layer Transformer. The single-step reasoning test accuracy reaches 100%, while the two-step reasoning accuracy is 8.4% (indicating failure). This result is consistent with our theoretical findings. The relevant results can be found in Appendix F.

**[Optimizing the paper structure to clarify motivation and logic]**

Following the reviewers’ suggestions, we improved the organization of the manuscript. We added the following content in the Introduction to make the motivation of this work clearer:

In this work, starting from the information propagation patterns induced by the attention mechanism in Transformer architectures, we propose a set of information propagation rules that faithfully simulate how Transformers transmit information. Using an abstract symbolic multi-step reasoning dataset, we then rigorously analyze, under these propagation rules, the theoretical limit of the reasoning depth that can be achieved by an $L$-layer Transformer \red{with a sufficiently large hidden dimension} in a single forward pass. Our analysis shows that the achievable number of reasoning steps lies between $O(2^{L-1})$ and $O(3^{L-1})$.

In addition, we removed some redundant definitions. Specifically, we now use a single paragraph to cover Definitions 4.4, 4.5, and 4.8 from the original manuscript, and we deleted these three definitions.

**[Clarification regarding model dimensionality]**

We note that the reviewers mentioned concerns about the practical capacity of the model. First, this study focuses on the mechanism by which a Transformer performs parallel multi-step reasoning, as well as the maximum number of reasoning steps that a Transformer can achieve under this mechanism. To simplify this problem, we consider precise information-matching rules while ignoring the limitations imposed by model capacity.

Second, in Appendix D of the original manuscript, we construct a Transformer architecture that can match our information propagation rules and point out that, under exact matching, the required hidden-space dimension is on the order of O($n * 3^L * d$), where n is the sentence length, L is the number of layers, and d is the vocabulary size. However, the constructed vectors are extremely sparse, and we do not need the exact matching required by this construction—approximate matching is sufficient (because layernorm and softmax can amplify small differences). According to *compressed sensing theory* and the *Johnson–Lindenstrauss lemma*, the intrinsic dimension we actually need can be far smaller than the dimensionality of the current construction. We will provide a rigorous proof in future work.

[1] E. J. Candes and T. Tao, "Near-Optimal Signal Recovery From Random Projections: Universal Encoding Strategies?"

[2] Johnson, W. B., & Lindenstrauss, J. (1984). Extensions of Lipschitz mappings into a Hilbert space.

---

### Author Response · Authors · 2025-12-01

Deer Area Chair,

Thank you for taking over the review of our submission after the recent procedural adjustments. Since you can already see all discussion and author responses during the rebuttal period, this letter serves only as a supplementary summary: outlining the origin and development of the reviewers’ disagreements, what key evidence and explanations we added during the rebuttal, and which points—under normal discussion procedures—would have had the opportunity to be further clarified and move toward consensus.

**Summary of the Paper**

In this paper we propose a set of information propagation rules based on Transformers and utilize symbolic reasoning tasks to theoretically analyze the limit reasoning steps. Our main contributions are:

- We propose and prove that the number of reasoning steps achievable by a fixed-depth Transformer has clear upper and lower bounds(i.e. 2^{L-1} and 3^{L-1}), which has not been systematically discussed in the existing literature.

- **A very critical and interesting phenomenon we discovered in the derivation is that the ordering of reasoning pairs in the input sequence affects the model’s performance, even when they correspond to the same reasoning chain**. For example, for the reasoning chain a→b→c→d→e, we find that when the input pattern is a→b, b→c, d→e, c→d, a 3-layer model can correctly predict that the 4-step reasoning result of a is e. However, when the input pattern is a→b, b→c, c→d, d→e, the model produces an incorrect result. This counter-intuitive phenomenon has not been discussed in prior literature. It further shows that the boundary of the model’s capability is not only determined by the model’s own capacity, but also has a deep relationship with the symbolic order structure in the input.


**Review Landscape and Signals During the Discussion Period**

Initial scores: owtv = 2, swgx = 8, qBEY = 8, HCxV = 2.

The two high-score reviewers (swgx, qBEY): after reading our response, both chose to maintain their original high scores and expressed support and recognition for our work.

owtv (2 → 4): this reviewer explicitly acknowledged our additional numerical validations and clarifications, raising the score from 2 to 4. At the same time, they raised a further key question: they hoped we could clearly explain the essential difference between our work and Transformers, parallel computation, and logarithmic depth (Sanford et al., 2024). We believe this is a point that “can be fully clarified, and whose clarification is likely to significantly increase recognition”: the two lines of work investigate different research questions and their conclusions focus on different aspects (Section 2 below summarizes the key points of our clarification). Therefore, we believe that under a normal continuation of the discussion period, there remained room for owtv’s score to increase further.

HCxV (2): this reviewer raised additional comments and suggestions during the discussion period, focusing on two core concerns: whether the theoretical assumptions are overly idealized, and whether the assumptions are sufficiently supported by empirical observations/experiments. We note that the reviewer did not “simply reject and stop engaging,” but continued to raise concrete questions, which usually implies that the reviewer believes the work still has academic value but requires stronger empirical alignment and motivational clarification. In our second-round response, we emphasized more systematically: our information-propagation assumptions originate from existing empirical studies and our own experimental observations, and we supplemented finer-grained experimental analyses and explanations of phenomena. Under a normal process, with a few more rounds of clarification regarding assumptions grounded in experimental observations and consistency between empirical phenomena and theoretical predictions, we believe there was space to encourage the reviewer to reconsider the score.

HCxV (2): During the discussion period, this reviewer maintained high-intensity communication with us and provided further comments and suggestions. The reviewer’s main concern was that our theoretical assumptions appear inconsistent with the experimental results. However, this concern mainly arose from a misunderstanding of some key aspects of our work; in our view, the theory and experiments are in fact highly consistent. In our second-round response, (1) we emphasized more systematically that our information-propagation assumptions come from existing empirical studies and our own experimental observations, and (2) we supplemented finer-grained experimental analyses and explanations of the phenomena. Under a normal process, if we had been able to continue several more rounds of clarification on “assumptions grounded in experimental observations” and “consistency between empirical phenomena and theoretical predictions,” we believe there would have been room to encourage the reviewer to reconsider the score.

---

> ### Author Response · Authors · 2025-12-01
>
> **Key Additions We Made During the Rebuttal Period**
>
> (1) Differences from existing work Sanford et al., 2024
>
> In our follow-up discussion with owtv, we explained: the work by Sanford et al. primarily proves that a Transformer with depth O(log L) can realize L-step reasoning, thereby providing a lower bound on the number of layers required to complete a given number of reasoning steps.
>
> In contrast, our research also focuses on another “reverse” question: **Given a fixed depth, how many reasoning steps can a model complete at most? Do there exist theoretical upper and lower bounds for this achievable number of reasoning steps?**
>
>
> Our main contributions are:
>
> - We propose and prove that the number of reasoning steps achievable by a fixed-depth Transformer has clear upper and lower bounds(i.e. 2^{L-1} and 3^{L-1}), which has not been systematically discussed in the existing literature.
>
> - A very critical and interesting phenomenon we discovered in the derivation is that the ordering of reasoning pairs in the input sequence affects the model’s performance, even when they correspond to the same reasoning chain. This counter-intuitive phenomenon has not been discussed in prior literature.
>
> (2) Additional experiments and validations
>
> To address reviewer HCxV’s concerns about empirical support, we emphasized and supplemented two key pieces of evidence for the L=3 case:
>
> - For the lower bound of reasoning steps, i.e., s = 3, the model can theoretically perform three steps, and empirically its test accuracy generalizes to 100% (Fig. 10).
>
> - For the upper bound of reasoning steps, i.e., s = 4, we theoretically prove that the Transformer has strong generalization ability only on data satisfying specific order relations. The proportion of such sequences in the entire dataset is about 33%, and Fig. 10 shows that in the early stage of training, the model’s generalization accuracy is indeed around 33%. **We further provide additional analysis of the experimental results (Fig. 11, 12)**. We find that: when the sequences obey our expected order relations, the Transformer can achieve over 85% accuracy on such sequences (Fig. 11), and the information-propagation paths match our theoretical results (Fig. 12, i.e., the first attention layer performs position matching, and subsequent attention layers perform same-token matching). However, when sequences do not satisfy our expected order relations, the Transformer performs poorly on this part of the dataset. This shows that the Transformer indeed develops order preferences during training, and these order preferences can be predicted by our theory.
>
> (3)Regarding concerns about model capacity
>
> In our response, we distinguish between *inherent expressive power* and *Dimension-Constrained Capacity* with this paper primarily focusing on the former. We also clarified that: the previously seemingly large dimension estimate arises from an upper-bound construction based on exact matching, but real attention + softmax allows fuzzy matching, so the actual intrinsic dimension required may be significantly lower (and we provided a compressed sensing and JL lemma explanation in our response). For example, for x = [0.4, 0.3, 0.3], softmax(100x) $\approx$ [1, 0, 0]. More importantly, we also reported the empirical phenomenon that “a reasonable hidden dimension suffices to obtain good generalization, and performance improves as the dimension increases,” showing that the study is not detached from practical realizability.
>
> Since the discussion period was forced to end early due to the incident, we were unable to receive further questions or feedback from reviewers after our second-round response. But judging from the signals that had already appeared in the discussion period: owtv had already raised the score based on the incremental content and raised a key question that could be clarified; HCxV was still raising concrete concerns rather than ending communication. The purpose of this letter is to present to you these key increments that would have helped move toward consensus in the subsequent discussion, so that you may more comprehensively evaluate, when writing the meta-review, which disagreements we have already reinforced or could have been further resolved under normal communication. Thank you very much for your time and effort.

---

### Meta-Review · Area_Chair_uu9b · 2026-01-05

**Summary:**

This work presents a theoretical analysis of the number of reasoning steps a Transformer model can perform in a single forward pass on a symbolic multi-step reasoning task, under a set of defined information propagation rules. They prove that for an L-layer Transformer, the maximum number of symbolic reasoning steps lies between $O(2^{L−1})$ and $O(3^{L−1})$.

The paper received 4 reviews with them being really divergent: 2 accepts and 2 rejects. The major points of contention among the negative reviewers were as follows:

* The proofs do not actually generalize to a soft-attention transformer and does not show that trained transformers actually implement those rules.  Reviewer owtV mentions and i quote: "The proofs are also dimensionality-agnostic, implicitly assuming each node can store arbitrarily many distinct reasoning items regardless of hidden size."

* The negative reviewers also question the underlying Information Propagation Rules as these, representing a transformer, is a program with infinite memory and thus does not meaningfully approximate real transformers.

* Weak experiments with only a single depth transformers being used for the evaluation in the initial version.

**Reviewer Concerns:**

The authors provided a detailed rebuttal and tried to address the major concerns of all the reviewers. Reading the rebuttal I think that the 1st question is moderately well answered. One thing that remains unanswered is that the analysis exclusively studies single-pass reasoning capacity, whereas iterative strategies such as CoT and ToT are used in real models along with multiple passes. Thus, how will the bounds generalize to such practical models is not discussed.

The 2nd concern remains largely unsolved since it still seems that the propagation rules are defined for theoretical convenience rather than being derived from empirical attention patterns and thus the question of whether the real learned weights obey these rules or if they only hold under ideal conditions remains unanswered.

The authors added experiments with new depths so I assume this is answered although whether these bounds hold for larger depths is a difficult question to answer.

**Reviewer Scores:**

Based on the rebuttal 1 reviewer did raise their score to 4 although mention that their central question of empirical validation as well as connection to previous work is not well answered. So I do not see them increasing their score based on the rebuttal.
I also do not oversee an improvement in other scores.

Overall, I think that the theoretical analysis is interesting and could be useful, there is still a lot of unanswered questions which will require more work. Thus, I recommend rejection at this point.

---

### Decision · Program_Chairs · 2026-01-26

Reject